# AIMS.au: A Dataset for the Analysis of Modern Slavery Countermeasures in Corporate Statements

**Adriana Eufrosina Bora**[1,2]    **Pierre-Luc St-Charles**[1]    **Mirko Bronzi**[1]
**Arsène Fansi Tchango**[1]    **Bruno Rousseau**[1]    **Kerrie Mengersen**[2]
[1]Mila - Quebec AI Institute
{adriana.eufrosina-bora, pl.stcharles, mirko.bronzi}@mila.quebec
{arsene.fansi.tchango, bruno.rousseau}@mila.quebec
[2]School of Mathematical Sciences, The Queensland University of Technology
adrianaeufrosina.bora@hdr.qut.edu.au
k.mengersen@qut.edu.au

## Abstract

Despite over a decade of legislative efforts to address modern slavery in the supply chains of large corporations, the effectiveness of government oversight remains hampered by the challenge of scrutinizing thousands of statements annually. While Large Language Models (LLMs) can be considered a well established solution for the automatic analysis and summarization of documents, recognizing concrete modern slavery countermeasures taken by companies and differentiating those from vague claims remains a challenging task. To help evaluate and fine-tune LLMs for the assessment of corporate statements, we introduce a dataset composed of 5,731 modern slavery statements taken from the Australian Modern Slavery Register and annotated at the sentence level. This paper details the construction steps for the dataset that include the careful design of annotation specifications, the selection and preprocessing of statements, and the creation of high-quality annotation subsets for effective model evaluations. To demonstrate our dataset's utility, we propose a machine learning methodology for the detection of sentences relevant to mandatory reporting requirements set by the Australian Modern Slavery Act. We then follow this methodology to benchmark modern language models under zero-shot and supervised learning settings.

## 1 Introduction

The proliferation of legal mandates requiring corporations to disclose specific information regarding their human rights and environmental actions has necessitated the development of robust platforms and tools to facilitate compliance analysis. In line with other countries, the Australian Modern Slavery Act of 2018 (the AU MSA, or the "Act", Australian Government, Act No. 153, 2018) requires over 3000 corporations to detail their efforts to combat modern slavery within their operations and supply chains (Australian Government, Attorney-General's Department, Modern Slavery Business Engagement Unit, 2023). The resulting number of freeform, annually-published statements worldwide exceeds the resources allocated by supervisory bodies to monitor modern slavery compliance. While numerous datasets have been created to support the development of automated approaches for text summarization and understanding such as in the medical and legal domains (Zambrano Chaves et al., 2023; Guha et al., 2023), there exists a gap in large-scale datasets that help detect and extract relevant information explicitly mandated by this type of legislation from corporate statements. We address this gap by introducing a novel dataset tailored to the analysis of modern slavery statements, focusing on the extraction of pertinent information as specified by the Act.

Traditional approaches in machine learning for legal and declarative text understanding have primarily centered on summarization and synthesis (Abdallah et al., 2023; Niklaus et al., 2024; Martinez-Gil, 2023). These methodologies aim to condense lengthy documents into concise summaries or to interpret their key points and link them with a given query. The introduction of legislation that

mandates corporations to share information without enforcing a document template motivates a shift from summarizing content to precisely identifying and extracting relevant disclosures while avoiding text distractions. These distractions encompass corporate jargon or assertions that, despite appearing positive, do not contain substantial actions or pertinent information.

This paper introduces a new, publicly available dataset that can significantly advance machine learning research on modern slavery statements. This dataset is meticulously curated to aid in developing extraction processes that accurately identify and make accessible all relevant information required by the legislation for further analysis. This is made possible by manual annotations aimed at determining whether each sentence contains any mandated information. It provides the largest and most consistent resource specifically designed for retrieving information mandated by legislation. Unlike previous efforts, which were often too inconsistent and relied on broader, self-defined metrics, our dataset includes a substantially larger number of annotated statements aligned strictly with the mandatory criteria of the Australian Modern Slavery Act. Developed with advice from various key stakeholders, including the Australian government team responsible for monitoring the Act, this data set ensures direct legal relevance and robustness for compliance monitoring. What is more, our benchmark results demonstrate that fine-tuned models trained on our annotations significantly outperform larger language models in zero-shot conditions, underscoring the dataset's value. By releasing this resource and its supporting materials as open source, we aim to foster broader adoption and further research, potentially enabling models to generalize to other legal frameworks with minimal adjustments and reducing the need for future large-scale annotation efforts.

This paper is organized as follows. First, we provide a short background on the Australian modern slavery legislation (the Act). Next, we detail the construction steps of our dataset, which include the careful design of specifications used by annotators to ensure that relevant information is captured as accurately as possible. We detail the distribution and preprocessing of corporate statements into text that models can ingest, and the distribution of the relevant text extracted by annotators. We also discuss the creation of high-quality annotated statements subsets, which are essential for effective model validation and testing. Next, we describe a machine learning methodology specifically tailored for detecting sentences that are relevant to each mandatory reporting requirement outlined by the Act. This methodology provides an approach to differentiate between substantive disclosures and non-relevant content, for zero-shot and supervised learning settings. We then present benchmarking results that demonstrate the performance of large language models in both zero-shot and supervised settings. Subsequently, we discuss related works and argue that our findings offer insights into the capabilities and limitations of current works in handling this complex task. Finally, we conclude by elaborating on limitations of this paper and by outlining directions for future works.

## 2 BACKGROUND

Modern slavery describes situations where coercion, threats, or deception are used to exploit victims and deprive them of their freedom. It encompasses any situation of exploitation that a person cannot refuse or leave due to threats, violence, coercion, deception, or abuse of power (Walk Free, 2022a). In 2021, an estimated 50 million people were subject to modern slavery, with 28 million in forced labor. This issue is believed to affect all industries worldwide, with industries such as agriculture, manufacturing, and construction being at higher risk.

A critical impediment to eradicating modern slavery is the lack of transparency and accountability in corporate efforts to eliminate it from their supply chains. Without clear due diligence, reporting requirements and oversight, it is difficult to hold companies responsible for unethical practices and recognize those that adhere to ethical standards. To address this issue, many governments have enacted legislation mandating companies to increase transparency in their supply chains. The movement began with the California Transparency in Supply Chains Act of 2010, which required large retailers and manufacturers doing business in California to disclose their efforts to eradicate slavery and human trafficking from their supply chains. This was followed by the UK's Modern Slavery Act of 2015, the first national law of its kind, mandating companies to publish a slavery and human trafficking statement approved by their governing body and posted on their website. However, these early laws primarily focused on disclosure without specifying mandatory reporting criteria or robust enforcement mechanisms (McCorquodale, 2022).

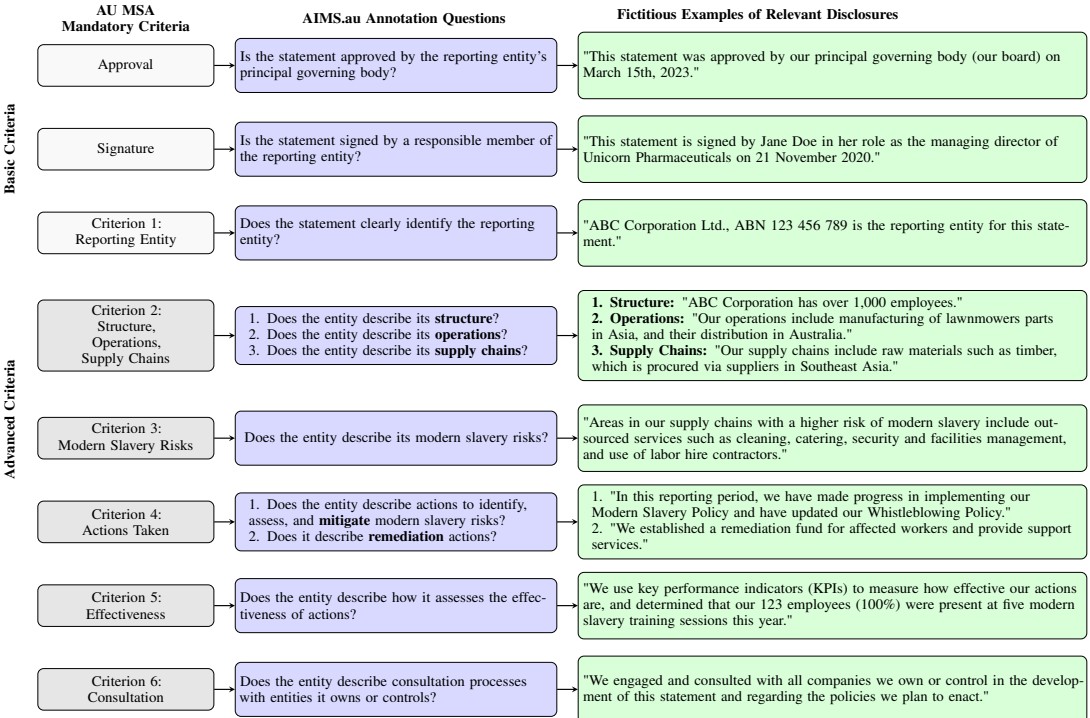

Figure 1: Correspondences between the AU MSA Mandatory Criteria and the questions designed for the annotation of the proposed AIMS.au dataset, with fictitious examples of disclosures that could be found in statements published by reporting entities.

The Australian Modern Slavery Act of 2018 is the first legislation to introduce mandatory reporting criteria; see Figure 1 for examples. These mandatory reporting requirements apply to companies with revenues exceeding AU$100 million and compel them to submit an annual statement where they report on specific criteria highlighting actions taken to address modern slavery within their operations and supply chains. Other similar legislation possess compatible mandatory criteria; a comparison is provided in Appendix J. Yet, despite such legislation, many companies provide vague and distracting disclosures that hinder effective monitoring and progress. We give examples of such declarations in Appendix C. The growth in the volume of corporate statements published annually also makes it difficult to hold corporations accountable for misleading statements and broken promises. As a recent report (Dinshaw et al., 2022) highlights, for a set of modern slavery statements published by 92 reporting entities and analyzed by experts: 1) the majority did not meet basic reporting requirements; 2) only a third provided evidence of some form of effective action to tackle modern slavery risks; and 3) over half of all promises made regarding future actions in the past were unfulfilled in later statements. We believe that this type of review is necessary across all modern slavery statements published annually, but modern tools to assist experts in their analysis are required to scale this process. We believe that the AIMS.au dataset could serve as a key milestone in the development of such tools, providing a foundation for further advancements in this area.

Note that we chose to focus on the Australian Modern Slavery Act (MSA) due to its strong alignment with reporting criteria in other laws, its comprehensiveness, and its established track record of enforcement, which has resulted in a substantial number of compliance statements. Furthermore, its supervisory body actively verifies whether companies meet their obligations. These factors make the Australian MSA an ideal baseline for developing the AIMS.au dataset, which can support transfer and adaptation studies and serve as a foundation for tools tailored to other legal contexts, such as those in the UK or Canada. We expand on this in Appendix J.

## 3 DATASET DESCRIPTION

Our proposed dataset, AIMS.au, is a combination of modern slavery statements published in PDF format by corporate entities and of sentence-level labels provided by human annotators and domain expert analysts. As shown Figure 2, a total of 5,670 statements were processed by hired annotators with respect to the three basic reporting criteria of the Act to determine whether each statement is approved, signed, and has a clearly-identified reporting entity. The other more advanced reporting criteria (previously shown in Figure 1) involve nuanced interpretations and required higher levels of scrutiny; for these, a subset of 4,657 statements that were found to be of a reasonable length were double annotated by hired annotators. Lastly, two specialized "gold" subsets with each 50 unique statements were created by experts to allow for evaluations with higher reliability across all criteria. The first gold subset was annotated by a single expert and validated through team discussions, while the second gold subset underwent a collaborative annotation process involving three experts. In all cases, disagreements were discussed until the experts achieved consensus. Given all these data subsets, we propose that future research utilizes statements annotated by hired workers for model training, statements in the first "gold" subset for model validation, and statements in the second gold subset for model testing; this should provide optimal trust in model performance assessments.

The final result is over 800,000 labeled sentences across 5,731 unique modern slavery statements covering 7,270 Australian entities between 2019 and 2023. As outlined in the following section and in Appendix E, the annotation process was highly complex and resource-intensive, far from being a low-cost crowdsourced task. This process took over a year and a half to complete and required a large team of highly skilled annotators, working under the close supervision of experts. Below, we detail the steps involved in the collection and preprocessing of statements, we discuss the choices that were made before and during the annotation process, and we provide summary statistics of our resulting dataset.

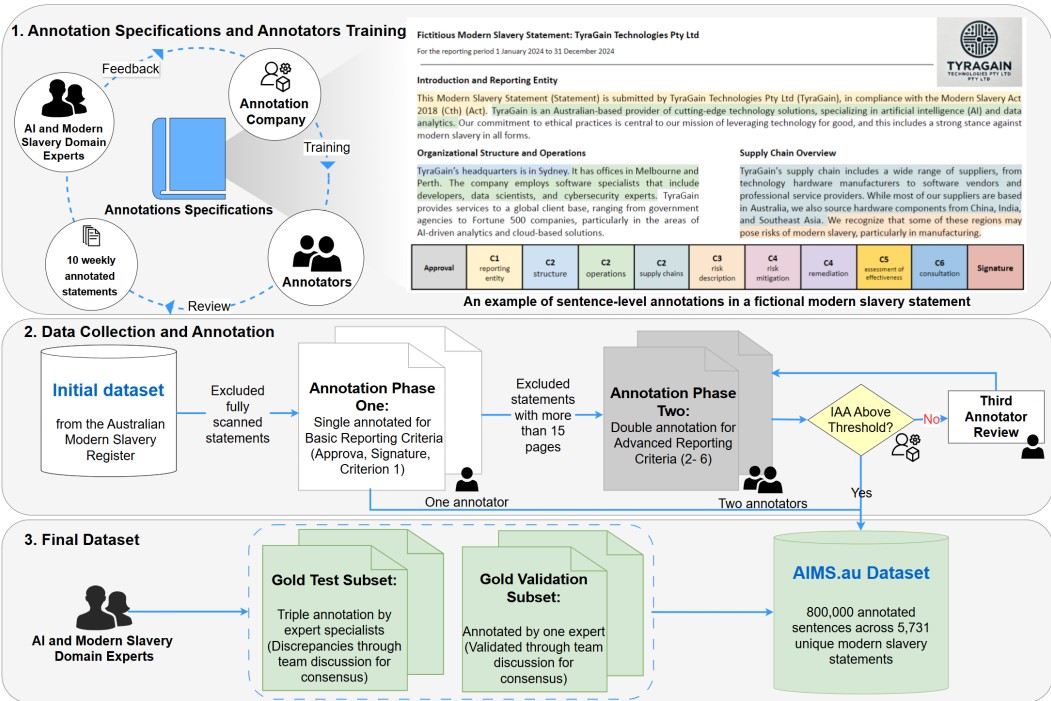

Figure 2: Overview of the annotation workflow for the AIMS.au dataset.

**Statement collection process.** Modern slavery statements to be annotated were first identified based on the already published and available PDF statements hosted on the Australian Modern Slavery Register (Australian Government, Attorney-General's Department, 2024) as of April 2023. We eliminated statements that were fully scanned from our selection to simplify the text extraction process and to minimize errors that would be due to the use of Optical Character Recognition (OCR)

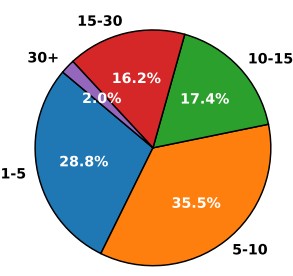
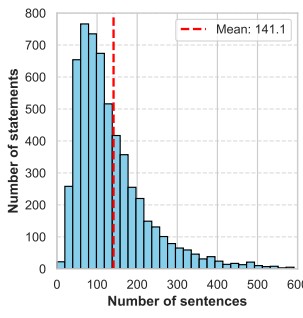
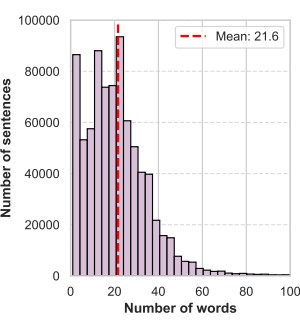

(a) Page count per statement.  (b) Sentence count per statement.  (c) Word count per sentence.

Figure 3: Overview of the distribution of text across the 5,731 statements in our proposed dataset.

tools. The 5,731 statements are associated with a total of more than 7,200 entities and 10,000 trademarks spanning more than 20 industrial sectors. These statements are issued by a diverse range of legal entities, including public and private companies, partnerships, sole proprietorships, trusts, government-owned corporations, and non-profit organizations. On average, each statement comprises 10.4 pages and 141 sentences, resulting in a combined total of nearly 60,000 pages and over 800,000 sentences. Other information on the data distribution is summarized in Figure 3 and in Appendix D.

**Conversion of text into sentences.** The text was extracted from the PDF statements using PyMuPDF ("fitz", PyMuPDF Contributors, 2024) as well as ABBYY FineReader PDF (a commercial software). This text was then split into sentences using regular expressions that considered various start and end-of-sentence tokens, including classic punctuation (such as periods, exclamation marks, and question marks) and more unusual tokens (such as bullet points). Special care was taken to avoid issues related to abbreviations with periods to ensure accurate sentence boundaries. Additionally, we removed section numbers and prefixes where possible at the start of sentences using regular expressions. Edge cases such as nested punctuation and enumerations were also handled using regular expressions to improve the accuracy and quality of sentence splitting. Once the sentences were obtained, we retained only those containing at least one two-letter word to eliminate orphaned text resulting from fragmented tables, page numbers, and other non-sentence elements.

**Development of the annotation specifications.** The Mandatory Criteria listed in Section 2 highlight two important challenges in the analysis of modern slavery statements with respect to the Act: 1) there is no explicit definition of what constitutes "relevant" information, or a specified amount of relevant information required to meet the Act's mandates; and 2) the criteria are fairly high-level, necessitating interpretation and refinement into more precise and actionable items that annotators can verify. To address these challenges, we reviewed guidance material and supplementary examples (Australian Government, Attorney-General's Department, Modern Slavery Business Engagement Unit, 2023), and consulted with the Australian Attorney General's Department to propose a breakdown of these criteria into more granular labeling tasks. Although labeling relevant information at the statement or paragraph level could be simpler than at the sentence level, it would offer limited utility for model training, evaluation, and downstream applications. Additionally, training laypersons to provide consistent and accurate high-level labels would be challenging and prone to significant subjectivity. Consequently, we translated the seven mandatory content criteria into eleven questions designed to be answered by extracting relevant sentences within the context of the entire statement. This approach was detailed in the annotation specifications provided to annotators, complete with training examples. The annotation specifications document is available as supplementary material with this paper. It was developed iteratively by a multidisciplinary team, where refinements alternated with small rounds of annotations to validate the proposed changes. The final version of the document was chosen based on its effectiveness in helping annotators avoid cognitive overload, minimizing inconsistencies in the annotations, and maintaining a reasonable large-scale annotation cost. A comprehensive description of the annotation labels associated with each of the eleven questions can be found in Appendix D.

**Annotator selection and training.** Prior to the annotation of our dataset, we conducted preliminary experiments using language models that highlighted the need for a human-driven annotation process. Specifically, language models did not seem able to provide high-quality labels that would directly

Table 1: Agreement scores averaged across all double-annotated statements. We report the intersection over union (IAA) and Cohen's Kappa (CK). The two scores are relatively comparable except for the most imbalanced criterion (C4, "remediation") whose CK score is more negatively impacted.

| Question | IAA | CK |
|---|---|---|
| C2 (operations) | 0.66 | 0.76 |
| C2 (structure) | 0.67 | 0.75 |
| C2 (supply chains) | 0.75 | 0.82 |
| C3 (risk description) | 0.67 | 0.73 |
| C4 (remediation) | 0.93 | 0.77 |
| C4 (risk mitigation) | 0.53 | 0.58 |
| C5 (effectiveness) | 0.69 | 0.68 |
| C6 (consultation) | 0.94 | 0.86 |
| Overall | 0.73 | 0.74 |

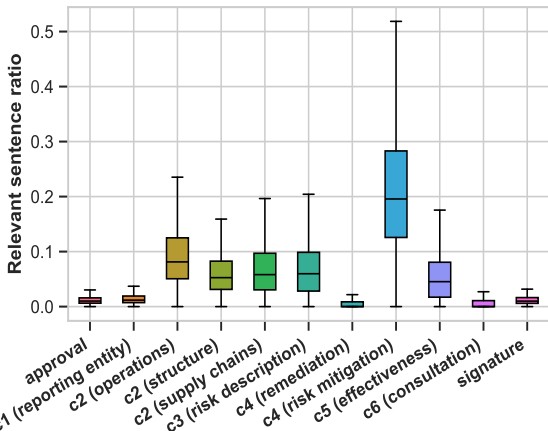

Figure 4: Distribution of relevant sentences found by annotators over the total number of sentences per statement for our eleven questions.

be adequate for subsequent analyses of modern slavery statements due to hallucinations and due to the impact of vague and distracting text. In fact, even experts can interpret legislative requirements differently and have varying opinions on the relevance of vague language depending on the context. This variability suggests that the most challenging questions should ideally be addressed by multiple annotators. However, assembling a large enough team of already-trained experts to annotate our entire dataset was impractical. Therefore, we engaged a private annotation company to provide workers with a strong understanding of English. We ensured that the company agreed to our contractual clauses on modern slavery, asking for the annotators to be fairly compensated and properly managed by the company; further details are provided in Appendix E. The annotators received training based on our annotation specifications and a set of 20 statements that we manually annotated after thorough internal reviews. This training included Q&A sessions and direct feedback on annotated examples. After the training phase, we initiated the broader annotation process.

**Quality assurance process.** As shown in Figure 2, the annotation process was divided into two phases. Initially, we focused on three simpler questions related to Criterion 1 (C1, "identifying the reporting entity") and to the approval and signature of the statement. This phase aimed to refine our interaction with annotators and clarify our quality expectations. Given that the accuracy of sentence-level labels depends on thorough extraction of relevant sentences, we emphasized that no relevant text should be overlooked and that entire statements needed to be read. This first phase lasted several weeks and targeted 5,670 statements, with a single annotator reviewing each statement. Each week, a random sample of 10 annotated statements was inspected to provide corrections and feedback. Upon completing this phase, we conducted a high-level review and found less than 1.2% of the annotations invalid due to improper formatting, mostly because dates for approval or signature were missed. The second annotation phase focused on the eight questions related to the remaining mandatory criteria. Here, two annotators independently reviewed each statement, and we set consistency targets using Inter-Annotator Agreement (IAA) thresholds. These eight questions are more challenging, so ensuring maximum consistency is critical. The IAA, defined as the intersection over union of relevant sentences found by the two annotators, was used to assess agreement. If the IAA for a statement was below the target threshold, a third annotator revisited and corrected the annotations. The IAA scores obtained for double-annotated statements are presented in Table 1, alongside Cohen's Kappa (CH) scores; we further discuss the usefulness of these scores in Appendix F. Due to time and budget constraints, this second phase included only statements shorter than 15 pages, which corresponds to 4,657 statements (82% of the total). We note that longer statements often required over 45 minutes to annotate, and were not necessarily more content-rich. For this phase, less than 1% of annotations were invalid due to improper formatting, primarily from text not being extracted from figures or tables that were tagged as relevant. Figure 4 illustrates the distribution of relevant labels across all sentences for our eleven questions. As expected, these plots reveal that the proportion of relevant sentences

among all sentences is low, with the highest average ratio reaching only 20% for the question related to C4 ("risk mitigation").

# 4 BENCHMARK EXPERIMENTS

**Splitting training and evaluation data.** For training and evaluation purposes, we cluster statements based on their associated entities and trademarks. We then assign each statement cluster to either the training set, validation set, or test set. This method ensures that similar statements made by related entities or by the same entity across different years are assigned to the same set, effectively preventing data leakage. For validation and testing, we created "gold" sets of statements that were annotated exclusively by extensively trained members of our team based on multiple rounds of review and discussion. Each of these sets contains 50 statements: the validation set was annotated by a single analyst, while the test set was annotated collaboratively by three analysts. These gold sets aim to minimize label noise, which is more prevalent in annotations provided by external annotators. Based on our observations, this noise primarily consists of omissions, such as missed relevant text. We emphasize that omissions are less problematic in the gold set annotations, where we use the union of multi-labeled sentences from multiple annotators; indeed, the likelihood of all annotators omitting exactly the same text is low. The statements in both gold sets were randomly selected based on clustering results while ensuring they were not used elsewhere, such as in the examples for the annotation specifications. We handled the statements and annotations with care (particularly those in the gold sets) to prevent indirect leakage to future generations of language models (Balloccu et al., 2024).

We detail limitations of our dataset in Section 6 and in Appendix F. For more specific details on the preparation of our dataset and on its contents, we refer the reader to Appendix D.

In this section, we outline our experimental setup and present the results of benchmarking various models for detecting sentences relevant to the mandatory reporting requirements of the Act. We evaluate the performance of these models under both zero-shot and fine-tuning settings to assess their effectiveness in extracting mandated information from statements. We then analyze the results to identify key insights and potential areas for improvement.

**Task definition.** Our proposed dataset includes a variety of labels that models could predict; these labels are detailed in Appendix D. For conciseness and clarity, we focus on a task that we believe will be of greatest interest to the machine learning community: predicting relevant or irrelevant labels according to our eleven questions. We frame this task as a sentence-level binary classification problem which we evaluate across the eleven questions using the F1 metric. We selected this metric over accuracy because it allows us to identify cases where models simply learn to predict all sentences as irrelevant, since those are over-represented in our dataset (see Figure 4).

For the statements that are double annotated by hired workers, we adopt a "union" label combination strategy, where a sentence is considered relevant if any annotator marks it as such. This approach addresses the possibility that individual annotators may have missed relevant text in some statements. We suggest that future works explore more sophisticated methods for leveraging annotator disagreements as a supervision signal. For our current experiments, models are evaluated exclusively using the subsets of "gold" annotated statements. Since these gold sets contain high-quality annotations, their smaller size (roughly 7000 sentences each) with respect to the overall dataset size should not significantly impact the reliability of model evaluations. Furthermore, this approach helps us, as well as future researchers, avoid incurring significant API usage costs when using state-of-the-art, closed-source language models for large-scale evaluations.

**Evaluated models.** We conduct our experiments using a range of language models that includes four open models — DistilBERT (Sanh et al., 2020), BERT (Devlin et al., 2019), Llama2 (7B) (Touvron et al., 2023) and Llama3.2 (3B) (Dubey et al., 2024) — and two closed models, namely OpenAI's GPT3.5 Turbo and GPT4o (see Appendix G for more details). We use the OpenAI and Llama3.2 (3B) models to evaluate zero-shot (prompt-based) approaches, and we compare them with DistilBERT, BERT, Llama2 (7B) and Llama3.2 (3B) models fine-tuned directly on statements annotated by hired workers. Our experiments are structured based on two input data setups: in the first ("*No context*" setup), models only have access to the target sentence being classified; in the second ("*With context*" setup), we provide additional context by including up to 100 words balanced before and after the

target sentence (see Appendix H for an example). These two input setups allow us to assess the impact of contextual information on model performance.

The open models DistilBERT, BERT, Llama2 (7B) and Llama3.2 (3B) are fine-tuned from self-supervised pre-training checkpoints available on the HuggingFace repository (Wolf et al., 2019). For DistilBERT and BERT, we fine-tune the full model weights, while for Llama2 (7B) and Llama3.2 (3B), we use the LoRA approach (Hu et al., 2021) to manage computation costs. All experiments are conducted on a A100L GPU with 80 GB memory using PyTorch. Token sequence lengths are capped at 512 for DistilBERT and BERT, and at 150 for Llama2 (7B) and Llama3.2 (3B), due to memory limitations. Models are trained with a batch size of 96 for DistilBERT, 64 for BERT, 32 for Llama2 (7B), and 64 for Llama3.2 (3B), using Adam (Kingma & Ba, 2014) with a fixed learning rate (0.00003). We select model checkpoints that maximize the Macro F1-score. Links to the model pages and checkpoint names are provided in Appendix G.

**Prompt design for zero-shot experiments.** Experiments with GPT3.5 Turbo, GPT4o and Llama3.2 (3B) zero-shot are conducted using prompt templates designed specifically and given in Appendix H. These templates were developed based on insights gained from five iterations of prompt exploration conducted on a small set of documents, while also following best practices on how to formulate intents, how to provide domain definitions, and how to constrain desired outputs (Ekin, 2023). The definitions provided in the prompt are taken from the Act and its guidance document (Australian Government, Act No. 153, 2018; Australian Government, Attorney-General's Department, Modern Slavery Business Engagement Unit, 2023), and are essentially a condensed version of the instructions given to the annotators. We leave the exploration of more sophisticated prompts, or very large prompts that may include multiple examples or even our entire annotation specifications document, for future works.

## 4.1 RESULTS

Table 2 presents results in the zero-shot setting. Alongside GPT3.5 Turbo and GPT4o, we include Llama3.2 (3B) for direct comparison within the same model architecture after fine-tuning. Both GPT3.5 Turbo and GPT4o outperforms Llama3.2 (3B) by a substantial margin. Notably, Llama3.2 (3B) exhibits a tendency to predict the criteria for almost all sentences, leading to poor F1 scores due to low Precision. This behavior also explains its relatively better performance on criterion with more positive examples, such as "C4 (risk mitigation)" (see Figure 4). In the "With context" experiments, GPT4o demonstrates significant performance improvements, whereas GPT3.5 Turbo shows a steep decline, defaulting to predicting the criteria for nearly every sentence, similar to the pattern observed with Llama3.2 (3B). We hypothesize that this discrepancy arises because GPT4o is better equipped to handle long prompts and inputs compared to GPT3.5 Turbo.

Table 2: F1 evaluation results for **zero-shot** approaches conducted using GPT3.5 Turbo, GPT4o and Llama3.2 (3B). Results in the "With context" case are unavailable for Llama3.2 (3B) due to time limitations.

| Question | No context | | | With context | |
|---|---|---|---|---|---|
| | GPT3.5 Turbo | GPT4o | Llama3.2 | GPT3.5 Turbo | GPT4o |
| Approval | 0.584 | 0.911 | 0.041 | 0.028 | 0.895 |
| C1 (reporting entity) | 0.148 | 0.378 | 0.054 | 0.031 | 0.427 |
| C2 (structure) | 0.371 | 0.661 | 0.168 | 0.097 | 0.616 |
| C2 (operations) | 0.268 | 0.616 | 0.172 | 0.167 | 0.601 |
| C2 (supply chains) | 0.317 | 0.543 | 0.211 | 0.174 | 0.556 |
| C3 (risk description) | 0.337 | 0.422 | 0.182 | 0.194 | 0.512 |
| C4 (risk mitigation) | 0.591 | 0.601 | 0.478 | 0.481 | 0.624 |
| C4 (remediation) | 0.269 | 0.548 | 0.055 | 0.048 | 0.555 |
| C5 (effectiveness) | 0.295 | 0.293 | 0.216 | 0.142 | 0.435 |
| C6 (consultation) | 0.383 | 0.481 | 0.050 | 0.038 | 0.620 |
| Signature | 0.684 | 0.480 | 0.091 | 0.030 | 0.763 |
| Overall (macro) | 0.386 | 0.439 | 0.156 | 0.130 | 0.600 |

We present evaluation results for all fine-tuned models jointly trained on the full eleven-question setting in Table 3. Results are significantly higher than the zero-shot case; in particular, fine-tuned Llama3.2 (3B), compared to the zero-shot results for the same architecture results in a increase in

performances from 0.156 to 0.694 Macro-F1. Overall, adding context to the input provides better results, with performances increasing for all the three models. Comparing the models, BERT and DistilBERT provides similar results, while Llama3.2 (3B) outperforms the other models by some margin; Llama2 (7B) instead provides the lowest results, which we speculate is due to having more capacity in the model weights, thus needing more fine-tuning iterations (see Appendix I.1 for more information).

Table 3: F1 evaluation results for jointly **fine-tuned** models on all eleven Mandatory Criteria questions. Llama2 (7B) results are available only for the "No context" case for computational constraints.

| Question | No context | | | | With context | | |
|---|---|---|---|---|---|---|---|
| | DistilBERT | BERT | Llama2 | Llama3.2 | DistilBERT | BERT | Llama3.2 |
| Approval | 0.957 | 0.965 | 0.889 | 0.940 | 0.955 | 0.964 | 0.932 |
| C1 (reporting entity) | 0.639 | 0.605 | 0.579 | 0.643 | 0.698 | 0.728 | 0.715 |
| C2 (structure) | 0.708 | 0.732 | 0.708 | 0.745 | 0.740 | 0.740 | 0.726 |
| C2 (operations) | 0.741 | 0.718 | 0.672 | 0.753 | 0.769 | 0.758 | 0.773 |
| C2 (supply chains) | 0.723 | 0.675 | 0.719 | 0.729 | 0.755 | 0.772 | 0.787 |
| C3 (risk description) | 0.653 | 0.660 | 0.650 | 0.686 | 0.705 | 0.741 | 0.752 |
| C4 (risk mitigation) | 0.631 | 0.614 | 0.602 | 0.611 | 0.629 | 0.640 | 0.667 |
| C4 (remediation) | 0.574 | 0.571 | 0.424 | 0.564 | 0.500 | 0.559 | 0.615 |
| C5 (effectiveness) | 0.533 | 0.483 | 0.242 | 0.527 | 0.491 | 0.560 | 0.500 |
| C6 (consultation) | 0.414 | 0.429 | 0.293 | 0.611 | 0.641 | 0.571 | 0.588 |
| Signature | 0.794 | 0.859 | 0.797 | 0.830 | 0.844 | 0.866 | 0.873 |
| Overall (macro) | 0.670 | 0.665 | 0.598 | 0.694 | 0.702 | 0.718 | 0.721 |

One final insight we emphasize is that, based on the presented results and our preliminary prompt engineering experiences, it is challenging to find prompts for zero-shot models that can match the performance of fine-tuned models. This highlights the necessity for high-quality, curated datasets like AIMS.au to allow for the reliable training and evaluation of language models. Additionally, this underscores the need for further exploration into the importance of context at various scales and the impact of vague and distracting text on large language models.

## 5 RELATED WORKS

**AI for analyzing supply chain disclosures under the California Transparency Act.** A few initiatives have considered machine learning to analyze statements in response to modern slavery legislation in the literature. For instance, LegalBench (Guha et al., 2023) proposed a benchmark for evaluating legal reasoning capabilities in language models. It consists of 162 tasks crafted by legal experts, and one of these is related to supply chain disclosures under the California Transparency in Supply Chains Act. The analysis of roughly 400 statements with one or two pages each using modern language models reveals only an accuracy of around 75%. Similar to the high-level decision process used by analysts, the proposed classification approach for this task relies on statement-level decision making for a limited set of questions. The researchers discuss in their report how model performance diminishes in tasks involving longer text or more numerous questions, which suggests that scaling this statement-level decision making strategy to much larger statements is probably not ideal.

**AI for the analysis of UK modern slavery statements.** Despite numerous studies analyzing a handful of modern slavery statements manually (details in Appendix A), only a few have investigated the use of machine learning to date. For instance, modern slavery statements from the UK are analyzed without supervision using topic modeling (Nersessian & Pachamanova, 2022; Bora, 2019). While this approach allows the authors to monitor disclosure trends and correlate them across different statements, it is unable to analyze each statement and differentiate vague claims and promises from substantive actions. Consequently, this approach cannot adequately verify compliance with respect to a specific legislation. Based on their analysis, the authors highlight that many companies "anchor" their disclosures in broader human rights language and that they emphasize their engagement in social causes in an effort to bolster their company's social reputation. This underlines the challenge of carefully avoiding distractions while assessing whether a statement contains mandated information.

UK modern slavery statements were also analyzed under an initiative of the Walk Free and of The Future Society organizations, resulting in an open-sourced project on GitHub (The Future Society, 2022) and a technical report (Weinberg et al., 2020). This initiative examined 16,000 statements and utilized approximately 2,400 annotated statements from WikiRate (WikiRate, 2023) for supervised machine learning experiments. In this work, classifiers were first trained to distinguish statements addressing specific mandatory content. These classifiers were then used to predict whether statements were correctly approved by a governing body based on annotator comments, keyword-based summaries, and n-gram representations. Limitations of this work noted by the authors include the difficulty in scaling to a large number of statements due to the usage of keyword-based and comment-based approaches, and due to the poor quality of the annotated statements. This previous research concluded that a stricter annotation process was necessary for developing new datasets and robust experimental protocols for subsequent studies. Moreover, as highlighted by other relevant studies on AI and sustainability reporting discussed in Appendix A, existing approaches continue to face difficulties in distinguishing concrete actions from vague text addressing relevant topics. Across these studies, many authors have emphasized challenges with training data quality and annotation biases. To the best of our knowledge, our paper now presents the largest annotated dataset globally, designed for machine learning research on modern slavery statements, while also marking the first academic study to scrutinize Australian modern slavery statements at scale, using machine learning techniques.

## 6 CONCLUSION

Our work presents a significant contribution to the field of machine learning and natural language processing by introducing a manually annotated dataset of modern slavery statements that is specifically curated to determine whether companies meet the mandatory reporting requirements outlined by the Australian Modern Slavery Act. This dataset is particularly valuable due to the unique and challenging nature of the sentence relevance classification task, characterized by vague and distracting text, as well as by the large amount of context required to understand the most complicated statements.

While this dataset provides a broad collection of annotated statements for future machine learning experiments, several limitations should be acknowledged. First, the reliance on external annotation services, despite extensive training and oversight, may introduce inconsistencies and biases in the labeled data. Annotators' varying interpretations of vague language and subjective judgment in identifying relevant information could affect the overall quality and consistency of the annotations. Another limitation involves figures and tables within statements, which cannot be easily analyzed without OCR or without a vision model. Although we can limit the scope of models to only focus on the extraction of relevant text that is not embedded inside figures or tables, some necessary context might sometimes be missing in order to understand a human annotator's decision. Lastly, we chose not to differentiate past and future information based on reporting periods to simplify the annotation process. In other words, corporations often detail past actions or future plans within their statements, and we consider all such disclosures relevant. This approach may complicate the assessment of whether a reporting entity meets the Act's requirements for a specific period, as it necessitates classifying relevant text according to each reporting period. We discuss potential solutions to these limitations in Appendix F.

We have conducted evaluations on modern language models, establishing performance benchmarks using both zero-shot and fine-tuning approaches. These benchmarks will serve as comparison baselines for future research in this domain. Our findings underscore the necessity of high-quality, curated datasets to reliably train and evaluate language models, especially in tasks that demand nuanced understanding and contextual analysis. Despite the promising results, there is significant room for future improvements, including the exploration of noisy label classification and more sophisticated context-handling techniques. Future research could also investigate the potential of integrating Vision-Language Models (VLMs, Bordes et al., 2024) to enhance the accuracy of information extraction in complex documents. Lastly, as we highlighted in Appendix J, this dataset can be considered a key resource for other relevant studies and tools tackling mandatory reporting legislation on business and human rights, such as the UK Modern Slavery Act UK Government (2015) and the Canadian Fighting Against Forced Labour and Child Labour in Supply Chains Act Canadian Government (2023).

ACKNOWLEDGEMENTS

Part of this research was supported by the National Action Plan to Combat Modern Slavery 2020-25 Grants Program, administered by the Attorney-General's Department of Australia. We sincerely thank Joumana Ghosn for her invaluable guidance in developing the research roadmap and annotation specifications. We would also like to express our gratitude to Akshatha Arodi and Jordan Bannister for their support in the final stages of completing this paper. We also extend our thanks to Jerome Solis, Allison Cohen, and Benjamin Prud'homme for their support in establishing and overseeing the project. We appreciate the collaboration of the Tasq.ai and WikiRate teams in the development of the annotation specifications and dataset. Additionally, we thank ABBYY for providing a free license to use their software for text extraction from PDFs.

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

## A    OTHER RELATED WORKS

**Australian modern slavery statement manual reviews.** Some academic groups and non-profit organizations have conducted analyses of Australian modern slavery statements to evaluate the legislation's effectiveness. For instance, in the work of Christ et al. (2019); Australian Council of Superannuation Investors (2021); Pham et al. (2023), researchers reviewed statements for 100, 151, and 300 companies listed on the Australian Stock Exchange, respectively. The Human Rights Law Centre, an Australian human rights group, also conducted extensive analyses, examining 102 and 92 statements in two separate studies (Sinclair et al., 2022; Dinshaw et al., 2022). The Domus 8.7 index, a benchmark initiative facilitated by the Catholic Archdiocese of Sydney, represents one of the more comprehensive analyses of statements conducted so far (ACAN, 2022). In this project, seventy interns manually reviewed 1,500 statements for a total investment of over 5,000 hours of work. Although these various studies all required significant effort over multiple years, they together cover less than 20% of all statements published so far on the Australian Modern Slavery Register (Australian Government, Attorney-General's Department, 2024), and none were scaled up in subsequent years. This underscores the significant challenges in analyzing modern slavery statements, even when only considering a single country and a single legislation. We also highlight that the data generated by analysts for individual statements is usually high-level and abstract (i.e. it consists of statement-wide labels indicating for example whether the issuer complies with the Mandatory Criteria, and justifications), and it is rarely made public or shared for research. Lastly, we note that the Australian Attorney-General's Department also performs an annual analysis that includes all statements in order to submit an annual report to Parliament (Australian Government, 2022). Unfortunately, we do not know the depth of this analysis, and the results are not made public directly. They are instead presented at an aggregated statistical level, making it difficult for researchers and organizations to track company-specific actions and promises.

**AI for the analysis of sustainability reports.** Several relevant studies exist that look at applications of artificial intelligence for compliance and document analysis beyond modern slavery. The Danish Institute for Human Rights (DIHR), for example, developed a text mining method based on a paragraph relevance classifier to analyze company sustainability reports against sustainability and human rights indicators, including modern slavery (Danish Institute for Human Rights, 2022). They processed approximately 145,000 UN system recommendations related to Sustainable Development Goal (SDG) targets and analyzed 9,374 reports with a simple text classifier trained to detect paragraphs related to key topics. In their conclusions, DIHR researchers highlight how relevant information may often be found in tables or figures that are challenging to convert into a machine-readable format for analysis. Other researchers also interested in sustainability disclosures studied the application of machine learning on Management Discussion and Analysis (MD&A) documents (Tian et al., 2023). In this case, 29,134 documents collected from the China Research Data Service (CNRDS) platform were analyzed using a Term Frequency, Inverse Document Frequency (tf.idf) weighting scheme to rank them based on their coverage of key sustainability topics. We note that this approach may also be sensitive to distractions, as, once again, it cannot differentiate concrete actions from vague text that covers a relevant topic.

As for advancements in the analysis of climate-related claims in corporate sustainability reports, several works should also be highlighted. Luccioni et al. (2020) developed ClimateQA, a language model that identifies climate-relevant sections in reports through a question-answering approach, processing 2,249 reports and emphasizing input quality. Ni et al. (2023) introduced ChatReport, which leverages language models to automate sustainability report analysis and compute conformity scores with international guidelines. This approach relies heavily on quality information retrieval and expert feedback. Nicolas Webersinke & Leippold (2022) proposed ClimateBERT, a model pre-trained on over 2 million climate-related paragraphs specialized for NLP in the climate domain. This led to a series of extensions, such as ClimateBERT-NetZero (Schimanski et al., 2023) for detecting net zero and emission reduction targets. Bingler et al. (2024) also explored climate disclosures and reputational risks with ClimateBertCTI, stressing the credibility of transition plans. Additionally, ClimateBERT and other language models such as BERT, RoBERTa, and Longformer were benchmarked on LobbyMap documents to estimate corporate climate policy engagement, highlighting the need for model fine-tuning across diverse formats (Morio & Manning, 2023). Across all of these works, many authors have highlighted that their proposed approach faced challenges with training data quality and annotation biases.

## B    DATA AVAILABILITY AND MAINTENANCE STRATEGY

The final dataset and annotation specifications can be accessed on Fighsare and Hugging Face. This dataset is associated with a Digital Object Identifier (DOI) to facilitate easy reference and citation. Additional details are available in the project's dedicated GitHub repository.

We make the dataset available in Activeloop DeepLake (Hambardzumyan et al., 2022) format. The Activeloop DeepLake format offers features specifically tailored for machine learning experimentation, including optimized PyTorch dataloaders, which facilitate seamless integration with machine learning workflows. This format is an open data format, promoting accessibility and ease of use. The dataset is packaged so that it directly contains raw PDF data as well as all metadata from the Australian Modern Slavery Register which may be useful for future studies. The content of the dataset is detailed in Appendix D in the data card style of Gehrmann et al. (2021); Suzgun et al. (2024).

The dataset is hosted on Figshare (Digital Science), an online open access repository, ensuring that it is freely available to the research community. By leveraging Figshare's robust infrastructure, we aim to provide a reliable and persistent platform for dataset access. To promote widespread use and proper attribution, the dataset will be licensed under the Creative Commons Attribution 4.0 International (CC BY 4.0) license. This license permits unrestricted use, distribution, and reproduction in any medium, provided the original authors and source are credited.

## C    EXAMPLES OF DISCLOSURES

In developing the annotation guidelines, our goal was to assist annotators in identifying concrete supporting evidence in statements. This was necessary as despite legislative mandates for specific disclosures, companies often provide vague, ambiguous, or distracting information that obstructs effective monitoring and progress. Table 4 provides, for all our questions related to the Mandatory Criteria of the Act, fictitious examples of: 1) relevant information; 2) irrelevant information due to ambiguity (i.e. due to a lack of context); 3) irrelevant information due to vagueness (i.e. unacceptable no matter the context); and 4) distracting information. These examples are inspired by the contents of real statements and highlight the significant challenge of distinguishing between relevant and irrelevant information.

Table 4: Examples of relevant and irrelevant information for questions related to the Mandatory Criteria of the Act.

| Question | Relevant information | Ambiguous information | Vague information | Distracting information |
|---|---|---|---|---|
| **Approval** | "This statement was approved by our principal governing body (our board) on March 15th, 2023." | "The ethics board approved the publication of this statement." | "Approval was received for this statement." | "Our code of conduct was approved by the board." |
| **C1 (reporting entity)** | "ABC Corporation Ltd., ABN 123 456 789 is the reporting entity for this statement." | (Company logo on the first page) | "This statement applies to numerous entities across our larger corporate family." | "Founded in 1980, X Corp. has a long history as a reporting entity in various jurisdictions." |
| **C2 (operations)** | "Our operations include the manufacturing of lawnmower parts in Asia and their distribution in Australia." | "We are a leader service provider in our sector." | "We operate globally." | "We produced 10,000 units last year, achieving a 15% increase in productivity." |
| **C2 (structure)** | "ABC Corporation has a hierarchical governance structure with over 1000 employees." | "This statement covers a number of wholly-owned subsidiaries." | "Our organization has a global structure leadership model." | "Here is the organizational chart for 2020 showing the department heads." |
| **C2 (supply chains)** | "Our supply chain includes raw materials such as timber, which is procured via suppliers in Southeast Asia." | "We may procure sensitive goods from higher-risk countries." | "We sometimes contract other companies for services." | "Our downstream supply chain distributes our products to over 10,000 customers." |
| **C3 (risk description)** | "Areas in our supply chains with a higher risk of modern slavery include outsourced services such as cleaning, catering, security and facilities management, and use of labor hire contractors." | "An assessment concluded that we have a low risk of modern slavery." | "Modern slavery has the potential to exist in the technology sector." | "We understand and have mapped our businesses risks with an extensive assessment strategy." |
| **C4 (remediation)** | "We established a remediation fund for affected workers and provided support services." | "We understand the importance of workers knowing their rights and we will directly address violations when needed." | "Remediation actions are a key priority for us." | "We deeply believe in the need for concrete remedies when cases are discovered, and the common industry practice is to terminate any contract with faulty suppliers." |
| **C4 (risk mitigation)** | "In this reporting period, we have made progress in implementing our Modern Slavery Policy and have updated our Whistleblowing Policy." | "We have established a zero-tolerance approach towards modern slavery." | "We have made sure that our suppliers comply with our policies." | "We are committed to maintaining the highest level of integrity and honesty throughout all aspects of our business." |
| **C5 (effectiveness)** | "We use key performance indicators (KPIs) to measure how effective our actions are, and determined that our 123 employees (100%) were present at five modern slavery training sessions this year." | "We conducted a review of our practices and spent time evaluating actions over the past year." | "Our team has spent time reflecting on our activities to enhance our approach." | "As part of our annual review process, we have also gathered and analyzed feedback from customer surveys." |
| **C6 (consultation)** | "We engaged and consulted with all companies we own or control in the development of this statement and regarding the policies we plan to enact." | "Our statement is the result of a comprehensive review process that engaged stakeholders from within our corporate family." | "We do not need to consult externally in the preparation of this statement." | "Our statement reflects a collaborative effort that draws from various perspectives within our organization." |
| **Signature** | "This statement is signed by Jane Doe in her role as the managing director of Unicorn Pharmaceuticals on 21 November 2020." | "Signed by John Doe, the company secretary of the Trustee." | "Signed by Jane Doe (21 November 2020)." | "Our company executives have all signed off on our modern slavery policies." |

# D AIMS.AU DATA CARD

## D.1 DATASET DESCRIPTION

**Dataset summary.** See Section 4 of the paper.

**Languages.** The dataset contains English text only.

**Domain.** Long, freeform statements made by corporate entities.

**Additional details.** The dataset contains modern slavery statements originally published in PDF format by Australian corporate entities between 2019 and 2023, metadata for those statements, and annotations (labels) provided by hired workers and ourselves. Additional unannotated statements published over the same period and beyond are also packaged in the dataset as supplementary data for unsupervised learning experiments.

**Motivation.** We publish this dataset to support the development and evaluation of machine learning models for extracting mandated information from corporate modern slavery statements. Our aim is to facilitate research in this domain and foster future efforts to assess companies' compliance with the Australian Modern Slavery Act and other similar legislation.

## D.2 META INFORMATION

**Dataset curators.** The dataset was compiled by the authors in collaboration with the annotators and management team from the hired annotation company.

**Point of contact.** Adriana Eufrosina Bora serves as the primary point of contact, while the public GitHub is also closely monitored.

**Licensing.** The dataset is released under the Creative Commons Attribution 4.0 International (CC BY 4.0) license.

**Funding sources.** Part of this research was supported by the National Action Plan to Combat Modern Slavery 2020-25 Grants Program, administered by the Attorney-General's Department of Australia.

## D.3 DATASET STRUCTURE

**Data format and structure.** We structure our dataset so that one "instance" corresponds to a single statement. Each statement is associated with a unique identifier, a PDF file, and a set of twelve metadata fields, all provided by the Australian Modern Slavery Register. These metadata fields are:

- Annual revenue;
- Countries where headquartered;
- Covered entities;
- Industry sectors;
- Overseas obligations;
- Reporting period end date;
- Reporting period start date;
- Publication date;
- Publication year in the register;
- Submission date;
- Associated trademarks;
- Statement type (normal or joint).

The PDFs are freeform, allowing reporting entities the flexibility to choose their format; some use a brochure-style layout, while others incorporate extensive background images or unique design elements. In addition to the provided metadata, we enhance these statements with several annotated fields, filled by our hired annotators or ourselves. These fields capture critical information such

as compliance with reporting requirements and supporting content, as detailed in the next few paragraphs.

**Data preparation.** See Section 4 ("Conversion of text into sentences") for information on text extraction. Following this step, we combine the raw PDF data (for researchers that intend on extracting the PDF contents themselves), its metadata, the extracted text (which, for ABBYY FineReader, includes the position of the text inside PDF pages and the OCR confidence levels), and the annotated fields into a single data archive. This archive is based on the Activeloop DeepLake format (Hambardzumyan et al., 2022) by default, and we provide a script to convert the dataset into HDF5 format.

**Annotated fields.** As detailed in Section 4 ("Development of the annotation specifications"), we translated the seven Mandatory Criteria of the Act into eleven questions. The questions are detailed in Appendix E, and are tied to a set of fields to be filled by annotators based on their answers. Specifically, the fields shared by all questions are:

- Label (yes/no/unclear): specifies whether the reporting entity has provided information that is relevant for the targeted criterion;
- Supporting text: contains all sentences found in the main body of the statement that are identified as relevant to justify the selection of the above label, or a justification if the "unclear" label was selected;
- Supporting visual element: contains several subfields that should be filled with 1) text found in relevant visual elements that also support the above label (if found in a format that allows direct extraction), 2) the page where these elements are found, and 3) the type of elements that were found (figures or tables);
- Scanned: a binary flag indicating whether relevant information was found in a "scanned" (i.e. embedded) format, for example in an image where the text cannot be copied;
- No supporting information: a binary flag indicating whether any information was found to justify the "no" label when it is used;
- Fully validated: a binary flag indicating whether our team has fully validated the annotations for this question, thus indicating whether the statement is part of a "gold" set or not.

Questions related to the presence of a signature or an approval have an extra "date" field which is filled with a signature or approval date (if available). The question related to the signature also has an extra "image" field, which is filled with a binary flag indicating whether the document contains an image of a signature. Lastly, the question related to the approval has an extra "joint option" field which is used in the case of joint statements to specify the type of arrangement used between the reporting entities.

Note that some fields ("no supporting information" and "scanned") are currently used solely for data validation and quality assurance purposes. Note also that the yes/no/unclear labels defined above would be used to determine whether companies have meet the Act's requirements, but these are not actually used in our current experiments. This is because these labels do not fully reflect the actual labels assigned by government analysts regarding whether entities have met the requirements of the Act. Hired annotators were instructed to mark "yes" for the label as soon as any relevant information was found. In practice, there is no agreed upon threshold for the amount of supporting evidence needed to ensure that a statement meets each Mandatory Criteria. We leave the refinement and evaluation of these labels to future works.

**Data split.** See Section 4 ("Splitting training and evaluation data").

**Data statistics.** Our dataset contains:

- Text, images, metadata, and raw PDF content for 8,629 modern slavery statements published as of November 2023. These statements were collected from the Australian Modern Slavery Register and processed using open-source and commercial PDF content extractors.
- Sentence-level annotations for 5,731 of these statements:
  - 5,670 statements published by the start of our annotation process (April 2023) were annotated for three out of our eleven mandatory content questions by hired workers;

- 4,657 statements published by April 2023 that are less than 15 pages were also double-annotated for the remaining eight questions by hired workers; and
- 100 statements sampled across the entire set were independently annotated for all questions by extensively trained members of our team. Of these, 50 were annotated by a single expert, and the remaining 50 were annotated by a team of three experts.

This dataset contains a total of more than 800,000 sentences that are labeled as relevant or irrelevant based on the Mandatory Criteria of the Australian Modern Slavery Act. The compressed size of the entire dataset is roughly 20 GB.

## D.4 DATASET CREATION

**Source data.** See Section 4 ("Statement collection process").

**Annotation process.** See Appendix E.

**Personal and sensitive information.** The dataset consists exclusively of publicly released statements available on the Australian Modern Slavery Register. As such, it contains no personal or sensitive information. All data included in the dataset are already in the public domain and have been made available for public access by the issuing entities.

**Data shift.** Potential data shifts for this dataset should be considered in light of several factors. Firstly, the annotated statements only cover the period from 2019 to 2023, which may not capture evolving practices, changes in corporate reporting standards, or emerging risks (due e.g. to conflicts, natural disasters, or pandemics). Over time, government analysts' interpretation of the Act may also evolve along with their expectations of adequate disclosures, resulting in future statements being evaluated differently. Additionally, it is anticipated that the Australian government will publish improved guidance materials, helping companies better understand their disclosure obligations. As companies become more familiar with these requirements, the quality and consistency of their statements may improve. Finally, while the the requirements set by the Australian Modern Slavery Act closely align with many other existing legislation such as the UK Modern Slavery Act (UK Government, 2015), the California Transparency in Supply Chains Act (Rao, 2019), or the Canadian Fighting Against Forced Labour and Child Labour in Supply Chains Act (Canadian Government, 2023), there are slight differences which could impact the generalizability of models trained on our dataset.

## D.5 CONSIDERATIONS FOR USING THE DATA

**Intended use.** The dataset is intended for researchers and developers to train and evaluate machine learning models that extract relevant information from corporate modern slavery statements. It may also be used for extracting specific details such as signature dates, the type of governing body approving a statement, and the location of relevant infographics or tables.

**Social impact of the dataset.** By improving the accuracy and efficiency of identifying mandated disclosures, this dataset can contribute to greater corporate transparency and accountability, helping to combat modern slavery practices. Additionally, the dataset supports the broader goal of fostering responsible business practices and ethical supply chains, potentially leading to better protection of human rights and improved working conditions worldwide.

**Known biases.** The dataset has several known biases that should be acknowledged. First, even if there are other legislation that have been enforced for longer, this dataset only includes statements from entities covered by the Australian Modern Slavery Act, limiting its geographic and regulatory scope. Second, while it allows for voluntary reporting, the Act primarily targets large organizations. In consequence, most statements are published by large companies with annual revenues exceeding AU\$100 million. This introduces a bias towards sectors that dominate the Australian economy, such as natural resource extraction. Companies operating in highly regulated industries or those already subject to modern slavery legislation are also likely to provide more comprehensive reports in their first reporting period. In contrast, companies newly required to examine their supply chains and assess modern slavery risks may have less to report initially. Lastly, while the annotation specifications were meticulously designed to minimize subjectivity and adhere closely to the Act and guidance materials, the process still involves human judgment from annotators and analysts, which can introduce variability and bias.

**Limitations.** See Section 6 of the paper and Appendix F.

# E ANNOTATION PROCESS

## E.1 ANNOTATION GUIDELINES

---

**Text extraction and labeling workflow for C2 ("supply chains")**

Does the reporting entity describe its supply chains?

→ **Yes**, the statement describes the supply chains of the reporting entity:

- Copy-paste the text passages from the statement that justify that the reporting entity described its supply chains.

- If any relevant information comes in other formats than text, fill in the required information in the "Visual Element" fields: note the page where the information is found, and extract any relevant text (if possible).

→ **No**, the statement does not describe the reporting entity's supply chains:

- Copy-paste the exact text passages from the statement that justifies that the entity does not meet this criterion, OR

- If no information is found about this criterion, set the "No relevant information found" flag.

→ **Unclear**, in any other case:

- Select this label if the information found is unclear or there are other concerns.

- If you decide to select this label, you have to provide an explanation that justifies your decision as supporting text.

---

Figure 5: Workflow used for supporting text extraction and labeling for C2 ("supply chains").

We provide a copy of our annotation specifications document as supplementary material with this appendix. This document contains guidelines for hired workers to annotate statements according to our eleven questions on the Mandatory Criteria of the Act (listed in Section 2 of the paper). It includes detailed instructions on handling non-contiguous text, intricate formatting, sections with embedded text, headings, and dates. Following the general guidelines, we outline the eleven questions related to the Mandatory Criteria and how to address them. Each of the first six Mandatory Criteria is associated with a question; for example, for C1, we ask which entities covered by the statement are the "reporting entities". Exceptions were made for C2 and C4, as these criteria encompass multiple disclosure topics. Specifically, C2 is divided into three questions covering the descriptions of operations, governance structure, and supply chains, while C4 is split into two questions addressing the descriptions of remediation actions and risk mitigation actions. We did not include a direct question for C7 ("any other relevant information") due to its subjective nature. Instead, we request that any relevant information be extracted in response to the appropriate questions. We note that this criterion was also omitted in the Australian Government's annual analysis report (Australian Government, 2022). Besides, all instructions and questions are accompanied by numerous examples based on real statements.

For each question, the annotators are presented with a labeling workflow; an example is given in Figure 5 for C2 ("supply chains"). Recognizing that ambiguous, vague, and distracting sentences can sometimes be challenging to assess, we provide annotators with the option to answer a question with an "unclear" label. This helped us understand confusing cases and improve our instructions during early iterations on the guidelines. Ultimately, only a very limited number of "unclear" labels were obtained in the final annotated dataset, and these are not considered in our experiments.

In Figure 6 we present a highly simplified fictitious example of an annotated statement for the proposed tasks and labels, offering readers a clearer high-level overview. However, we strongly

encourage readers to consult the full annotation specification document attached to this paper, which contains real examples and highlights the complexity of the task.

## E.2   CONTRACTING AND QUALITY ASSURANCE DETAILS

We contacted and evaluated several companies offering professional annotation services, and short-listed two of them for a potential contract. A crucial requirement for our project was that the chosen company must agree to clauses on legal, ethical, and best practice obligations (covering procurement practices, subcontracting and sub-funding, modern slavery, and diversity), ensuring fair compensation and treatment for the annotators. Another key element was for the company to ensure that it has a solid quality assurance process in place and a good annotation platform for PDF files. Following the initial assessment, quotation, and agreement on collaboration terms, we chose one of the two withheld companies. Based on the analysis of the selected company's payment structure and operational details, we strongly believe that the participants were fairly compensated. The annotation team consists of management and senior annotators combined with hired annotators that were primarily university students and graduates. These annotators were hired following thorough background checks and interviews. The payment structure for the work allowed us to estimate that the company was paid at least USD$18 per hour of annotation. Even after deducting the company's costs, it is estimated that the annotators receive a fair wage.

The annotation specifications were created by a multidisciplinary team, including experts in machine learning, business, human rights, modern slavery, and in the annotation process. Once the initial version of the specifications was finalized, it was tested multiple times by our team until no general patterns of errors were identified. The specifications document was then sent to the professional annotation company which tested it independently and validated it on a small sample of annotations. Afterward, it was sent back to the expert team for validation. If significant patterns of errors were identified, the annotation specification was reviewed and updated, and the entire process was repeated. This occurred with questions related to Approval, Signature, and Criterion 1, where we had to re-annotate approximately 1000 statements.

The internal quality assurance process of the contracted company includes selective recruitment, comprehensive training for annotators, and dedicated project managers. At various stages of the annotation process, random sampling is conducted to verify the reliability and consistency of annotations. Annotators are also given unseen documents from a testing set at different intervals to check if they remain consistent. Additionally, in cases of double-annotated statements, annotators work independently without seeing each other's work. If the Inter-Annotator Agreement (IAA) is below a specified threshold for those statement, a third annotator steps in to correct the answers. Combined with regular communication and feedback on weekly samples, this process ensures a level of confidence in the quality of the annotated dataset.

## E.3   DECISIONS AND OBSERVATIONS

During the creation of the annotation specifications, we documented essential decisions and observations that may influence future studies and experiments. Key points that are considered limitations are discussed in Appendix F; here, we discuss other noteworthy points.

**Annotators are instructed to never extract section titles or headers.** This means that if the section title itself provides supporting evidence or context, it will still not be extracted. This is sometimes problematic: for example, Criterion 1 ("reporting entity") evidence is often presented in a section titled "Reporting Entity". In those cases, annotators extract sentences from that section containing company names, but that often do not explicitly identify those companies as "reporting". This may lead to confusion under the *no-context* experiment setup. Ignoring section titles is however necessary, as they often do not accurately reflect the content of the paragraphs they precede. For example, a section titled "Supply Chains" might primarily discuss operations or risks, which could mislead annotators if they rely on the heading rather than thoroughly reading the paragraphs. This also helps avoid the concatenation of section titles with sentences when copy-pasting text from the PDF files, which would be a challenging problem to solve.

**Statements are expected to be self-contained.** Only text within the statements can be considered: annotators are instructed to NEVER follow URLs or other documents cited in the statements. In

**Fictitious Modern Slavery Statement: TyraGain Technologies Pty Ltd**

For the reporting period 1 January 2024 to 31 December 2024

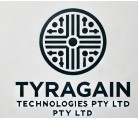

**Introduction and Reporting Entity**

This Modern Slavery Statement (Statement) is submitted by TyraGain Technologies Pty Ltd (TyraGain), in compliance with the Modern Slavery Act 2018 (Cth) (Act). TyraGain is an Australian-based provider of cutting-edge technology solutions, specializing in artificial intelligence (AI) and data analytics. Our commitment to ethical practices is central to our mission of leveraging technology for good, and this includes a strong stance against modern slavery in all forms.

**Organizational Structure and Operations**

TyraGain's headquarters is in Sydney. It has offices in Melbourne and Perth. The company employs software specialists that include developers, data scientists, and cybersecurity experts. TyraGain provides services to a global client base, ranging from government agencies to Fortune 500 companies, particularly in the areas of AI-driven analytics and cloud-based solutions.

**Supply Chain Overview**

TyraGain's supply chain includes a wide range of suppliers, from technology hardware manufacturers to software vendors and professional service providers. While most of our suppliers are based in Australia, we also source hardware components from China, India, and Southeast Asia. We recognize that some of these regions may pose risks of modern slavery, particularly in manufacturing.

**Modern Slavery Risks:** TyraGain acknowledges the potential risks of modern slavery within its global supply chain. Specific areas of concern include:

- Electronics manufacturing, where forced labor may be present in the production of hardware components.
- Outsourced IT and support services, particularly in regions with less stringent labor laws.
- Third-party contractors providing maintenance and logistics services.

In line with our commitment to ethical practices, TyraGain has implemented several initiatives to mitigate the risks of modern slavery:

**Supplier Vetting and Onboarding**. All new suppliers undergo a rigorous vetting process that includes checks for compliance with modern slavery laws. This process ensures no supplier is overlooked. They must also agree to the terms in our Supplier Code of Conduct as a condition of doing business with Tyragain, which covers modern slavery topics and reporting requirements.

**Regular Audits and Monitoring**. We conduct annual audits of high-risk suppliers, focusing on those located in regions with known labor issues. These audits are performed by Supplycheck, an independent third partiy to ensure objectivity and thoroughness.

**Whistleblower Mechanism**. We have established a confidential whistleblower mechanism that allows employees and suppliers to report concerns about unethical practices, including modern slavery. We are committed to investigating reports promptly and taking appropriate action. We use key performance indicators (KPIs) to measure the effectiveness of this mechanism: this year, zero incidents of forced labor were reported or suspected.

**Training Programs**. TyraGain has developed training programs to educate employees and suppliers on the risks of modern slavery. These programs emphasize the importance of vigilance.

**Effectiveness of Actions and Future Steps**

Throughout 2024, TyraGain has made significant strides in addressing modern slavery risks. However, we remain committed to continuous improvement. In 2025, we plan to enhance our supplier engagement by introducing more frequent audits and expanding our training programs to include more in-depth case studies on modern slavery.

**Approval**

This Statement was approved by the Board of Directors of TyraGain Technologies Pty Ltd on 30 June 2025. It was signed by our Chief Executive Officer, John Doe.

John Doe
Chief Executive Officer, TyraGain Technologies Pty Ltd
30 June 2025

| Approval | C1 reporting entity | C2 structure | C2 operations | C2 supply chains | C3 risk description | C4 risk mitigation | C4 remediation | C5 assessment of effectiveness | C6 consultation | Signature |
|---|---|---|---|---|---|---|---|---|---|---|

Figure 6: Example of a fictitious modern slavery statement with sentence-level annotations. Sentences are highlighted based on their relevance to different criteria, as determined by annotators. Sentences that are not highlighted are considered irrelevant for all criteria. In our actual dataset, the statements are typically much longer and often contain sentences that are relevant to multiple criteria simultaneously.

consequence, annotators also cannot always ascertain whether the right "governing bodies" are providing approval, whether the right individuals are providing signatures, or whether all owned or controlled entities are included in the statement due to a lack of external context.

**Statements are expected to be understandable by a layperson.** While we provided a glossary of key terms in the annotation specifications, we do not ask annotators to search for information on specific business or legal terms, on existing legislation or legal frameworks, or on risk assessment tools. We expect the statement issuers to use clear terminology and avoid terminology that may be misleading.

**Statement types indicated in the Modern Slavery Register are not reliable**. This metadata is likely provided by the statement issuer, but may be incorrect. Specifically: "joint" statements can sometimes be presented by only one reporting entity, and "normal" statements can be issued by a parent entity and cover many of its owned/controlled entities.

**The "principal governing body" of an entity is often implicitly defined.** Identifying whether a statement is correctly approved is therefore challenging when dealing with multinational corporations with complex structures, or in the case of trusts. Also, in joint statements, seemingly independent entities can have the same board members, and this rarely mentioned in statements.

**Only the most relevant mentions of "reporting entities" are extracted.** This is specific to the question related to Mandatory Criterion 1: we decided to extract only the most obvious declarations. This is done to avoid having to exhaustively extract each sentence where an entity is named, as this approach does not scale well to large statements.

**Arrangements with suppliers do not describe operations.** This is in contradiction with the Australian government's guidance material (see Table 2 of Australian Government, Attorney-General's Department, Modern Slavery Business Engagement Unit, 2023). Specifically, we consider that "explaining in general terms the type of arrangements the entity has with its suppliers and the way these are structured" is vague, hard to convey to annotators, and relates more to descriptions of suppliers or supply chains. We found that annotation quality improved following this decision.

**The "structure" of an entity is a vague concept.** A reporting entity may for example describe its management and governance structure (e.g. naming executives or members of its board of directors), while another might focus more on its organizational structure (e.g. naming parent companies, subsidiaries, and affiliates). The latter is usually understood to be more relevant, but the Australian government also considers, for example, Australian Business Number (ABN) and registered office location to be relevant information (Australian Government, Attorney-General's Department, Modern Slavery Business Engagement Unit, 2023) while making no mention of the importance of capital structure, reporting structure, or taxation structure descriptions. Classifying information on shareholders is also difficult, as it may sometimes be relevant when few shareholders have significant control over the reporting entity. Lastly, we note that descriptions of "brick-and-mortar" locations (e.g. facilities, stores) are often provided as descriptions of structure by companies, but this is instead considered relevant for operations.

**The number of workers is considered structure information.** According to the Australian government's guidance material (Australian Government, Attorney-General's Department, Modern Slavery Business Engagement Unit, 2023), this information may be relevant for both structure and operations. However, for simplicity and clarity, we considered it only relevant for structure in our guidelines to annotators.

**Descriptions of customers are not relevant for supply chains.** In reality, customers can be considered as part of the "downstream" supply chain of an entity, but we do not consider this information relevant in our guidelines. The Australian government's guidance material (Australian Government, Attorney-General's Department, Modern Slavery Business Engagement Unit, 2023) also mentions that entities are not required to report this information. However, the distribution of products or services to customers is considered a relevant activity (or operation).

**Risks and actions may not always apply to owned or controlled entities.** Specifically, Mandatory Criteria 3, 4, and 5 require entities to provide information about risks and actions that apply to "the reporting entity and any entities it owns or controls." However, based on consultations with the Australian Attorney General's Department and annotation experts, we decided that if a description of risks or actions only seem to apply to the reporting entity, this information is still considered relevant.

We initially decided to have a separate data field to flag information that would also apply to owned and controlled entities, but we determined during testing that it was rarely used; it was eventually removed from labeling workflows.

**Owned or controlled entities might not always be consulted.** Due to ambiguities and the lack of external context, it is difficult to determine whether the list of owned and controlled entities perfectly overlaps with the list of "consulted" entities. Although Mandatory Criterion 6 requires reporting entities to consult with all entities they own or control, there are also various reasons why they might not be able to do so. Some of those entities may, for example, be dormant, inactive, or non-trading. Furthermore, only consultation "on the preparation of the statement" is considered relevant for this criterion, but reporting entities rarely describe their actual consultation process.

**Statement signatures are sometimes difficult to interpret.** For example, large statements often contain a "message from the CEO" with general comments on the importance of the statement or on the achievements of their company. These message are often signed, but it is unclear if that signature applies to the whole statement, or just to that message. Documents may also occasionally lack the actual image of a signature, or may only include a blank space or a box where a signature is supposed to be. Such cases are still considered valid evidence, as the image of the signature is not necessary, but the intent to sign is acknowledged.

## F LIMITATIONS

We concluded the paper by highlighting some of the key limitations of our dataset (Section 6). Among these, the most significant challenge is the subjective and noisy nature of the relevant sentence annotation process. Although our guidelines for annotators were designed to minimize subjectivity and maximize consistency, the Inter-Annotator Agreement (IAA), as shown in Table 1 of the paper, varies significantly across different questions. Based on qualitative analyses of the annotated data, we believe that the IAA is not an ideal measure of annotation quality. Good IAA scores were observed in some statements where a significant amount of relevant information was missed by annotators and where obviously relevant information was correctly extracted. Initially, we set high thresholds for expected IAA scores with the annotators, but we later encouraged lower IAA scores for statements deemed more difficult to annotate. This approach aimed to promote the extraction of more potentially relevant text. Ultimately, we believe that modeling approaches capable of handling noisy labels and leveraging annotator disagreements as an additional supervision signal may lead to more effective solutions for sentence relevance classification.

A somewhat subjective annotation process can also introduce bias in the labeling of disclosures, potentially leading to unfair assessments of whether certain companies (or those operating in specific industrial sectors) meet the requirements of the Act. This bias might result from individual annotators' interpretations of the guidelines or their preconceived notions about particular industries. To mitigate this risk, we consulted with experts in the design of our annotation guidelines, aiming to minimize any disadvantage to specific businesses, and relied on the professionalism of the annotation company and their internal QA process to vouch for their work. Furthermore, for transparency and to allow for external review and improvement, we make both the annotations and the guidelines publicly available.

The extraction of text from PDFs poses other significant challenges. Beyond the difficulty of correctly extracting text from embedded figures and tables, matching any sentence annotated by a human to the automatically extracted text from the PDF is also complex. This difficulty arises due to text fragmentation, OCR errors, non-ASCII character mismatches, and out-of-order parsing. In practice, we found that using ABBYY FineReader, a commercial software with an OCR engine, reduced the match rate for annotated sentences compared to using PyMuPDF (fitz), which lacks an OCR engine, even when employing a Levenshtein sentence matching approach. Revisiting the text extraction and matching methodology, potentially replacing regular expressions with a more advanced method for determining sentence boundaries and matching them, would likely enhance the reliability of evaluations for relevant text classification models.

As for the challenge of differentiating past and future information in our dataset, one potential solution is to introduce temporal labels, where markers indicating whether the information pertains to past actions, ongoing activities, or future plans would be added to annotations. Language models could be

employed to automatically infer these markers from the text, reducing the re-annotation burden and providing scalability.

Experiments for single-sentence classification with API-based language models with large context windows can be wasteful due to the high number of model requests required, significantly increasing costs. Future works might want to explore the simultaneous classification of multiple sentences at once, such as paragraph-by-paragraph, to reduce the number of model requests. This approach would however necessitate more substantial prompt engineering and output parsing efforts. Additionally, a hierarchical context processing approach, which involves structuring the input to provide broader context on the statement before drilling down to specific sentence-level details, could be worth investigating for both zero-shot and supervised learning settings.

## G  Implementation and Experimentation Details

Details on the models we selected as baselines for our experiments are presented in Table 5. In addition to the experimentation details presented in Section 5 of the paper (Benchmark Experiments), we report that the models are fine-tuned with a cross-entropy loss using the Adam optimizer and without a learning rate scheduler. Each model is trained for 24 hours on a A100L GPU, with the exception of Llama2 (7B), which is trained for 48 hours to allow the model more time to converge. In the case of Llama2 (7B), a batch size of 32 is simulated using gradient accumulation, where the real batch size is set to 2 and the gradient is accumulated over 16 steps. All the fine-tuning is conducted in 16-bit mixed precision mode. For DistilBERT and BERT, we attach a classification head directly to the CLS token positioned at the beginning of the target sentence for both the *no-context* and *with-context* setups. For Llama2 (7B) and Llama3.2 (3B), we use the last token as is typically done with other causal models. In the zero-shot case, we used the default temperature of 0.6 for Llama3.2 (3B); in the GPT model cases, the default temperature means that "the model will use log probability to automatically increase the temperature until certain thresholds are hit" (from OpenAI API reference page).

For training data preparation, the pre-extracted statement text is split into sentences with various amounts of context at training time. These sentences are then shuffled and assembled into minibatches using a fixed-size sentence buffer (containing up to 8192 sentences). We assign a positive relevance label to any extracted sentence that matches a sentence tagged by an annotator as being relevant, and assign a negative relevance label otherwise. The matching of extracted and tagged sentences is done following text cleanups using regular expressions, and by considering perfect matches, partial matches, and noisy matches based on the Levenshtein distance.

Table 5: Baseline model details. For BERT and DistilBERT, full model weights are fine-tuned, and for Llama2 (7B) and Llama3.2 (3B), we use the LoRA approach (Hu et al., 2021), resulting in a smaller number of trainable parameters. The [*] suffix denotes zero-shot models.

| Model name | URL | Total params | Trainable params |
|---|---|---|---|
| DistilBERT | https://huggingface.co/distilbert/distilbert-base-uncased | 66.8M | 66.8M |
| BERT | https://huggingface.co/google-bert/bert-base-uncased | 109M | 109M |
| Llama2 (7B) | https://huggingface.co/NousResearch/Llama-2-7b-hf | 6.6B | 4.2M |
| Llama3.2 (3B) | https://huggingface.co/meta-llama/Llama-3.2-3B | 3.2 B | 2.3 M |
| GPT3.5 Turbo[*] | https://platform.openai.com/docs/models/gpt-3-5-turbo | ? | - |
| GPT4o[*] | https://platform.openai.com/docs/models/gpt-4o | ? | - |
| Llama3.2 (3B)[*] | https://huggingface.co/meta-llama/Llama-3.2-3B-Instruct | 3.2 B | - |

## H  Prompt Design and Examples

To develop the final version of the prompt, we began with preliminary tests using a small set of five PDFs. These initial documents were excluded from the final analysis to avoid any potential contamination. The prompt development process incorporated a variety of resources, including raw PDFs, extracted text, a complete annotation specification document, a summary cheat sheet,

and annotated examples. This iterative approach involved refining the prompts based on manual evaluations conducted by a domain expert in modern slavery reporting, while also accounting for constraints such as token limits and computational costs. Version 1 focused on classifying sentences using raw PDFs and relevant text from the annotation specification. Version 2 incorporated both the PDFs and the full annotation specification document. Version 3 experimented with subsets of the annotation specification, cheat sheet, and examples. Version 4 shifted to using extracted text instead of raw PDFs. Finally, Version 5 involved optimizing prompt text using ChatGPT, aiming to generate outputs that included labels and justifications, supported by examples from the annotation specification. Each iteration was refined to achieve a balance between accuracy and efficiency, following best practices on how to formulate intents, how to provide domain definitions, and how to constrain desired outputs.

We present in Figures 7 and 8 the exact prompt templates we used for the *no-context* and *with-context* setups for zero-shot model experiments. Note that the `TARGET_SENTENCE` and `SENTENCE_IN_CONTEXT` placeholders are respectively substituted with the target sentence to classify and the same sentence with surrounding context in actual model prompts. For an example of a target sentence that would be classified along with its context, see Figure 9.

Prompt template (C2, "supply chains", *no-context*)

You are an analyst that inspects modern slavery declarations made by Australian reporting entities. You are specialized in the analysis of statements made with respect to the Australian Modern Slavery Act of 2018, and not of any other legislation.

You are currently looking for sentences in statements that describe the SUPPLY CHAINS of an entity, where supply chains refer to the sequences of processes involved in the procurement of products and services (including labour) that contribute to the reporting entity's own products and services. The description of a supply chain can be related, for example, to 1) the products that are provided by suppliers; 2) the services provided by suppliers, or 3) the location, category, contractual arrangement, or other attributes that describe the suppliers. Any sentence that contains these kinds of information is considered relevant. Descriptions that apply to indirect suppliers (i.e. suppliers-of-suppliers) are considered relevant. Descriptions of the supply chains of entities owned or controlled by the reporting entity making the statement are also considered relevant. However, descriptions of 'downstream' supply chains, i.e. of how customers and clients of the reporting entity use its products or services, are NOT considered relevant. Finally, sentences that describe how the reporting entity lacks information on some of its supply chain, or how some of its supply chains are still unmapped or unidentified, are also considered relevant.

Given the above definitions of what constitutes a relevant sentence, you will need to determine if a target sentence is relevant or not. You must avoid labeling sentences with only vague descriptions or corporate talk (and no actual information) as relevant. The answer you provide regarding whether the sentence is relevant or not can only be 'YES' or 'NO', and nothing else.

The target sentence to classify is the following:
————
TARGET_SENTENCE
————
Is the target sentence relevant? (YES/NO)

Figure 7: Prompt template used for zero-shot model experiments under the *no-context* setup.

> **Prompt template (C2, "supply chains", *with-context*)**
>
> You are an analyst that inspects modern slavery declarations made by Australian reporting entities. You are specialized in the analysis of statements made with respect to the Australian Modern Slavery Act of 2018, and not of any other legislation.
>
> You are currently looking for sentences in statements that describe the SUPPLY CHAINS of an entity, where supply chains refer to the sequences of processes involved in the procurement of products and services (including labour) that contribute to the reporting entity's own products and services. The description of a supply chain can be related, for example, to 1) the products that are provided by suppliers; 2) the services provided by suppliers, or 3) the location, category, contractual arrangement, or other attributes that describe the suppliers. Any sentence that contains these kinds of information is considered relevant. Descriptions that apply to indirect suppliers (i.e. suppliers-of-suppliers) are considered relevant. Descriptions of the supply chains of entities owned or controlled by the reporting entity making the statement are also considered relevant. However, descriptions of 'downstream' supply chains, i.e. of how customers and clients of the reporting entity use its products or services, are NOT considered relevant. Finally, sentences that describe how the reporting entity lacks information on some of its supply chain, or how some of its supply chains are still unmapped or unidentified, are also considered relevant.
>
> Given the above definitions of what constitutes a relevant sentence, you will need to determine if a target sentence is relevant or not inside a larger block of text. The target sentence will first be provided by itself so you can know which sentence we want to classify. It will then be provided again as part of the larger block of text it originally came from (extracted from a PDF file) so you can analyze it with more context. While some of the surrounding sentences may be relevant according to the earlier definitions, we are only interested in classifying the target sentence according to the relevance of its own content. You must avoid labeling sentences with only vague descriptions or corporate talk (and no actual information) as relevant.
>
> The answer you provide regarding whether the sentence is relevant or not can only be 'YES' or 'NO', and nothing else.
>
> The target sentence to classify is the following:
> ————
> `TARGET_SENTENCE`
> ————
> The same target sentence inside its original block of text:
> ————
> `SENTENCE_IN_CONTEXT`
> ————
> Is the target sentence relevant? (YES/NO)

Figure 8: Prompt template used for zero-shot model experiments under the *with-context* setup.

---

**Target sentence example**

The compliance with these communicated expectations is ensured by regular unannounced audits of all suppliers in this region.

---

**Target sentence example with 100-word context**

[...] we have established clear and stringent expectations for all our suppliers in Southeast Asia regarding labor practices and ethical standards. These expectations are communicated through detailed supplier agreements and comprehensive training programs. Additionally, we collaborate closely with local communities and stakeholders to promote awareness and understanding of ethical labor practices. **The compliance with these communicated expectations is ensured by regular unannounced audits of all suppliers in this region.** Furthermore, our commitment to transparency and accountability extends beyond audits, as we engage with independent third-party auditors to validate our findings and ensure the integrity of our supply chains. Any detected non-compliance triggers immediate corrective actions and follow-up reviews, demonstrating our dedication to resolving issues swiftly and [...]

---

Figure 9: Example of a fictitious sentence to be classified as relevant or irrelevant, with and without context. The amount of context here (roughly 100 words) is the same one used in our experiments. For the question related to C5 (assessing the effectiveness of actions), classifying this sentence is difficult when context is not provided, as it is unclear whose and what expectations were communicated, and whose suppliers are audited. With context, it is clear that the sentence contains relevant information mandated by Mandatory Criteria 5 of the Act.

# I    ADDITIONAL RESULTS

## I.1    F1 EVOLUTION OVER THE EPOCHS

Figure 10 illustrates the evolution of fine-tuned model performance, measured by validation Macro F1, during training in the *No context* setup. While BERT and DistilBERT achieve strong performance from the first epoch, Llama2 (7B) requires several epochs to reach comparable levels, with Llama3.2 (3B) falling in between, needing only a few epochs to perform well. We hypothesize a trend where larger model sizes require more epochs to achieve optimal performance. Furthermore, we observe that Llama2 (7B) could benefit from extended fine-tuning, as its Macro F1 curve has not plateaued even after 48 hours of training. Additionally, we observe that Llama2 (7B) may benefit from extended fine-tuning, as the macro F1 curve has not plateaued even after 48 hours of training.

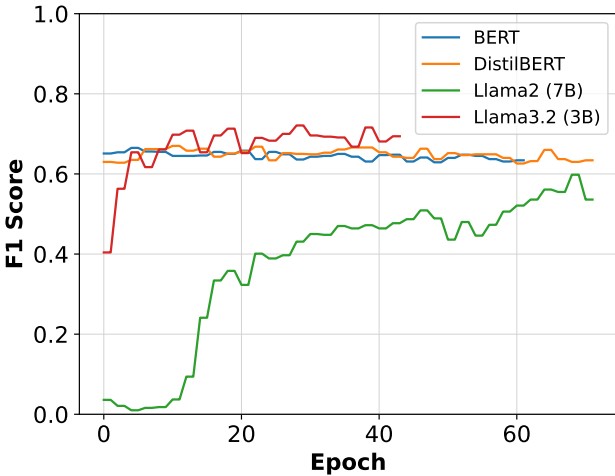

Figure 10: Macro F1 score over the epochs for the fine-tuned models in the all-label case.

# J    COMPARISON OF MODERN SLAVERY REPORTING CRITERIA AND METRICS

Since the enactment of the Australian Modern Slavery Act, various existing laws, such as the UK Modern Slavery Act (UK Government, 2015), have been strengthened with more robust reporting requirements, and new legislation has been introduced, such as the Canadian Fighting Against Forced Labour and Child Labour in Supply Chains Act of 2023 (Canadian Government, 2023). These laws share overlapping reporting criteria, whether recommended or mandated. To demonstrate how our dataset and annotations could be used to build predictive models that generalize to other legal frameworks, Table 6 compares the questions in our annotation specifications with the reporting obligations set by the Australian MSA, the UK MSA, and the Canadian legislation. This table also includes metrics used by civil society organizations (specifically, those proposed by Walk Free, 2022b) to assess modern slavery statements.

Table 6 highlights areas of overlap and divergence based on text color:

- Green sections represent requirements where our existing annotations can be used to train algorithms without any or with minimal modifications.
- Orange sections indicate areas that may necessitate the use of a subset of our annotations, additional data mining, or potential adjustments and expansions to our current annotation set.
- Red sections highlight where there is no overlap; here, our annotations do not apply and would require complete re-annotation to accommodate these aspects.

This comparative analysis underscores the adaptability of our annotation framework and identifies specific areas for enhancement to achieve broader applicability across different legislative contexts, with the potential to also support civil society efforts in their assessments.

Table 6: Comparison of Modern Slavery Reporting Criteria and Metrics

| AIMS.au Dataset Annotation Specification Questions | Australian Modern Slavery Act Mandatory Reporting Criteria | UK Modern Slavery Act Reporting Suggestions | Canadian Fighting Against Forced Labour and Child Labour in Supply Chains Act Reporting Obligations | The Walk Free's "Beyond Compliance" Study Metrics |
|---|---|---|---|---|
| Question: Is the statement approved by the entity's principal governing body? | Ensure that the statement is approved by the board. | Approval from the board of directors (or equivalent management body) | Approval by the organization's governing body. | MSA Statement Approval |
| Question: Is the statement signed by a responsible member of the reporting entity? | The statement is signed by a responsible member of the organization. | Signature from a director (or equivalent) or designated member | Signature of one or more members of the governing body of each entity that approved the report. | MSA Statement Signed |
| Question: Does the statement clearly identify which entities covered by the statement are the relevant reporting entities? | Mandatory Criterion 1: The statement clearly identifies the Reporting Entity. | N/A | N/A | N/A |
| Question: Does the reporting entity describe its structure? Question: Does the reporting entity describe its operations? Question: Does the reporting entity describe its supply chains? | Mandatory Criterion 2: Describe the reporting entity's structure, operations, and supply chains. | The organisation's structure, business and supply chains. | Description of the organisation's structure, activities and supply chains. | MSA Organizational Structure and Operations MSA Supply Chain Disclosure |
| Question: Does the reporting entity describe its modern slavery risks? | Mandatory Criterion 3: Describe the risks of modern slavery practices in the operations and supply chains of the reporting entity and any entities the reporting entity owns or controls. | Risk assessment and management. | Description of the parts of its business and supply chains that carry a risk of forced labour or child labour being used and the steps it has taken to assess and manage that risk. | MSA Identification of Risks |
| Question: Does the reporting entity describe the actions applied to identify, assess, and mitigate the modern slavery risks it identified? | Mandatory Criterion 4: Describe the actions taken by the reporting entity and any entities it owns or controls to assess and address these risks, including due diligence and remediation processes. | Description of the organisation's policies in relation to slavery and human trafficking. Description of the organisation's due diligence processes in relation to slavery and human trafficking in its business and supply chains. Description of the parts of the organisation's business and supply chains where there is a risk of slavery and human trafficking taking place, and the steps it has taken to assess and manage that risk. The training and capacity building about slavery and human trafficking available to its staff. | Description of the organisation's policies and due diligence processes in relation to forced labour and child labour. Description of the parts of organisation's activities and supply chains that carry a risk of forced labour or child labour being used and the steps it has taken to assess and manage that risk. The training provided to employees on forced labour and child labour. | MSA Policy MSA Risk Assessment MSA Risk Management MSA Whistleblowing Mechanism MSA Training |
| Question: Does the reporting entity describe remediation actions for modern slavery cases? | Mandatory Criterion 4: Describe the actions taken by the reporting entity and any entities it owns or controls to assess and address these risks, including due diligence and remediation processes. | The organisation should paint a detailed picture of all the steps it has taken to address and remedy modern slavery, and the effectiveness of all such steps. | Description of any measures taken to remediate any forced labour or child labour. | MSA Incidents Remediation |
| Question: Does the reporting entity describe how it assesses the effectiveness of its actions? | Mandatory Criterion 5: Describe how the reporting entity assesses the effectiveness of these actions. | Description of the organisation's effectiveness in ensuring that slavery and human trafficking is not taking place in its business or supply chains, measured against such performance indicators as it considers appropriate. The organisation should paint a detailed picture of all the steps it has taken to address and remedy modern slavery, and the effectiveness of all such steps. | Description of how the entity assesses its effectiveness in ensuring that forced labour and child labour are not being used in its business and supply chains. | MSA Performance Indicators |
| Question: Does the reporting entity describe how it consulted on its statement with any entities it owns or controls? | Mandatory Criterion 6: Describe the process of consultation with any entities the reporting entity owns or controls. | N/A | N/A | N/A |
| N/A | Mandatory Criterion 7: Provide any other relevant information. | N/A | Any measures taken to remediate the loss of income to the most vulnerable families that results from any measure taken to eliminate the use of forced labour or child labour in its activities and supply chains. | MSA Impact on Company Behaviour MSA Business Performance Indicators MSA Historic Record |