# Project AIMS

# (AI against Modern Slavery)

## Annotation Specifications

Shared with ICRL Camera-Ready of the Paper:

"AIMS.AU: A Dataset for Analyzing Modern Slavery Countermeasures in Corporate Statements"

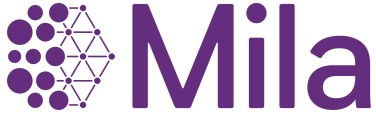 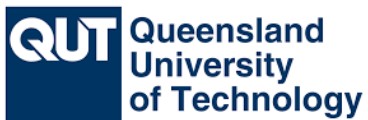

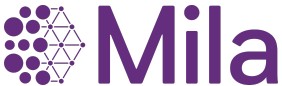
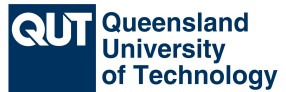

# Table of contents

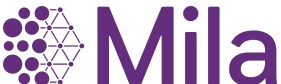
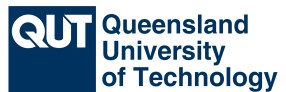

# Glossary

**Modern slavery:** refers to the coercion, deception, or exploitation involved in trafficking in persons, forced labor, debt bondage, or servitude. Specifically, slavery entails the exercise of powers akin to the right of ownership, where an individual significantly deprives another person of their individual liberty, often controlling their labor. This definition also encompasses the offense of trafficking in persons, which involves the recruitment, transportation, or transfer of individuals using coercion or deception for the purpose of exploitation. Modern slavery situations therefore refer to situations of exploitation in which a person cannot refuse to participate and cannot leave because of threats, violence, coercion, deception, or abuse of power.

**The Modern Slavery Act ("the Act"):** the 2018 Australian law requiring entities to provide statements on modern slavery; available [here](#).

**Mandatory criterion (plural: mandatory criteria):** a requirement set by the Act regarding the contents of the statements made by reporting entities. There are seven criteria in the Act, and these are translated into questions that must be answered by annotators.

**Reporting entity:** an entity required to report under the Act (or "subject to the law"), or that is voluntarily reporting, and which carries on business in Australia.

**Question:** a question to which we are seeking an answer (or "label"), which is supported by evidence (or "supporting text"), regarding whether a reporting entity has adequately met one or more criteria of the Act. These are described in Section 2.

**Labels:** the possible labeling options for a question.

**Supporting text:** the exact (copy-pasted) text taken from a statement that justifies the label selected by the annotator for a certain question.

**Supplementary buttons and boxes:** interactive fields in the annotation tool allowing annotators to provide additional information regarding the statements when answering a question.

**Principal governing body:** the body or group of members with primary responsibility for the governance of an entity.

**"Normal" statement:** a statement made by a single reporting entity for a given reporting period.

**"Joint" statement:** a collective statement made by multiple reporting entities for a given reporting period.

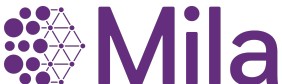
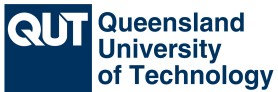

**Responsible member:** a decision-making member of a reporting entity. Where a reporting entity has a principal governing body, the responsible member will be a member of that body.

**Reporting period:** a financial year or another annual accounting period applicable to the reporting entity.

**Due diligence:** refers to an ongoing management process to identify, prevent, mitigate, and account for how an entity addresses actual and potential adverse human rights impacts in their operations and supply chains, including modern slavery.

**Control or ownership of an entity by another entity**: the control of an entity within the meaning of the Australian accounting standards.

**Explanation for failure to comply:** an explanation given by a reporting entity in its statement as to why it could not meet its obligations under the Act. Such explanations are rarely provided, but when they are, they must typically be extracted as supporting text for the negative answer to a question.

**Statement identifier (SID):** identifier used to refer to a specific statement stored in the [Australian Modern Slavery Register](). To access the statement from the SID, add the SID number in the following URL format: https://modernslaveryregister.gov.au/statements/SID

# 1. Overall annotation guidelines

Below, we provide guidelines that apply to the annotation process of all questions described in Section 2.

## 1.1. Extracting supporting text passages from the statements

The PDF statements should contain information detailing how the reporting entities meet the obligations and requirements set by the Act.

For each label associated with a question, annotators should include a supporting text passage from the statement where they found the corresponding answer. The supporting text passage should be copied and pasted into the supporting text passage box **without any additional text, quotation marks, or edits (i.e. do NOT introduce or correct typos, spelling mistakes, add extra words, forget words, change the capitalisation of words, etc.).**

The purpose of including supporting text passages is to provide additional context and evidence for the decision-making process and to ensure consistency and transparency in the annotation process.

When selecting a supporting text passage, annotators should choose passages that directly relate to the question and that clearly support their annotation.

## 1.2. Using // when the supporting text passages are not contiguous

For each label, the annotator should collect all the supporting text passages in the associated supplementary text box. If a passage contains several contiguous sentences or paragraphs, the entire passage should be copy-pasted as a single chunk to preserve its internal structure. On the other hand, if relevant sentences or paragraphs are found in different parts of the statement or if they are separated by other elements, they should be extracted independently and separated in the text box by **a space, a double forward slash //, and another space:**

> first text passage // second text passage

Note that color is only used in this document to help see the separator as part of the guideline descriptions and in examples. If a passage is spread over more than one page, and if there are footnotes, page numbers, or page headers, the annotator should copy each page separately, using // to separate different pages.

When dealing with a visual element such as a table or figure, only the **relevant** text should be extracted. Whenever distinct copy-pastes must be performed to avoid capturing irrelevant information, the distinct text passages must be separated using // .

**Example 1.2.a** (Allgas Energy, SID#10736) for the question of Section 2.3 (Reporting entity), showing the use of the // separator:

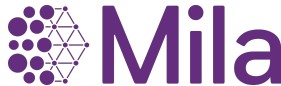
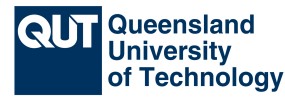

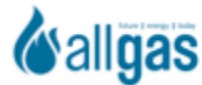

FY22 Modern Slavery Statement
**FY22**

# 1     About GDI

## 1.1     Introduction

GDI (EII) Pty Limited (ACN 154 766 524) (**GDI (EII)**) is an unlisted investment vehicle that wholly owns the Allgas natural gas distribution network (**Allgas** Network or **Allgas**). GDI (EII) Pty Limited has 100% ownership of the trading entity Allgas Energy Pty Limited (ACN 009 656 446) (**Allgas**).

Allgas has three wholly owned subsidiaries which do not trade, Allgas Toowoomba Pty Limited ACN 009 655 645 (which holds network infrastructure assets), Allgas Pipelines Operations 1 Pty Limited ACN 154 784 853 (which holds the investment in Allgas Pipelines Operations Pty Limited), and Allgas Pipelines Operations Pty Limited ACN 076 666 807 (which holds easements).

The Allgas Network extends from Brisbane, south of the river, to the northern tip of New South Wales, with separate networks in Toowoomba and Oakey. It is supplied with natural gas from a variety of upstream producers that connect into APA's East Coast Grid via the Roma – Brisbane Pipeline (**RBP**). The Allgas Network connects to the RBP at Oakey, Toowoomba and Brisbane.

The Allgas Network consists of 3,919km of pipelines and distribution mains, delivering gas to circa.119,087 end users in QLD and NSW. Its major customers are energy retailers including AGL (circa.39% of revenue in FY22) and Origin Energy (circa.40% of revenue in FY22).

The Allgas Network has been subject to light regulation since July 2015 and is regulated under the NGR & NGL. Accordingly, tariffs are negotiated directly with customers.

APA Group (**APA**) (20% equity owner) provides operations and maintenance and corporate services to GDI under a long term services agreement. APA is responsible for the supply chain of GDI and their owned or controlled entities.

In line with the UN Guiding Principles on Business and Human Rights (**UNGPs**), we respect all internationally recognised human rights as set out in the Universal Declaration of Human Rights and the International Labour Organization's Declaration on Fundamental Principles and Rights at Work.

## 1.2     About this statement

In this statement, when we refer to "GDI" it is a reference to GDI (EII) and Allgas, which are both reporting entities under the Modern Slavery Act 2018 (Cth).

This joint Modern Slavery Statement (**Statement**) has been prepared by APA, the operator, to meet the requirements of the Australian Modern Slavery Act 2018 (Cth) for the financial year ending 30 June 2022 (FY22). The reporting entities covered by the Statement are identified in Appendix 1.

This Statement has been prepared by APA's Modern Slavery Team, in conjunction with APA's Modern Slavery Working Group, with the assistance of external subject matter experts. Information about the approach to consultation with reporting entities and owned or controlled entities to develop this statement is set out on page 20.

This Statement was approved by the GDI Board on 28 November 2022 on behalf of all reporting entitles covered by this Statement. The Statement has been signed by the Chair of GDI (page 3).

2

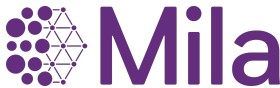
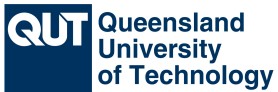

FY22 Modern Slavery Statement
FY22

**allgas**

## 7    Appendix 1 – Reporting Entities

| Name | Description |
| --- | --- |
| GDI (EII) Pty Limited

ACN 154 766 524 | An unlisted investment vehicle that wholly owns the Allgas natural gas distribution network. |
| Allgas Energy Pty Limited

ACN 009 656 446 | The trading entity for the Allgas Network which extends from Brisbane, south of the river, to the northern tip of New South Wales, with separate networks in Toowoomba and Oakey. |
| Allgas Toowoomba Pty Limited

ACN 009 655 645 | Holds network infrastructure assets. |
| Allgas Pipelines Operations 1 Pty Limited

ACN 154 784 853 | Holds the investment in Allgas Pipelines Operations Pty Limited. |
| Allgas Pipelines Operations Pty Limited

ACN 076 666 807 | Holds easements. |

21

- ■ Add the supporting text: *In this statement, when we refer to "GDI" it is a reference to GDI (EII) and Allgas, which are both reporting entities under the Modern Slavery Act 2018 (Cth).*
- ■ Select Supplementary Button: "Visual Element Category" and select "Table".
- ■ Copy-paste the relevant supporting text of the Appendix 1 table in its associated box:
  *GDI (EII) Pty Limited*
  *ACN 154 766 524 //*
  *AAllgas Energy Pty Limited*
  *ACN 009 656 446 //*
  *Allgas Toowoomba Pty Limited*
  *ACN 009 655 645 //*
  *Allgas Pipelines Operations 1 Pty Limited*
  *ACN 154 784 853 //*
  *Allgas Pipelines Operations Pty Limited*
  *ACN 076 666 807*
- ● Select the Supplementary Button: "PDF page number" and add the page number: 21
- ● Note that only the relevant text for the question of Section 2.3 is extracted from the table. The non-relevant text is not extracted.

**Example 1.2.b** (Dalrymple Bay Infrastructure, SID#4480) for the question of Section 2.1 (Approval); in a statement, if the relevant information is presented twice, non-contiguously, then both instances should be extracted:

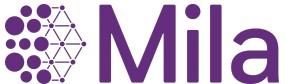
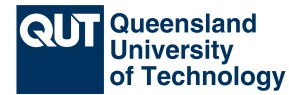

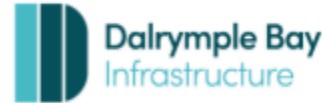

# Modern Slavery Statement

Dalrymple Bay Infrastructure Limited

for the period ended 31 December 2020

Adopted by the Board on 24 June 2021

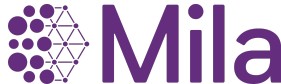

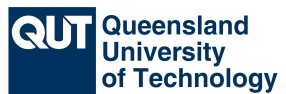

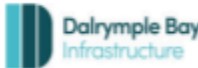

## 8. Board Approval

This statement was approved by the Board of Dalrymple Bay Infrastructure Limited on 24 June 2021.

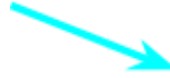

**David Hamill**
Chairman
Dalrymple Bay Infrastructure Limited
24 June 2021

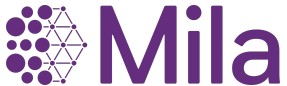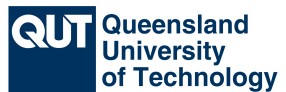

- Add the supporting text: *Adopted by the Board on 24 June 2021* // *This statement was approved by the Board of Dalrymple Bay Infrastructure Limited on 24 June 2021.*
- Select the Supplementary Button: "Date" and add: *24 June 2021* // *24 June 2021*

## 1.3. Formatting passages extracted from the statements

Annotators should aim to extract **full sentences** and not just parts of sentences. Sentences may contain other information that is not relevant to the label, or information that is relevant to other criteria. Even in these cases, full sentences should be extracted. Note: we are only referring to sentences, and not paragraphs.

The relevant information for some criteria may not always be presented in the format of a full sentence, but the annotator should still extract it. For example, for signatures, titles and names must be extracted, although they are not sentences.

**Example 1.3.a** (Reece Group, SID#2503) for the question of Section 2.1 (Approval); this is a case where the entire sentence should be extracted:

# OUR COMMITMENT

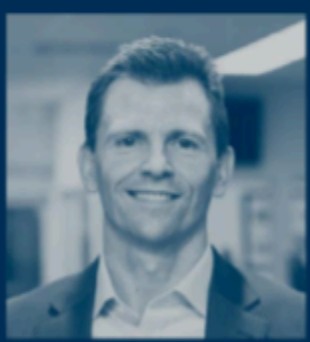

At Reece, our purpose guides our decisions and actions, big and small, and our values ground us. Together, they make up The Reece Way.

We're focused on improving the lives of our customers and our people by striving for greatness every day.

Working together, we can create thriving and sustainable communities. This is only possible if we lead by example and respect and uphold the human rights of our people, customers, and broader network across our supply chain.

We oppose all forms of slavery and forced labour in our operations and the operations of our suppliers.

We have always maintained a rigorous program to ensure supplier compliance with our expectations, and our culture of continuous improvement drives our commitment to building ethical and sustainable supply chains and supplier partnerships.

Our first modern slavery statement shares our journey of identifying and understanding, as well as our initial actions taken to address modern slavery risks. We are committed to working collaboratively across the industry to address this important issue.

This statement has been made in accordance with the Australian Modern Slavery Act (Commonwealth) 2018. This statement is made by Reece Limited on behalf of Reece Australia Pty Ltd, Actrol Pty Ltd, Reece New Zealand Pty Ltd, MORSCO Inc and Todd Pipe Holdings Inc. for the year ended 30 June 2020 and has been approved by the Board of Directors of Reece Limited.

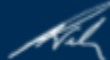

**Peter Wilson**
Reece Group Chief Executive Officer
and Managing Director

29 March 2021

4      Reece Group

- ■ Add the supporting text: *This statement is made by Reece Limited on behalf of Reece Australia Pty Ltd, Actrol Pty Ltd, Reece New Zealand Pty Ltd, MORSCO Inc and Todd Pipe Holdings Inc. for the year ended 30 June 2020 and has been approved by the Board of Directors of Reece Limited.*
- ■ Note: when extracting information for Approval, do **not** include the following sentence which doesn't provide any information pertaining to the approval process: *This statement has been made in accordance with the Australian Modern Slavery Act (Commonwealth) 2018.*

**Example 1.3.b** (Uniqlo, SID#1535) for the question of Section 2.3 (Reporting entity):

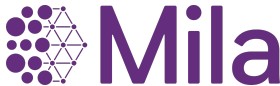
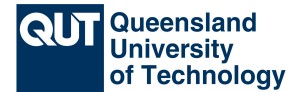

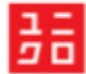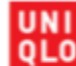

## MODERN SLAVERY STATEMENT OF UNIQLO AUSTRALIA

This statement is in accordance with the requirement that Uniqlo Australia Pty Ltd (**Uniqlo Australia**) report for the fiscal year from 1 September 2019 to 31 August 2020 (**Fiscal Year 2020**) under the *Modern Slavery Act 2018* (Cth) (**Modern Slavery Act**). In this statement, Uniqlo Australia adopts the meaning of modern slavery as defined in the Modern Slavery Act.

Uniqlo Australia is committed to taking action to eliminate modern slavery from our supply chain and business. As a member of the Fast Retailing Group (**Fast Retailing**), our mission is to create truly great clothing with new and unique value, and to enable people all over the world to experience the joy, happiness and satisfaction of wearing such great clothes. The concept of truly great clothing includes conducting ourselves in good faith, respecting the environment and upholding human rights across all business activities.

## REPORTING ENTITY AND CORPORATE STRUCTURE OF REPORTING ENTITY

Uniqlo Australia is a wholly owned subsidiary of Fast Retailing Singapore, itself a wholly owned subsidiary of Fast Retailing Co., Ltd. (**Fast Retailing**), a company which is listed on the Tokyo Stock Exchange. Uniqlo Australia does not itself own or control other entities.

## OPERATIONS AND SUPPLY CHAINS OF THE REPORTING ENTITY

### Operations: retail sale of UNIQLO casual wear

Uniqlo Australia is a retailer of UNIQLO branded men's, women's and children's fashion apparel, footwear, small gift items and related accessories (**UNIQLO casual wear**). We employ over 1,800 people across our Melbourne headquarters and 25 Australian retail stores.

### Supply chains: UNIQLO casual wear

Uniqlo Australia does not manufacture UNIQLO casual wear.

Fast Retailing is an apparel retail group with global operations, mainly conducted through its UNIQLO casual wear brand. Fast Retailing engages with garment factories (**Partner Factories**) and fabric mills to produce fabric and manufacture garments. Fast Retailing does not own Partner Factories or fabric mills.

Uniqlo Australia procures UNIQLO casual wear through our supplier agent, Uniqlo Co., Ltd., also a subsidiary of Fast Retailing, however the Partner Factories are the ultimate suppliers of the product in the supply chain.

In this way, the Partner Factories form part of Uniqlo Australia's supply chain.

### Supply chains: other goods and services

Uniqlo Australia procures a range of other goods and services from Australian and international suppliers. For example, information and communication technology services and products, marketing and media collateral and services, various professional services

UNIQLO Australia Pty Ltd    Level 15, 385 Bourke Street, Melbourne, VIC 3000
ABN 36161922103

- Add the supporting text: *This statement is in accordance with the requirement that Uniqlo Australia Pty Ltd (Uniqlo Australia) report for the fiscal year from 1 September 2019 to 31 August 2020 (Fiscal Year 2020) under the Modern Slavery Act 2018 (Cth) (Modern Slavery Act).*
- Note: we do **not** include the following text because it does not pertain to Criterion 1: *In this statement, Uniqlo Australia adopts the meaning of modern slavery as defined in the Modern Slavery Act.*

**Example 1.3.c** (Duratec, SID#7832) for the question of Section 2.2 (Signature), a case with a full sentence and a copy-paste across the signature, all the way to the title:

The Board of Directors of Duratec, its principal governing body, has unanimously approved this Statement on 28 January 2022 and authorised Robert Philip Harcourt as the responsible executive of Duratec to sign this Statement in accordance with the Act.

Robert Philip Harcourt

Managing Director

- Add the supporting text: *The Board of Directors of Duratec, its principal governing body, has unanimously approved this Statement on 28 January 2022 and authorised Robert Philip Harcourt as the responsible executive of Duratec to sign this Statement in accordance with the Act.*
*Robert Philip Harcourt*
*Managing Director*

**Example 1.3.d** (Queensland Rail, SID#2314) for the question of Section 2.2 (Signature), a case without a full sentence, but with contiguous fragments:

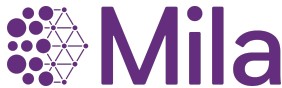

**David Marchant AM**
Chair
Queensland Rail
Queensland Rail Limited

March 2021

■ Add the supporting text: *David Marchant AM*
*Chair*
*Queensland Rail*
*Queensland Rail Limited*

*March 2021*

## 1.4. Dealing with headings

Headings (i.e. section titles) should not be extracted for any of the criteria.

**Example 1.4.a** (McGrath, SID#8609) for the question of Section 2.1 (Approval):

Principal Governing Body Approval

This modern slavery statement was approved by the board of JV McGrath Holdings Pty Ltd
as the controlling entity of the group on 30 October 2020.

Signed by

John McGrath
Managing Director

■ The section heading "*Principal Governing Body Approval*" should **not** be included in the supporting text for the Approval criterion.
■ The correct annotation for the Approval criterion is:
   ○ Add the supporting text: *This modern slavery statement was approved by the board of JV McGrath Holdings Pty Ltd as the controlling entity of the group on 30 October 2020.*
   ○ Fill the "Date" box: *30 October 2020.*

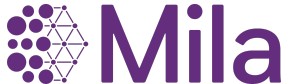

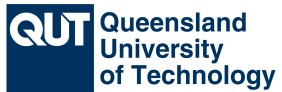

## 1.5. Dealing with difficult text formatting

If a passage contains **a list of bullet points**, annotators should copy-paste the text from the PDF without changing the pasted text with respect to the representation of the bullet point symbols. If only one or some of the bullet points contain relevant information, then **only those should be extracted**.

If some of the relevant information comes in any other format than pure text, annotators should:

- Flag that they have found this information in a visual element by selecting **as many times as necessary** from the multi-option Supplementary Button: "**Visual Element Category"** the type of visual element, namely "Infographic" (for figures and drawings) and "Table".
- For each visual element found:
    a. Check if it is possible to extract the **relevant** text from it, and if so, copy-paste it in its separate text box. Otherwise, select the supporting button "**Information found in a scanned format**".
    b. Select the Supplementary Button: "PDF page number" and add the page number(s).

Notes:

- The page number is the PDF page number from the PDF reader and not the document page number.
- If the visual element spreads over multiple pages (e.g. when it is a long table), add the list of page numbers where the relevant information within that element are found. Provide the page numbers separated by spaces.

## 1.6. Dealing with partially scanned documents

For each label, there is a possibility that sometimes the information cannot be extracted ("copy-pasted") as it is in a scanned format. In this case, please select the supporting button "**Information found in a scanned format**".

**Example 1.6.a** (Duratec, SID#7832) for the question of Section 2.4.2 (Operations):

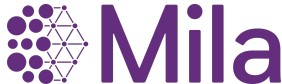

**OPERATIONS (SERVICES)**

Since Duratec's inception in 2010, the Group has grown to become a well-respected market leader in physical asset remediation and protection for clients in the Defence, Mining, Industrial, Oil and Gas and Marine sectors, as well as Government Assets and Commercial Infrastructure.

### Asset Protection

The protection of assets from corrosion is a key commercial, safety and environmental issue. Our expertise in the selection and application of high protective coating systems will significantly enhance your asset's operational performance and working lifespan.

| SERVICE OFFERING | |
|---|---|
| Abrasive blasting & painting | Cathodic protection |
| Concrete repair | High performance coatings |
| Joint remediation | Waterproofing |
| Acid protection | Flooring systems |

### Durability Engineering

We rapidly uncover the root causes of deterioration so that issues can be addressed in a timely and cost-effective manner. Once an assessment is complete, all findings are presented in a report including remedial options and realistic budget estimates.

| SERVICE OFFERING | |
|---|---|
| Condition assessment | Non-destructive testing |
| Remedial engineering | Coating inspections |
| ROV, UAV & robotic inspection | Heritage Preservation |
| Acid protection | Durability planning & materials |

### Building Refurbishment

We understand that building refurbishments need to be executed with critical attention to public safety, minimal disruption to tenants and visitors, and detailed planning and communication with all stakeholders.

| SERVICE OFFERING | |
|---|---|
| Facade refurbishments | Wet and dry fire systems |
| Asbestos & lead paint removal | Cladding replacement |
| Building fit out | Glazing replacement & repair |
| Electrical upgrades | Painting |

### Specialist Access Systems

Our in-house engineering capability enables the rapid design and installation of bespoke access systems for the delivery of inspection, maintenance and upgrade services across Australia.

| SERVICE OFFERING | |
|---|---|
| Rope access | Scaffolding |
| Drone | Technical rigging |
| Elevated work platform | Fall arrest |
| Mastclimber | Work platforms |

### Infrastructure Upgrades

Our expertise in maintaining and upgrading existing infrastructure that has reached or is nearing the end of its design life provides significant cost and time savings in comparison to new-build replacements.

| SERVICE OFFERING | |
|---|---|
| Structural strengthening | Fuel infrastructure upgrades |
| Pile cleaning & protection | Abrasive blasting and painting |
| Fender repairs and upgrades | Concrete repair |
| Electrical upgrades | Precision grouting |

### Construction Services

Our foundation was built on construction services and we are a licensed builder in all states of Australia. We deliver a range of multi-disciplined new and renovation construction projects for clients across a wide range of industry sectors.

| SERVICE OFFERING | |
|---|---|
| Commercial buildings | Bridges |
| Utilities infrastructure | Oil and gas infrastructure |
| Wharves | Healthcare facilities |
| Electrical infrastructure | Defence infrastructure |

Note: this statement is predominantly digital; however, the page shown here only has the header in a digital format. In this case, the annotator should select the supplementary button "Information found in a scanned format" when extracting supporting text for the question of Section 2.4.2 (Operations).

## 1.7. Dealing with dates

Dates are particularly relevant for the Approval and Signature criteria (Sections 2.1 and 2.2). If a date is found in the relevant text in a **copy-pastable format**, it should be copy-pasted **both** in the supporting text passage **and** in the Supplementary Button: "Date". If the date cannot be copy-pasted, the supporting

button "**Information found in a scanned format**" associated with the "Date" box should be selected and no text should be typed in the "Date" box.

When filling the Supplementary Button "Date", if there are multiple approval dates or signature dates, extract them all separated by //.

Make sure not to confuse the approval date with the signature date.

**Example 1.7.a** (Johns Lyng Group, SID#6198) for Section 2.2 (Signature):

## RISK MITIGATION ACTIONS

The independent risk assessment set out a road map consisting of additional measures Johns Lyng should take to mitigate risk of modern slavery within the Group's operations and supply chains. During FY21, John Lyng Group implemented a number of the key road map measures, including the adoption of a new Supplier Code of Conduct setting out minimum standards for Johns Lyng suppliers in the area of modern slavery, as well as other social and environmental issues.

Click the link below:
https://investors.johnslyng.com.au/FormBuilder/_Resource/_module/juURNaHpX0a6NaEh-f8lLg/file/JLG_Supplier_Code_of_Conduct_Final.pdf

Compliance with the Supplier Code of Conduct will now be a requirement in supplier Terms and Conditions and will be considered during supplier selection processes.

In addition to the above, the Group has identified modern slavery risk criteria which have been integrated into the due diligence processes prior to all new acquisitions commencing FY22. During FY22, the focus will be on strengthening modern slavery due diligence, remediation and training processes within Johns Lyng's Contractor Management Framework.

## ASSESSING EFFECTIVENESS

Responsibility for assessing and addressing modern slavery risks has been assigned to Johns Lyng's National, HSE Risk and Compliance Manager under the oversight of the Executive General Manager and the Board's Risk and Compliance Committee.

The independent risk assessment assessed the adequacy of the Group's approach to modern slavery risk management against best practices in Australia and internationally. Any identified gaps in Johns Lyng's modern slavery risk management approach are being addressed through the recommended modern slavery road map (see above).

The Group's modern slavery risk management approach is reviewed annually and reported to the Board's Risk and Compliance Committee. The next review will take place at the conclusion of FY22.

## CONSULTATION WITH ENTITIES OWNED OR CONTROLLED

Johns Lyng consulted with the relevant companies it owns or controls in the development of this statement.

This statement was approved by the Johns Lyng Group Board on 13 October 2021.

**Peter Nash**
Chairman
13 October 2021

**Scott Didier AM**
Managing Director
13 October 2021

- Add the supporting text: *Peter Nash Chairman 13 October 2021 Scott Didier AM Managing Director 13 October 2021*
- Select the Supplementary Button: "Date" and add: *13 October 2021 // 13 October 2021*

■ Select the Supplementary Button: *"Signature Image"*

**Example 1.7.b** (McGrath, SID#8609) a case with a date for Approval, but without a signature date:

Principal Governing Body Approval

This modern slavery statement was approved by the board of JV McGrath Holdings Pty Ltd
as the controlling entity of the group on 30 October 2020.

Signed by

John McGrath
Managing Director

**Example 1.7.c** (SXP Group, SID#1276) a case with a date for Signature, but without an approval date:

This statement has been approved by the Board of SXP Group.

David Medhurst
Managing Director
SXP Group

23 December 2020

**Example 1.7.d** (Uniqlo, SID#1535) a case with a clear distinction between approval and signature dates:

**APPROVAL OF STATEMENT**

In accordance with section 13 of the Modern Slavery Act, this statement was approved by the Board of Uniqlo Australia Pty Ltd on 23 February 2021.

Kensuke Suwa
Director and Co–Chief Operating Officer, Uniqlo Australia Pty Ltd
24 February 2021

**Example 1.7.e** (Dalrymple Bay Infrastructure, SID#4480) a case where both the approval and signature dates are present, and where they should be extracted in their respective date box:

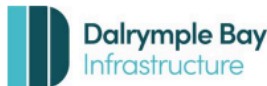

**8.    Board Approval**

This statement was approved by the Board of Dalrymple Bay Infrastructure Limited on 24 June 2021.

**David Hamill**
Chairman
Dalrymple Bay Infrastructure Limited
24 June 2021

Note: in **Example 1.7.e** above, and as shown in **Example 1.2.b,** when a date relevant for a specific question is repeated multiple times, all instances of this date should be included in the "Date" box. In that particular case, it is the approval date (24 June 2021) that is repeated twice, once on the cover page, and once at the end.

## 1.8. General observations

- The extracted text is unlimited, so the annotators should extract all relevant text for each question.

- The extracted text should always come from the statement itself, and never from other documents.

- The entire content of each statement should be read to extract information for all questions.

- The same sentence can be used as supporting text for the labels of multiple questions when relevant for each one of them.

- **Negative declarations** can be considered useful supporting text for the labels of all questions, and should be extracted when relevant (e.g. "*we do not have facilities in Australia*" or "*we do not manufacture our own parts*" for the question of Section 2.4.2, Operations).

- The annotators should ensure no noisy data is included in the supporting text box, such as page numbers, information from headers, footers, etc.

- The only place where the annotators can add their own words is under the "Explanation" Supplementary text box when selecting the "Unclear" label.

- Some footnotes contain relevant information for a label and should also be added in the supporting text passages. The numbers of the footnotes should be kept.

**Example 1.8.a** (Volvo, SID#8717) for the question of Section 2.5.1 (risk description), a case where the footnote should be extracted, along with its number:

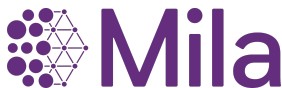
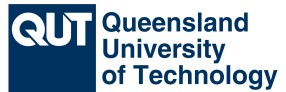

## Risks of modern slavery in our supply chains

As a subsidiary of a global company with complex supply chains, Volvo Car Australia acknowledges that challenges connected to our operations exist. Modern slavery risk covers a broad range of practices and includes slavery, debt bondage, forced labour, child labour, forced marriage and other exploitative practices including wage theft, abuse and suppression of freedom of association.

In our work to secure responsible sourcing we take a risk-based approach. Based on this approach, we have enhanced our due diligence efforts for suppliers located in high-risk countries (discussed further below). Moreover, with our shift to electrification, we acknowledge the ethical, social and environmental risks (including modern slavery) connected to the extraction, processing, trade and transportation of the raw materials needed for the production of EV batteries (e.g. cobalt, nickel, lithium). Tin, tantalum, tungsten and gold – also known as '3TG' or 'conflict minerals²' – are other raw materials used in components in our cars, which in some cases are related to negative impacts on people and planet, including modern slavery. To prevent and mitigate these risks, we have implemented due diligence efforts for these supply chains (discussed further below).

The Volvo Car Group acknowledges that due diligence is an ongoing, proactive, and reactive

process and that information and risk monitoring systems will progressively be improved over time as a result from constructive engagement with suppliers and other stakeholders.

There is also the risk that modern slavery can occur more locally, closer to home. Volvo Car Australia is aware and remains vigilant about the increased risk connected to other vulnerable groups such as base-skilled workers engaged by our business partners in Australia.

## Our approach towards subcontracting

Subcontracting by our suppliers is a usual practice in the manufacturing and automotive industry. Volvo Cars manages subcontracting by imposing rules for subcontracting in its contracts with suppliers. The terms include a requirement that subcontractors accept Our Code of Conduct for Business Partners, follow the same terms as Volvo Cars has agreed with suppliers and includes the right to audit any subcontractor. Volvo Cars continuously seeks improvements on its ability to monitor its entire supply chain.

At Volvo Car Australia we manage this by permitting subcontracting, subject to our consent and subject to any subcontractor also agreeing to Our Code of Conduct for Business Partners and ensuring we have the right to audit any subcontractor.

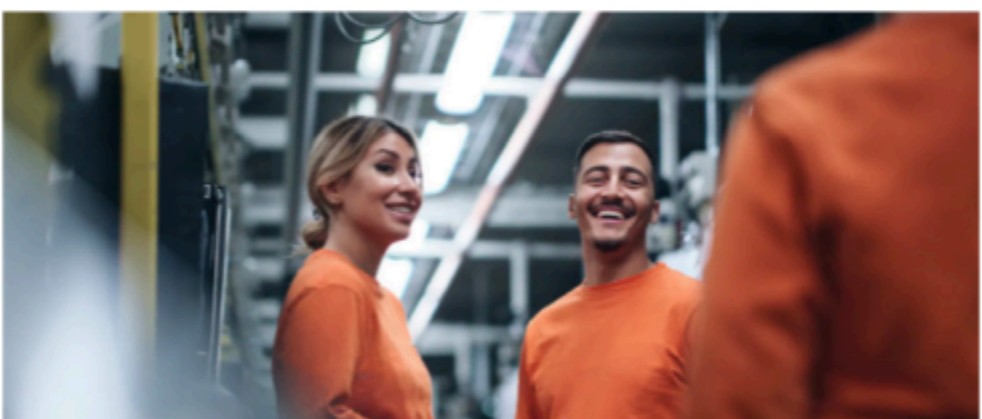

2. "Conflict minerals" i.e. tin, tantalum, tungsten and gold (3TG) from the Democratic Republic of the Congo or its nine adjoining countries as well as other minerals from conflict-affected and high-risk areas.

VOLVO CAR AUSTRALIA
MODERN SLAVERY STATEMENT 2021          V O L V O                7

■ Add the supporting text: *In our work to secure responsible sourcing we take a risk-based approach. Based on this approach, we have enhanced our due diligence efforts for suppliers located in high-risk countries (discussed further below). Moreover, with our shift to electrification, we acknowledge the ethical, social and environmental risks (including modern slavery) connected to the extraction, processing, trade and transportation of the raw materials needed for the production of EV batteries (e.g. cobalt, nickel, lithium). Tin, tantalum, tungsten and gold – also known as '3TG' or 'conflict minerals²' – are other raw materials used in components in our cars, which in some cases are related to negative impacts on people and planet, including modern slavery. To prevent and mitigate these risks, we have implemented due diligence efforts for these supply chains (discussed further below). // There is also the risk that modern slavery can occur more locally, closer to home. Volvo Car Australia is aware and remains vigilant about the increased risk connected to other vulnerable groups such as base-skilled workers engaged by our business partners in Australia. // 2. "Conflict minerals" i.e. tin, tantalum, tungsten and gold (3TG) from the Democratic Republic of the Congo or its nine adjoining countries as well as other minerals from conflict-affected and high-risk areas.*

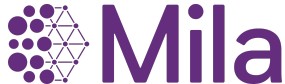 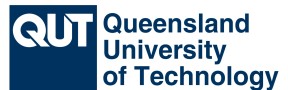

# 2. Questions to be answered by annotators

## 2.1. Approval

**Question:** is the statement approved by the entity's principal governing body?

There are two categories of statements:

1. "Normal" statement that pertains to a single entity subject to the law in a given year, having a turnover exceeding 100 million AUD.
2. "Joint" statement is a collective statement from multiple entities subject to the law, each having a turnover exceeding 100 million AUD.

The type of each statement ("normal" or "joint") is provided by the reporting entity and available as metadata on the Australian Modern Slavery Register, but it is unreliable.

Note that for this question, approval dates should always be extracted if available.

## 2.1.1. "Normal" statement - for a single reporting entity

**Background:** Section 13 of the Act requires statements given by single reporting entities to be approved by the principal governing body of the reporting entity.

The principal governing body is the body or group of members with primary responsibility for the governance of the reporting entity.

The Act does not permit delegation of approval authority, including to an Individual, Executive Committee, Sub-Committee, or Workgroup.

To ensure an entity's statement can be published on the government's Register, entities should ensure there is no ambiguity in who approved the statement. Groups like 'Executive Leadership Committee', 'Members of the Board', and 'Senior Executive on behalf of the Board' leave it unclear whether the entity's principal governing body has cleared their modern slavery statement, or if it was approved by a select few members. Any language that indicates that the statement was "considered by the board" or "put forward to the board" is not sufficient to meet this criterion.

**Label options & their associated supporting text passages extracted from the statements:**

○ Yes:
   ■ Copy-Paste the exact text passage or text passages from the statement that shows that the statement is approved by the reporting entity's principal governing body, including the date of approval if it exists.

- ■ If any relevant information comes in other formats than text, select: Supplementary Button: "Visual Element Category".
- ■ If an approval date is found, make sure to copy-paste it in the supporting text as well as to select the Supplementary Button: "Date" and insert the date. If the signature date is in a scanned (not-copy-pastable) format, you must select "Information found in a scanned format".
- ○ No:
    - ■ Copy-Paste the exact text passage or text passages from the statement that justifies that the reporting entity does not meet this criterion (e.g. "the board does not approve this statement"), or
    - ■ If no information is found about this label, please select the Supplementary Button: "No relevant information found in the statement".
- ○ Unclear:
    - ■ Please select this label if the information found is unclear or there are other concerns. If you decide to select this label, you have to choose the Compulsory Supplementary Button "Explanation" and justify why.

---

**Annotation Workflow:**
**The labels and their supplementary buttons should be:**
- ● Yes
    - ■ Copy-Paste supporting text passage
    - ■ Supplementary Button: "Visual Element Category"
    - ■ Supplementary Button: "Date"
- ● No
    - ■ Copy-Paste supporting text passage
        **OR**
    - ■ Supplementary Button: "No relevant information found in the statement"
- ● Unclear
    - ■ Compulsory Supplementary Button "Explanation"

---

**Example 2.1.1.a** (Mineral Resources Limited, SID#1674):

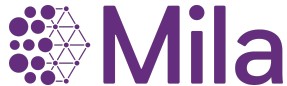
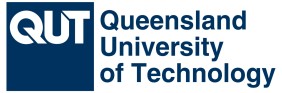

## CONTENTS

## ABOUT THIS STATEMENT

This is Mineral Resources Limited's (MRL) first Modern Slavery Statement. It has been prepared to meet the reporting requirements under the *Modern Slavery Act, 2018 (Cth)* for the year ended 30 June 2020 (FY20).

All references to 'MRL', 'the Company', 'the Group', 'we', 'us' and 'our' refer to Mineral Resources Limited (ABN 33 118 549 910) and the entities it controlled, unless otherwise stated. References in this report to a 'year' are to the financial year ended 30 June 2020 unless otherwise stated. All dollar figures are expressed in Australian dollars (AUD) unless otherwise stated. All references to 'Indigenous' people are intended to include Aboriginal and/or Torres Strait Islander people.

Any questions about this statement can be sent to:
investorrelations@mrl.com.au

This statement has been approved by MRL's Board of Directors on 25 February 2021.

Peter Wade
Non-Executive Chairman

- Select label: **Yes**
- Add the supporting text: *This statement has been approved by MRL's Board of Directors on 25 February 2021.*
- Select the supplementary Button: "Date" and insert the date: *25 February 2021*

**Example 2.1.1.b** (SXP Group, SID#1276):

Modern Slavery Statement 2020

**WHISTLEBLOWER HOTLINE**

In January 2020, SXP's Whistleblower Policy came into effect. The policy outlines the various types of reportable conduct, protections for individuals who disclose wrongdoing, our legal obligations, means for individuals to make disclosures and our investigation processes. The policy was published and made accessible on our workforce communications platform and our company website. Training was also provided to relevant employees, management personnel and employees with direct responsibilities for the oversight and management of the policy.

With the Whistleblower Policy we established our Whistleblower hotline which enables individuals to anonymously report suspected misconduct, including matters related to modern slavery practices. The hotline is managed by an external provider which assists in maintaining anonymity of the whistleblower and communications between SXP and the whistleblower. Any allegations of misconduct received are reviewed, appropriately investigated and where necessary corrective or disciplinary actions undertaken; including, how to prevent further breaches occurring.

## ASSESSMENT & EFFECTIVENESS OF OUR ACTIONS

SXP considers the measures undertaken in FY19-20 have been effective thus far in identifying and addressing potential modern slavery risks in our operations and supply chains.

We recognise the importance of continuing monitoring to determine effectiveness and make improvements in our modern slavery processes. Further and future assessment of the effectiveness of our actions will be undertaken by monitoring and analysing:

- Ethical sourcing questionnaire results
- Audit and review results
- Completion rates for training and awareness

The Governance, Risk and Compliance team will be responsible for monitoring and reporting on the effectiveness of our modern slavery processes. These reports will be regularly shared with SXP's senior leadership team with the goal of collaborating on the management of modern slavery risks in the operations and supply chains, and process improvement.

## OUR FUTURE COMMITMENTS

SXP recognises the importance of developing our understanding, oversight, and management of modern slavery risks in our operations and supply chains.

Over the coming financial year, we will:

- Continue to review our subcontractor and supplier arrangements
- Continue to review and improve policies and procedures
- Incorporate terms into our subcontractor and supplier agreements which provide assurance that they are aware of and comply with modern slavery obligations
- Further education and awareness for our workforce

This statement has been approved by the Board of SXP Group.

David Medhurst
Managing Director
SXP Group

23 December 2020

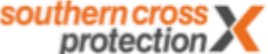

5

■ Select label: **Yes**

■ Add the supporting text: *This statement has been approved by the Board of SXP Group.*

**Example 2.1.1.c** (Equity Trustees, SID#7480)**:**

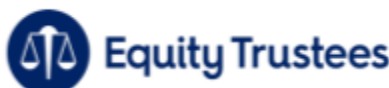

## Background

This document is a Modern Slavery Statement (**'Statement'**) made pursuant to section 14 of the *Modern Slavery Act 2018(Cth) (the* **'Act'**)in respect to the financial year ending 30 June 2021 (the **'Reporting Period'**) in relation to the Regal Investment Fund a 'Reporting Entity' under the Act.

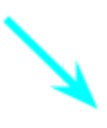

The Statement is issued by Equity Trustees Limited ABN 46 004 031 298 (**'ETL'** or **'Responsible Entity'**) in its capacity as the responsible entity of the Reporting Entity in accordance with the Act and has been approved by the Board of ETL as their principal governing body

This Statement discloses ETL and the Fund's commitment and steps taken in identifying and addressing modern slavery risks within its business and supply chain, which includes the activities implemented in financial year ending 30 June 2021.

This is the first Statement for the Regal Investment Fund.

[many pages]

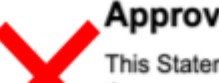

## Approval

This Statement was approved and signed by Philip Gentry on behalf of the Board of Directors of ETL, the responsible entity of the Reporting Entities, on 23 December 2021.

Philip Gentry
Chair

Date: 23 December 2021

This is an example where the Approval label should be "Yes", but care must be taken to extract the relevant supporting information. The "Approval" section with the Chair's signature does not provide

support for the Approval criterion: approval should be from the complete board, not from someone on behalf of the board. However, earlier in the text, there is a sentence that provides strong and explicit evidence of the board's approval. Thus, the correct annotation is:

- Select label: **Yes**
- Add the supporting text: *The Statement is issued by Equity Trustees Limited ABN 46 004 031 298 ('ETL' or 'Responsible Entity') in its capacity as the responsible entity of the Reporting Entity in accordance with the Act and has been approved by the Board of ETL as their principal governing body*

**Example 2.1.1.d** (Langdon, SID#11808)**:**

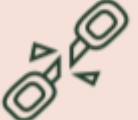
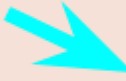

# MODERN SLAVERY REPORT

This is the third Modern Slavery Statement from H.J. Langdon & Co Pty Ltd (ABN 89 006 641 701) under the Modern Slavery Act 2018 Australia. This report was prepared for fiscal year 2021-2022.

The purpose of this statement is to comply with reporting requirements under the Australian Act to demonstrate an understanding of risk within our organisation and extended supply chain for Modern Slavery and the measures taken to address those risks.

This statement has been approved by the Board of H.J. Langdon & Co Pty Ltd (referred to in this statement as the "Company").

We recognise that our corporate and social responsibilities reside as much in our supply chain as they do in our own activities. Therefore, we continually seek to ensure the ingredients and services we source are ethically produced. We do this by working with our suppliers to positively influence our social, ethical and environmental performance. The Langdon Supplier Code of Conduct, Modern Slavery & Vendor Risk Review along with the Supplier Ethical Data Exchange (SEDEX) program are tools used to assess our Supplier's social and environmental compliance.

The Company is a family-owned business. Our supply partners and processes are an extension of our business and the protection of human rights across our wide-reaching supplier base is paramount.

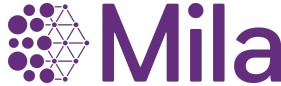

*[many pages]*

# CONSULTATIONS & NEXT STEPS

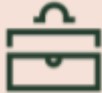

The company is a collaborative organisation. Further development and continuous improvement of our ESR framework including modern slavery is managed through consultation with all key stake holders within our business and in our supply chain.

Over the next reporting period, we will;

- Leverage further the SEDEX portal tool and principles to manage our suppliers
- Continue our Labour Hire compliance audits
- Continue to process Modern Slavery Risk Assessments
- Continue to raise awareness of modern slavery throughout the organisation
- Strengthen Ethical Sourcing training by outsourcing training with industry expertise.
- Continue to deliver on our ESR annual Goals
- Monitor for the impacts of Covid 19 on the food supply chain

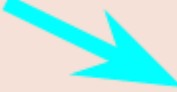

Approval

This statement was approved by the Board of H.J. Langdon & Co Pty Ltd and signed by a responsible member of H.J. Langdon & Co Pty Ltd on 22/12/22

Chris Langdon
Director & CEO of H.J. Langdon & Co Pty Ltd

- Select label: **Yes**
- Add the supporting text: *This statement has been approved by the Board of H.J. Langdon & Co Pty Ltd (referred to in this statement as the "Company").* // *This statement was approved by the Board of H.J. Langdon & Co Pty Ltd and signed by a responsible member of H.J. Langdon & Co Pty Ltd on 22/12/22*
- Select the supplementary Button: "Date" and insert the date: *22/12/22*

**Example 2.1.1.e** (IRT, SID#11954):

# Statement by the Chair and CEO

As a leading not for profit provider of aged care services, our approach to modern slavery is based on our values of Integrity, Respect and Trust. We are very aware of the need to protect the most vulnerable members of our society by ensuring that our processes identify potential risks in our supply chains and where necessary, establish remediation plans.

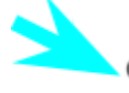 Our modern slavery statement for 1 July 2021 – 30 June 2022 has been endorsed by the IRT Board of Directors.

**Mike Halloran**
Chair

**Patrick Reid**
CEO

This is an example where the statement is correctly approved ("endorsed" is synonymous to "approved" in this context). However, there is no approval date; the range 1 July 2021 - 30 June 2022, which represents the reporting period, should not be confused for the approval date.

- Select label: **Yes**
- Add the supporting text: *Our modern slavery statement for 1 July 2021 – 30 June 2022 has been endorsed by the IRT Board of Directors.*

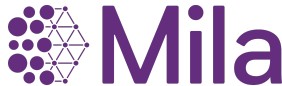
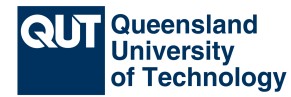

**Example 2.1.1.f** (LLREIL, SID#11569):

# Consultation process in preparing this Modern Slavery Statement

In preparing this Statement, fund managers and property managers have been engaged in considering modern slavery risk issues, actions being taken to mitigate those risks and reporting requirements.

This has been done through the adoption of systems and processes, which were developed and implemented by Lendlease's Group Supply Chain function.

APPFC Passive also participates in Lendlease's Australian Modern Slavery Community of Practice, which has cross-functional representation across Lendlease in Australia that includes APPFC Passive's operations and entities.

Representatives include sustainability, supply chain, risk, legal, and operations functions from Lendlease's construction, investment and development businesses.

Refer to pages 32 and 33 of the Lendlease Group Modern Slavery Statement FY22 for further details.

There has also been consultation with a range of personnel who operate across the entities covered by this Statement.

The LLREIL Board considers and approves the Statement.

This Statement is made pursuant to section 13 of the *Modern Slavery Act 2018* (Cth). Development of this Statement has been considered by the Board of Directors. ✗

This Statement has been approved by the Board of Directors of Lendlease Real Estate Investments Limited, on behalf of APPFC Passive, who will review and update it as necessary.

Signed:

Director name: Vanessa Orth

Date:⁴  8 December 2022

- Select label: **Yes**
- Add the supporting text: *The LLREIL Board considers and approves the Statement.* // *This Statement has been approved by the Board of Directors of Lendlease Real Estate Investments Limited, on behalf of APPFC Passive, who will review and update it as necessary.*
- Note: the paragraph starting with "*This statement is made pursuant…*" is not added to the supporting text since it does not mention anything regarding the approval by the board.

## 2.1.2. "Joint" statement - for multiple entities

**Background**: section 14 of the Act provides three options for approval of joint statements:

- **OPTION ONE**: The principal governing body of each reporting entity covered by the statement approves the statement.
- **OPTION TWO**: The principal governing body of a higher entity (such as a global parent), which is in a position to influence or control each entity covered by the statement, approves the statement. The higher entity does not have to be a reporting entity itself.
- **OPTION THREE**: If it is not practicable to comply with Option One or Option Two, the principal governing body of at least one reporting entity covered by the statement may approve the statement. In this case, the statement must also explain why this option was taken.

**Label options & their associated supporting text passages extracted from the statements:**

- Yes:
    - Copy-Paste the exact text passage or text passages from the statement that explains that it is approved by the principal governing body of the reporting entities or of a higher entity, including the date of approval if it exists.
    - **Compulsory** Supplementary Button: "Options"
        - If Option 3 is chosen: the explanation provided in the statement must be included in the supporting text passages.
    - If any relevant information comes in other formats than text, please select Supplementary Button: "**Visual Element Category**".
    - If a date is found, make sure to copy-paste it in the supporting text as well as to select the Supplementary Button: "Date" and insert the date. If the signature date is in a scanned (not-copy-pastable) format, select "Information found in a scanned format".
- No:

- ■ Copy-Paste the exact text passage or text passages from the statement that justifies that a reporting entity does not meet this criterion (e.g. "the board does not approve this statement"), or
- ■ If no information is found about this label, please select the Supplementary Button: "No relevant information found in the statement".
- ○ Unclear:
  - ■ Please select this label if the information found is unclear or there are other concerns. If you decide to select this label, you have to choose the Compulsory Supplementary Button "Explanation" and justify why.

---

**Annotation Workflow:**
**The labels and their supplementary buttons should be:**
- ● Yes
  - ■ Copy-Paste supporting text passage
  - ■ Compulsory Supplementary button: "Options"
  - ■ Supplementary Button: "Visual Element Category"
  - ■ Supplementary Button: "Date"
- ● No
  - ■ Copy-Paste supporting text passage
    **OR**
  - ■ Supplementary Button: "No relevant information found in the statement
- ● Unclear
  - ■ Compulsory Supplementary Button "Explanation"

---

**Example 2.1.2.a** (NQPB+PCQ+MPL, SID#10698):

2. NQBP will undertake an internal review of its tender forms and contracts for compliance with the Supplier Code, Ethical Mandate and modern slavery criteria during 1 July 2022 – 30 June 2023 reporting period.

3. NQBP together with AMSA will run a modern slavery presentation on the risk of modern slavery with international vessels for NQBP's Pilotage team.

4. NQBP (through its Pilotage team) will engage with AMSA and report any instances of modern slavery.

5. NQBP will undertake an audit of its General Workforce EA for compliance during the 1 July 2022 – 30 June 2023 Reporting Period.

## Consultation with controlled entities

NQBP's controlled entities, PCQ and MPL are wholly owned and controlled by NQBP. NQBP and its controlled entities are all covered by NQBP's policies and procedures, and are treated as one operational entity from a corporate governance perspective. Accordingly, NQBP did not consider that formal consultation with its controlled entities was necessary or useful in preparing this Statement.

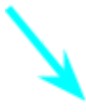

## Board approval

This Statement was approved by the Boards of NQBP, PCQ and MPL on Tuesday, 22 November 2022.

Jane McTaggart

Chair of the Boards of NQBP, PCQ and MPL.

- Select label: **Yes**
- Add the supporting text: *This Statement was approved by the Boards of NQBP, PCQ and MPL on Tuesday, 22 November 2022.*
- Select the supplementary Button: "Date" and insert the date: *22 November 2022*
- Select the supplementary Button: "Options" and select "Option 1" because the statement was approved by the board of each reporting entity.

**Example 2.1.2.b** (Honda, SID#10029):

\*\*\*\*\*\*\*\*

This statement was approved by the Board of Directors for Honda AUH and
Honda MPE on September 6th, 2022.

> DocuSigned by:
>
> _[signature]_
>
> 23EA4406648348E

Stephen Collins
Chief Operating Officer
Honda Australia Pty. Ltd

- ■ Select label: **Yes**
- ■ Add the supporting text: *This statement was approved by the Board of Directors for Honda AUH and Honda MPE on September 6th, 2022.*
- ■ Select the supplementary Button: "Date" and insert the date: *September 6th, 2022*
- ■ Select the supplementary Button: "Options" and select "Option 1" because the statement was approved by the board of each reporting entity.

**Example 2.1.2.c** (Queensland Rail, SID#2314):

# Consultation and endorsement

### Consultation

This Statement was prepared in consultation with both reporting entities covered by this joint statement (Queensland Rail and QRL) and with subsidiaries of QRL. The consultation included discussion with many business units including Procurement, Contracts and Supplier Relations, Legal, Analytics and Reporting, Remuneration and Organisational Design, Property and Probity.

### Endorsement

The Board for both Queensland Rail and QRL has endorsed this Queensland Rail Modern Slavery Statement FY2019-20.

_[signature]_

**David Marchant AM**
Chair
Queensland Rail
Queensland Rail Limited

March 2021

- Select label: **Yes**
- Add the supporting text: *The Board for both Queensland Rail and QRL has endorsed this Queensland Rail Modern Slavery Statement.*
- Select the supplementary Button: "Options" and select "Option 1" because the statement was approved by the board of all reporting entities.
- Note that the words "approval" or "approved" are not used. Instead, we see the word "endorsed" which is synonymous with "approved".

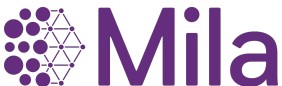 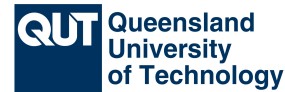

## 2.2. Signature

**Question:** is the statement signed by a responsible member of the reporting entity?

**Background:** Section 13 of the Act requires statements given by single reporting entities to be signed by a responsible member of the reporting entity. Section 14 of the Act sets out the same requirement for joint statements. In case there is a joint statement, the signature of a responsible member of one reporting entity's governing body is sufficient.

A responsible member is a decision-making member of the reporting entity. Where a reporting entity has a principal governing body, the responsible member will be a member of that body.

A statement cannot be signed by individuals who do not meet the definition of a responsible member.

It is important to ensure that the text accompanying the signature shows that the individual signing has the authority to sign the statement by including all the following information:

- The **name of the responsible member;**
- The **title or position** (e.g., CEO, Director, Member of the Board, Chairman, President, etc.) of the responsible member; and
- A **copy of their signature** (an image) OR **wording to explicitly indicate the signature of the statement** (e.g. "signed by", "signature", "s/", a signature ID, or any form of electronic signature).

For this criterion, we should also check for a signature date, if available.

**Note:** when extracting the supporting text for this criterion, if there are **small** segments of irrelevant formatting text (for example, "*NAME:*        ", "*DATE: xx/xx/xx*", or "*Signed by: ___________*"), it is OK to include those segments. It is also OK if the copy-paste operation is done over the signature image without using //.

**Label options & their associated supporting text passages extracted from the statements:**

○ Yes:
- Copy-Paste the exact text passage or text passages from the statement that shows that it is signed by a responsible member of the reporting entity, including the date of signature if it exists.
- If there is an image of the signature, please select the Supplementary Button: "Signature Image".
- If a signature date is found in a copy-pastable format, make sure to copy-paste it in the supporting text as well as to select the Supplementary Button: "Date" and insert the date. If

the signature date is in a scanned (not-copy-pastable) format, you must select "Information found in a scanned format".

- ○ No:
    - ■ Copy-Paste the exact text passage or text passages from the statement that justifies that the reporting entity does not meet this criterion, **OR**
    - ■ If no information is found about this label, please select the Supplementary Button: "No relevant information found in the statement".
- ○ Unclear:
    - ■ Please select this label if the information found is unclear or there are other concerns. If you decide to select this label, you have to choose the Compulsory Supplementary Button "Explanation" and justify why.

---

**Annotation Workflow:**
**The labels and their supplementary buttons should be:**
- ● Yes
    - ■ Copy-Paste supporting text passage
    - ■ Supplementary Button: "Signature Image"
    - ■ Supplementary Button: "Visual Element Category"
    - ■ Supplementary Button: "Date"
- ● No
    - ■ Copy-Paste supporting text passage
      **OR**
    - ■ Supplementary Button: "No relevant information found in the statement"
- ● Unclear
    - ■ Compulsory Supplementary Button "Explanation"

---

**Example 2.2.a** (TransferWise, SID#1004):

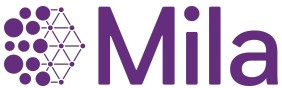
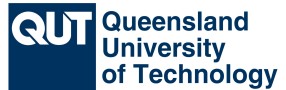

## ONGOING ASSESSMENT

On an annual basis, our senior management will delegate a review of the effectiveness of the processes we have in place within the TransferWise group to assess and address modern slavery and human trafficking risks. We also carry out internal audits which cover such compliance processes.

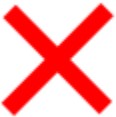

_______________

**Name**: Kristo Kaarmann

**Title**: Director

**Date**: 29 December 2020

- Select label: **No**
- Note: This statement is missing a signature or wording indicating that it was signed. As a result, it should be labeled as "No" for the Signature criterion.

**Example 2.2.b** (Duratec, SID#7832):

The Board of Directors of Duratec, its principal governing body, has unanimously approved this Statement on 28 January 2022 and authorised Robert Philip Harcourt as the responsible executive of Duratec to sign this Statement in accordance with the Act.

Robert Philip Harcourt

Managing Director

- Select label: **Yes**
- Add the supporting text: _The Board of Directors of Duratec, its principal governing body, has unanimously approved this Statement on 28 January 2022 and authorised Robert_

*Philip Harcourt as the responsible executive of Duratec to sign this Statement in accordance with the Act. Robert Philip Harcourt Managing Director*

- ■ Select the Supplementary Button: "Signature Image"

**Example 2.2.c** (Queensland Rail, SID#2314):

David Marchant AM
Chair
Queensland Rail
Queensland Rail Limited

March 2021

- ■ Select label: **Yes**
- ■ Add the supporting text: *David Marchant AM Chair Queensland Rail Queensland Rail Limited March 2021*
- ■ Select the Supplementary Button: "Signature Image"
- ■ Fill the "Date" box: *March 2021*

**Example 2.2.d** (Vestas, SID#9309):

This Statement was approved by the Board of Directors of Vestas - Australian Wind Technology Pty Ltd on 29 June 2022.

Danny Nielsen
Director, Vestas – Australian Wind Technology Pty Ltd
DATE: 29 June, 2022

- ■ Select label: **Yes**
- ■ Add the supporting text: *Danny Nielsen Director, Vestas – Australian Wind Technology Pty Ltd DATE: 29 June, 2022*
- ■ Select the Supplementary Button: "Signature Image"
- ■ Fill the "Date" box: *29 June, 2022*

- Note: it is OK to include the string "*DATE:*" in the supporting text as this simplifies the copy-pasting of the text. But do **not** include it in the "Date" box.

**Example 2.2.e** (Swiss Re, SID#8933):

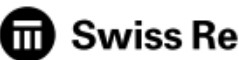

Swiss Re Australian Modern Slavery Statement

## Approval

Swiss Re Asia Holding Pte. Ltd is the parent entity of the three Reinsurance Swiss Re Reporting Entities i.e. SRAL, SRLHA and SRAuL, approving the Statement on their behalf.

The Board of Directors of Swiss Re Asia Holding Pte. Ltd, Swiss Re International SE and Swiss Pillar Investments Ltd have duly approved this Statement respectively.

Digitally signed by John Dacey
Date: 16.06.2022
11:39:43

Signature ________________________
Name: John Dacey
Director
Swiss Re Asia Holding Pte. Ltd.
on behalf of
Swiss Re Asia Pte. Ltd.
Swiss Re Australia Ltd
Swiss Re Life & Health Australia Limited

Digitally signed by Hermann Geiger
Date: 22.06.2022
16:24:41

Signature ________________________
Name: Hermann Geiger
Director
Swiss Re International SE

Swiss Reinsurance Company Ltd.
Zurich
Digitally signed by Claudia Cordioli
Date: 16.06.2022
13:59:03

Claudia_Cordioli@swissre.com

Signature ________________________
Name: Claudia Cordioli
Director
Swiss Pillar Investments Ltd

- Select label: **Yes**
- Add the supporting text: *Signature _______________________ Signature _______________________ Name: John Dacey Name: Hermann Geiger*

*Director Director Swiss Re Asia Holding Pte. Ltd. on behalf of Swiss Re Asia Pte. Ltd. Swiss Re Australia Ltd Swiss Re Life & Health Australia Limited Swiss Re International SE Signature ___________________________ Name: Claudia Cordioli Director Swiss Pillar Investments Ltd*

- ■ Select the Supplementary Button: "Signature Image".
- ■ Select "Information found in a scanned format" for the "Date" box because the signature dates are in non-copy-pastable formats (scanned).
- ■ Note that it is ok to do a single copy-paste of the full text even though it includes strings such as "*Signature ___________________________*" and "*Name:*".

**Example 2.2.f** (Solo Group, SID#7997):

This statement has been approved and endorsed by the Solo Group Board of Directors on the 26th of February 2021.

Robert Richards (Managing Director)

_______________________________

Gillian Richards (Director)

_______________________________

Rhys Richards (Director)

_______________________________

- Select label: **Yes**
- Add the supporting text: *Robert Richards (Managing Director)*
  _______________________________________ *Gillian Richards (Director)*
  _______________________________________ *Rhys Richards (Director)*
  _______________________________________
- Select the Supplementary Button: "Signature Image".
- Note: it is OK to do a single copy-paste of the full text even though it includes strings such as "_______________________________________".

**Example 2.2.g** (Equity Trustees, SID#2499):

**Approval**

This Statement was approved by Philip Gentry on behalf of the ETL Board of Directors

_______________________________________
Philip Gentry

Chair

Date:    31 March 2021

_______________________________________

- Select label: **Yes**
- Add the supporting text: *Philip Gentry Chair Date: 31 March 2021*
- Select the Supplementary Button: "Signature Image"
- Fill the "Date" box: *31 March 2021*
- Note: it is OK to include the "Date:" in the supporting text as this simplifies the copy-pasting of the text. But do **not** include it in the "Date" box.

**Example 2.2.h** (Herbert Smith Freehills, SID#5916):

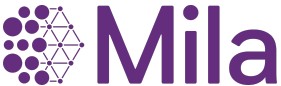
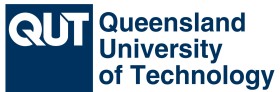

The Designated Members of Herbert Smith
Freehills Paris LLP approved this statement on
behalf of the Members of Herbert Smith
Freehills Paris LLP on 6 October 2021.

*Rebecca Maslen-Stannage*

**Rebecca Maslen-Stannage**
**Chair and Senior Partner**
**Herbert Smith Freehills, an Australian**
**partnership**

**September 2021**

**Donald Rowlands**
**Partner**
**Herbert Smith Freehills LLP**

**September 2021**

**Alissa Anderson**
**Director**
**Exchange House Services Australia Pty Ltd**

**September 2021**

**Frédéric Bouvet**
**Designated Member**
**Herbert Smith Freehills Paris LLP**

**October 2021**

**Alison Brown**
**Executive Partner**
**Exchange House Services Limited**

**October 2021**

- Select label: **Yes**
- Add the supporting text: *Rebecca Maslen-Stannage*
  *Chair and Senior Partner Herbert Smith Freehills, an Australian partnership*
  *September 2021*
  *Donald Rowlands Partner Herbert Smith Freehills LLP*
  *September 2021*
  *Alissa Anderson Director Exchange House Services Australia Pty Ltd*
  *September 2021*
  *Frédéric Bouvet Designated Member Herbert Smith Freehills Paris LLP*
  *October 2021*
  *Alison Brown Executive Partner Exchange House Services Limited*
  *October 2021*
- Select the Supplementary Button: "Signature Image"
- Fill the "Date" box: *September 2021 // September 2021 // September 2021 // October 2021 // October 2021*

**Example 2.2.i** (Safety Mate, SID#10017):

**SIGNATURE OF RESPONSIBLE MEMBER**

This modern slavery statement is signed by a responsible member of Safety Mate, as defined by the Act, on this 7th day of September 2022:

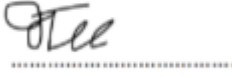

**Clint Tee**

(Director)

- Select label: **Yes**
- Add the supporting text: *This modern slavery statement is signed by a responsible member of Safety Mate, as defined by the Act, on this 7 th day of September 2022: Clint Tee (Director)*
- Select the Supplementary Button: "Signature Image"
- Fill the "Date" box: *7 th day of September 2022*

**Example 2.2.j** (Beca, SID#10015):

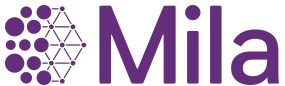
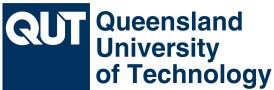

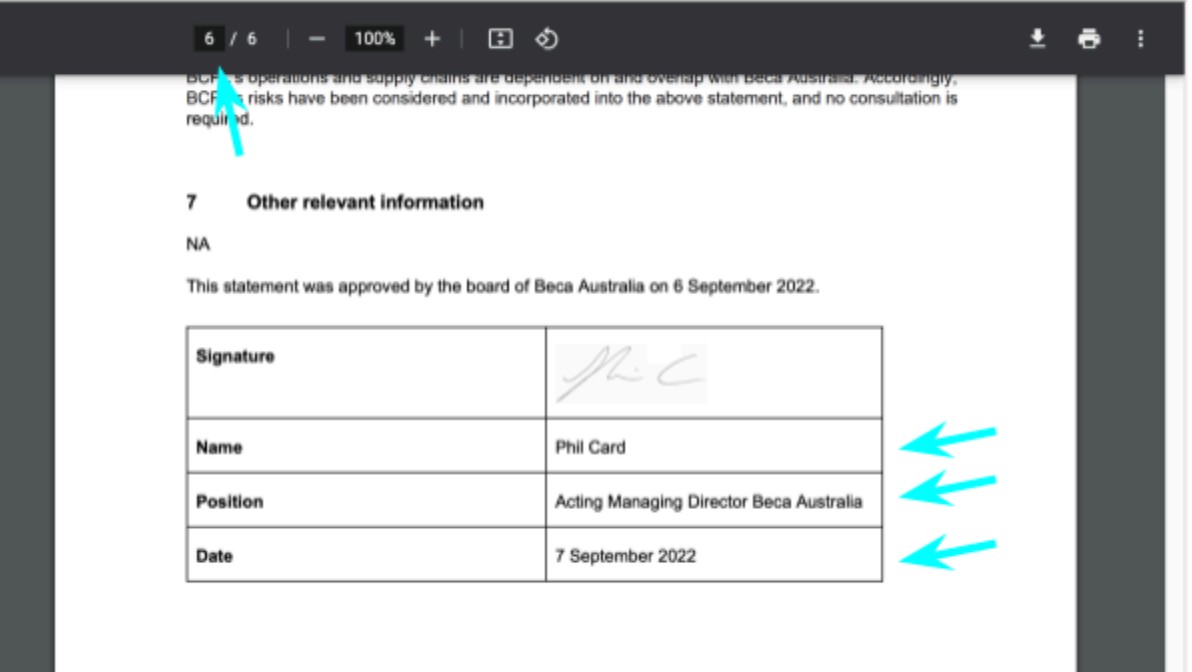

- ■ Select label: **Yes**
- ■ Select Supplementary Button: "Visual Element Category" and select "Table".
- ■ Copy-paste the relevant supporting text from the table in its associated box: *Signature Name Phil Card Position Acting Managing Director Beca Australia Date 7 September 2022*
- ■ Fill the "Date" box: *7 September 2022*
- ■ Select the Supplementary Button: "PDF page number" and add the page number: 6
- ■ Select the Supplementary Button: "Signature Image"

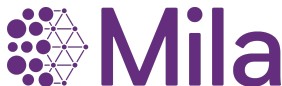
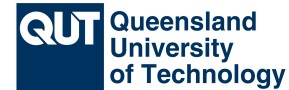

## 2.3. Mandatory Criterion 1: Identify the reporting entity

**Question:** does the statement clearly identify which entities covered by the statement are the relevant reporting entities?

**Background:** entities may identify who is required to report in a number of ways, but they must ensure they clearly identify the reporting entities somewhere in the statement.

To justify a label of "yes, this criterion is satisfied", prioritize the information according to the following **cases (1-3),** where only one case need apply:

1. Extract all sentences from the main text (including its visual elements) that explicitly state who are the reporting entities:
    a. Normal statement: when the text explicitly names a "**reporting entity**" (or similar language such as "required to report" or "required to submit a modern slavery statement").
    b. Joint statement: when the text explicitly names several "**reporting entities**" (or similar language such as "required to report" or "required to submit a modern slavery statement").
2. If the information above is not explicitly stated, look for an introductory sentence that clearly presents the statement issuers with entity names in the main text (including its visual elements).
3. In rare cases where none of the above information is provided, if an entity name or logo are found on a title page or in a significant location in the statement, then they should be extracted only once.
    a. Note: A logo should only be flagged with the "Infographic" category in the "Visual Element Category" button if it is the only way in which the entity is identified in the statement.

**Notes**:

- Evidence for this criterion is most commonly found on the first page of the statement but can be found anywhere in the statement.
- If there is relevant company information duplicated across multiple page footers or headers in the statement, this information should only be extracted once.
- There could be multiple reporting entities for one statement. In such cases, all of them should be extracted in the supporting text. See Example 2.3.a below.

**Label options & their associated supporting text passages extracted from the statements:**

- ○ Yes:
  - ■ Copy-Paste the exact text passage or text passages from the statement that justifies that the reporting entity is identified in the statement
  - ■ If any relevant information comes in other formats than text, please select: Supplementary Button: "Visual Element Category".
- ○ No:
  - ■ Copy-Paste the exact text passage or text passages from the statement that justifies that the reporting entity does not meet this criterion (could be extremely unlikely to say, "we do not want to disclose who we are").
  - ■ If no information is found about this label, please select the Supplementary Button: "No relevant information found in the statement".
- ○ Unclear:
  - ■ Please select this label if the information found is unclear or there are other concerns. If you decide to select this label, you have to choose the Compulsory Supplementary Button "Explanation" and justify why.

---

**Annotation Workflow:**
**The labels and their supplementary buttons should be:**
- ● Yes
  - ■ Copy-Paste supporting text passage
  - ■ Supplementary Button: "Visual Element Category"
- ● No
  - ■ Copy-Paste supporting text passage
    **OR**
  - ■ Supplementary Button: "No relevant information found in the statement
- ● Unclear
  - ■ Compulsory Supplementary Button "Explanation"

---

**Example 2.3.a** (Swiss Re, SID#8933):

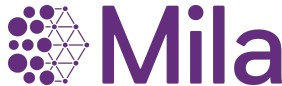
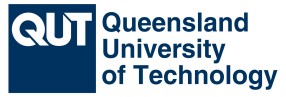

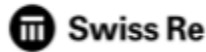

# Swiss Re Modern Slavery Statement

## Introduction

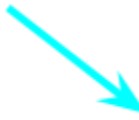

This Modern Slavery Statement (**Statement**) for the financial year ending 31 December 2021 is made pursuant to the *Modern Slavery Act 2018* (Cth) jointly by the following Swiss Re Reporting Entities:

1) Swiss Re Asia Pte. Ltd. (**SRAL**);
2) Swiss Re Australia Ltd (ACN 004 360 909) (**SRAuL**);
3) Swiss Re Life & Health Australia Limited (ACN 000 218 306) (**SRLHA**);
4) Swiss Re International SE (**SRI**); and
5) Swiss Pillar Investments Ltd (**SPIAG**)

(**Swiss Re Reporting Entities, we** or **us**).

Swiss Re Reporting Entities form part of the Swiss Re Group of companies ultimately fully owned by Swiss Re Ltd, a company incorporated in Switzerland and listed on the Swiss Stock Exchange (SIX Swiss Exchange symbol: SREN) (the **Swiss Re Group** or **Swiss Re**).

The Swiss Re Group is a leading wholesale provider of reinsurance, insurance and other insurance-based forms of risk transfer in both the Property and Casualty and Life and Health sectors.

Swiss Re Reporting Entities recognise that 'modern slavery' is used to describe a situation where coercion, threats or deception are used to exploit victims and undermine or deprive them of their freedom. It is used to describe serious exploitation such as human trafficking, servitude, forced marriage, forced labour, debt bondage, deceptive recruiting for labour or services and child labour.

This Statement considers the risk of modern slavery practices in the operations and supply chain of the Swiss Re Reporting Entities and the relevant governance framework.

## Structure, Operations and Supply Chains

### Structure and Operation of Swiss Re Reporting Entities

**SRAL** is a company established in Singapore. SRAL is directly wholly owned by Swiss Re Asia Holding Pte. Ltd. (**SRAHL**), which in turn is ultimately wholly owned by Swiss Re Ltd. SRAL is the entity within Swiss Re that writes Property and Casualty and Life & Health Reinsurance in Asia. In addition to operating in its country of establishment in Singapore, it operates via branches in Australia, Korea, Malaysia, Japan and Hong Kong respectively. As at 31 December 2021, SRAL employed 933 employees in the jurisdictions of its operation described in the foregoing, comprising professional executives, managers and administrative officers.

**SRAuL** is a company incorporated in Australia, wholly owned by SRAHL. SRAuL is the holding company of SRLHA and serves as the Agent in Australia for SRAL and SRI pursuant to s118 of the *Insurance Act 1973* (Cth). SRAuL does not have any employees or operate any business.

**SRLHA** is a company incorporated in Australia, wholly owned by SRAuL. SRLHA also operates in New Zealand via a branch office. SRLHA is a registered life insurance company under

1

- ■ Select label: **Yes**
- ■ Add the supporting text: *This Modern Slavery Statement (Statement) for the financial year ending 31 December 2021 is made pursuant to the Modern Slavery Act 2018 (Cth) jointly by the following Swiss Re Reporting Entities:*

  *1) Swiss Re Asia Pte. Ltd. (SRAL);*

  *2) Swiss Re Australia Ltd (ACN 004 360 909) (SRAuL);*

  *3) Swiss Re Life & Health Australia Limited (ACN 000 218 306) (SRLHA);*

  *4) Swiss Re International SE (SRI); and*

  *5) Swiss Pillar Investments Ltd (SPIAG)*

  *(Swiss Re Reporting Entities, we or us).*

**Example 2.3.b** (Vermilion, [SID#10045](#)):

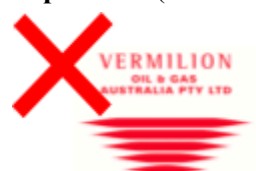

Supporting text that would have been chosen if none of the other supporting texts highlighted in the next page were present (case 3)

Level 5
30 The Esplanade
Perth
Western Australia

Tel:     08 9215 0300
ABN:  29 113 023 591

**Vermilion Oil and Gas Australia Pty Ltd**

**2020 Modern Slavery Statement**

**Message from the Managing Director of Vermilion Oil and Gas Australia:**
We are committed to playing our part in the global fight to end modern slavery in all its forms. As a responsible energy producer, we have always focused on three priorities: the safety of our operations and our communities; environmental stewardship; and profitability – in that order. Nothing is more important than safety. This is directly linked to our efforts to identify and mitigate modern slavery risks in our own operations, and in our supply chain, as it relates to the responsibility of all of us to support the safety and wellbeing of everyone in our wider communities.

We are proud of our efforts, as described in our first Modern Slavery Statement, and committed to growing them in the coming years. We also welcome your feedback, and invite you to contact us with any questions or comments at vogamscompliance@vermilionenergy.com

[2 pages later]

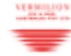

**Overview**

This is Vermilion Australia's inaugural Modern Slavery Statement ('the **Statement**') for the year ending 31 December 2020 ('the **reporting period**').

The purpose of this Statement is to provide a comprehensive overview of Vermilion Australia's commitments and efforts to address potential modern slavery risks in our operations and supply chain.

Vermilion Oil and Gas Australia Pty Ltd or Vermilion Australia (ABN 29 113 023 591) is a wholly owned subsidiary of Vermilion Energy Inc., an entity headquartered in Canada, with global operations, spanning a number of geographies, including North America and Europe. Further information relating to our corporate structure is outlined below.

The Statement, which was prepared in accordance with the requirements of the Australian Modern Slavery Act 2018 ('the **Act**'), provides details relating to our structure, operations and supply chain and offers an overview of any associated modern slavery risks. The Statement was approved by the principal governing body of Vermilion Australia as defined by the Modern Slavery Act 2018 on 1st July 2022.

The Statement further expands on some of the key mechanisms adopted and actions taken to assess and address modern slavery risks. Please refer to section four of this Statement for additional information ('Assessing and addressing modern slavery risks').

**Our structure, operations & supply chain**

**Our Structure**

Vermilion Australia is a wholly owned subsidiary of Vermilion Energy Inc. located in Calgary, Canada. This is Vermilion Energy Inc.'s only Australian subsidiary.

Beyond Canada and Australia, Vermilion Energy Inc. also has operations in the United States, France, the Netherlands, Germany, Ireland, Croatia, Hungary and Slovakia.

Vermilion Australia is a reporting entity under the Act and therefore the preparer of this Statement. Vermilion Australia has also established a wholly owned subsidiary, Vermilion Energy Australia Pty Ltd (ACN 623624395). Vermilion Energy Australia Pty Ltd is not an operating entity and does not have any employees or internal business operations.

> Supporting text that would have been chosen had the text below not been present (case 2)

> Supporting text that should be chosen because it explicitly names the reporting entity (case 1)

- ■ Select label: **Yes**
- ■ Add the supporting text: *Vermilion Australia is a reporting entity under the Act and therefore the preparer of this Statement.*
- ■ Note: this example can be used to explain the prioritization strategy used to find the supporting text for a "yes" label:
  - a. We start by searching for text that clearly states which entities are reporting entities (using explicit terms such as "reporting entity", "reporting entities", "X is required to report…"), according to **case 1**. This statement contains the sentence: *Vermilion Australia is a reporting entity under the Act and therefore the preparer of this Statement.*

If other text passages explicitly name a reporting entity, they should also be extracted. Once this is done, there is no need to extract text passages according to the other cases.

b. Had an explicit text passage as mentioned above not been present, we should have extracted the following sentence according to **case 2**: *This is Vermillion Australia's inaugural Modern Slavery Statement ('the **Statement**') for the year ending 31 December 2020 ('the **reporting period**').*
This sentence corresponds to an introductory sentence that clearly identifies the issuer of the statement. Once this is done, there is no need to extract text passages according to the last case.

c. Ultimately, had no other information been present regarding the reporting entity, we should have extracted the following sentence according to **case 3**: *Vermilion Oil and Gas Australia Pty Ltd*
This sentence corresponds to a company name and is located on a page that corresponds to a title page. Note that here the company name should be extracted, and not the logo (as a visual element). The logo should only be extracted as a last resort.

**Example 2.3.c** (Reece Group, [SID#2503](#)):

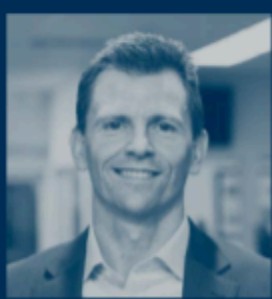

## OUR COMMITMENT

At Reece, our purpose guides our decisions and actions, big and small, and our values ground us. Together, they make up The Reece Way.

We're focused on improving the lives of our customers and our people by striving for greatness every day.

Working together, we can create thriving and sustainable communities. This is only possible if we lead by example and respect and uphold the human rights of our people, customers, and broader network across our supply chain.

We oppose all forms of slavery and forced labour in our operations and the operations of our suppliers.

We have always maintained a rigorous program to ensure supplier compliance with our expectations, and our culture of continuous improvement drives our commitment to building ethical and sustainable supply chains and supplier partnerships.

Our first modern slavery statement shares our journey of identifying and understanding, as well as our initial actions taken to address modern slavery risks. We are committed to working collaboratively across the industry to address this important issue.

This statement has been made in accordance with the Australian Modern Slavery Act (Commonwealth) 2018. This statement is made by Reece Limited on behalf of Reece Australia Pty Ltd, Actrol Pty Ltd, Reece New Zealand Pty Ltd, MORSCO Inc and Todd Pipe Holdings Inc. for the year ended 30 June 2020 and has been approved by the Board of Directors of Reece Limited.

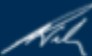

**Peter Wilson**
Reece Group Chief Executive Officer and Managing Director

29 March 2021

4    Reece Group

- ■ Select label: **Yes**
- ■ Add the supporting text: *This statement is made by Reece Limited on behalf of Reece Australia Pty Ltd, Actrol Pty Ltd, Reece New Zealand Pty Ltd, MORSCO Inc and Todd*

*Pipe Holdings Inc. for the year ended 30 June 2020 and has been approved by the Board of Directors of Reece Limited.*

**Example 2.3.d** (Mineral Resources Limited, SID#1674):

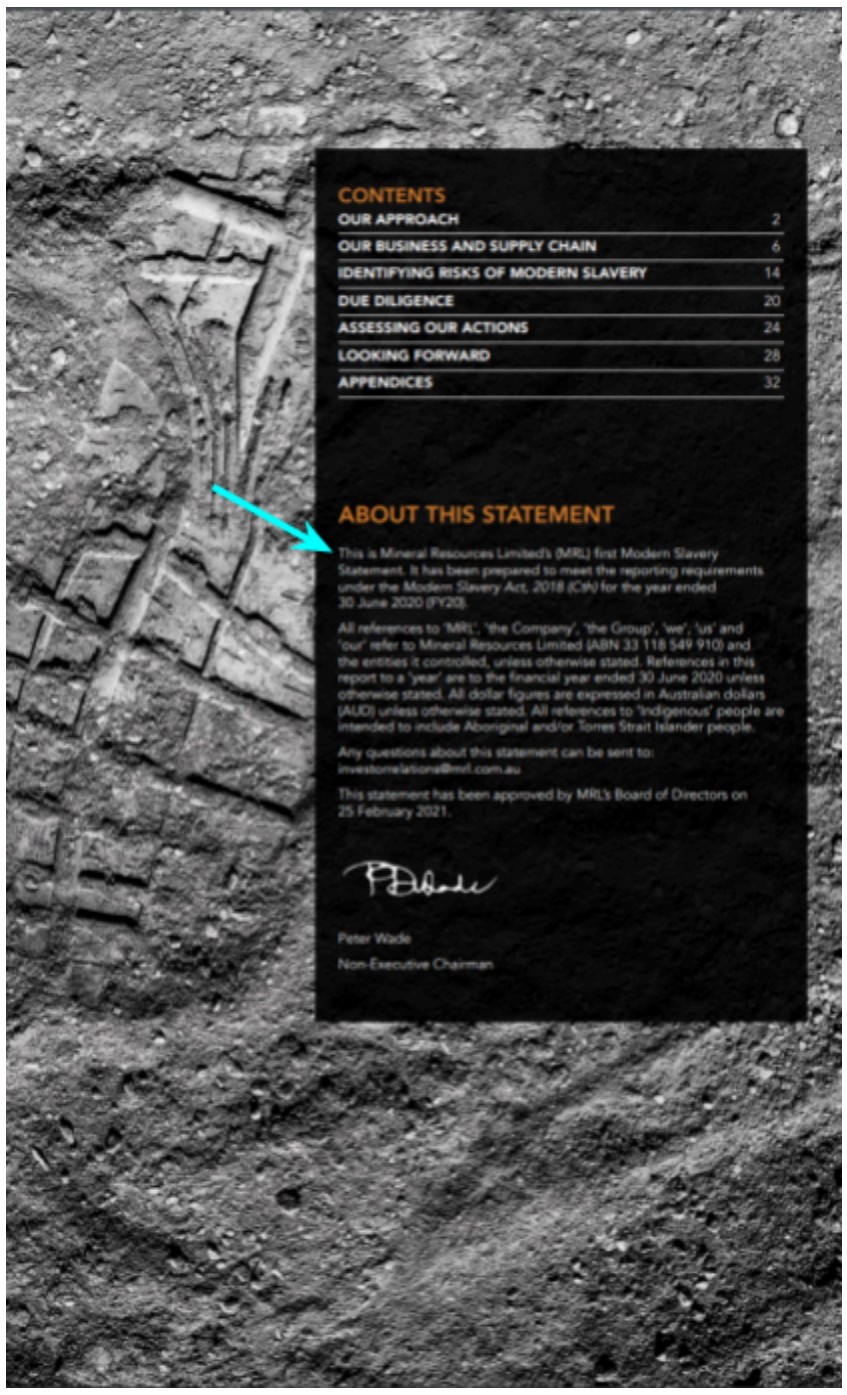

- Select label: **Yes**
- Add the supporting text: *This is Mineral Resources Limited's (MRL) first Modern Slavery Statement.*

**Example 2.3.e** (Cerner, SID#5508):

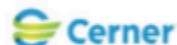

### STATEMENT AGAINST MODERN SLAVERY

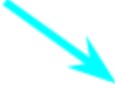

This statement sets out the steps Cerner Corporation and its controlled subsidiaries – including Cerner Limited and Cerner Corporation Pty Limited (ABN 23 051 063 435) (collectively, "Cerner", "we", "our" or "us") – have taken to ensure that slavery, human trafficking, child labor and forced labor are not taking place in its business or its supply chains. As part of our overall commitment to supporting human rights around the world, we oppose and prohibit all forms of modern slavery, and we expect the same from our suppliers and business associates.

We are issuing this statement pursuant to the UK Modern Slavery Act and the Australia Modern Slavery Act, which require Cerner and certain of our subsidiaries to disclose efforts to ensure that modern slavery is not taking place in our supply chains and business operations. This statement covers our 2020 financial year (January 1, 2020 – December 31, 2020).

**Business and Organization**

Cerner is a leading supplier of health care information technology ("HCIT") solutions and tech-enabled services. Our mission is to relentlessly seek breakthrough innovation that will shape health care of tomorrow. We offer a wide range of intelligent solutions and tech-enabled services that support the clinical, financial and operational needs of organizations of all sizes. Cerner® solutions and services help clinicians make care decisions and assist organizations in managing the health of their populations. Cerner also offers integrated clinical and financial systems to help manage day-to-day revenue functions, as well as a wide range of services to support clinical, financial and operational needs.

Cerner is headquartered in the United States and operates globally, primarily in Australia, Canada, Europe, India and the Middle East. Cerner Corporation started doing business in 1980, is a public company which trades on the Nasdaq stock exchange under the symbol "CERN", and is headquartered in Kansas City, Missouri. Cerner Corporation Pty Limited (ABN 23 051 063 435) and Cerner Limited are wholly owned, indirect subsidiaries of Cerner Corporation. Cerner Corporation Pty Limited has its registered office and principal place of business at Level 13 & 14, 1 Pacific Highway, North Sydney, NSW 2060, Australia, and Cerner Limited has its registered office and principal place of business at 6th Floor, The Point, 37 North Wharf Road, London W2 1AF.

Cerner employed approximately 26,400 associates worldwide as of December 31, 2020. Of that total population, approximately 74% of our associates were employed in the United States and the remaining associates were employed outside the United States. As reported in Cerner's annual report on Form 10-K for the year ended December 31, 2020, approximately 35% of our associates

- Select label: **Yes**
- Add the supporting text: *This statement sets out the steps Cerner Corporation and its controlled subsidiaries – including Cerner Limited and Cerner Corporation Pty Limited (ABN 23 051 063 435) (collectively, "Cerner", "we", "our" or "us") – have taken to ensure that slavery, human trafficking, child labor and forced labor are not taking place in its business or its supply chains.*

**Example 2.3.f** (McGrath, SID#8609):

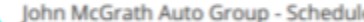

### Introduction

The McGrath Auto Group, opposes modern slavery in all forms. We are committed to the highest standards of ethical behavior in our business activities and to promoting a culture of honesty, corporate compliance, and good corporate governance.
This statement is made in accordance with the Australian Modern Slavery ACT 2018.

### McGrath Auto Group Business Structure

### John McGrath Auto Group - Schedule

J & V Holdings Pty Ltd        ACN 81 122 411 778   is the controlling entity of the trading entities listed below, collectively called "the Group".

McGrath Canberra Pty Ltd      ABN 78 093 024 107
John McGrath Holden Pty Ltd   ABN 47 628 781 582
Gusherd Pty Ltd               ABN 16 093 033 197

Trading As
McGrath Auto Group, John McGrath Auto Group, Belconnen MG, Phillip MG, John McGrath GMSV, John McGrath Holden, John McGrath HSV, Prestige Porformance Centre, Hyundai Commercial Vehicles Queanbeyan, John McGrath LDV, John McGrath Maserati, John McGrath MG, John McGrath Prestige, John McGrath Fleet and Demonstrator Clearance Centre, Belconnen Budget Cars, John McGrath Ssangyong, John McGrath Kia, Mercedes-Benz Canberra, McGrath Budget Cars, Phillip Budget Cars, Phillip Used Cars, John McGrath Isuzu Ute, John McGrath Ford, John McGrath Auto Group - Phillip.

### John McGrath Auto Group - Operations and supply chains

The Group is composed of 5 Automotive Dealerships with over 200 staff in various locations in Canberra - ACT and Queanbeyan - NSW.
The dealerships retail vehicles, supply spare parts and provide servicing requirements.

Our major suppliers are international vehicle manufacturers or importers, including, Ford, LDV, Peugeot, Mercedes-Benz, Citroen, Isuzu Ute, MG, Maserati, Kia.

We also identify the following non-manufacturer services within our supply chain:
* Cleaning Services
* Office and Print Supplies
* Advertising Services
* Provision of financial services related to automotive supply.

### Identify the risks of modern slavery practices in the operations and supply chains.

These non-manufacturer suppliers comprise a very small proportion of our supply chain, but we consider them to have potential breaches of the modern slavery obligations.

The Vehicle manufacturers are the largest suppliers our organisation trades with. We acknowledge that risks of modern slavery within this group could exist. We acknowledge that we lack visibility in overseas markets and this carries additional risks especially in secondary levels of suppliers to the manufacturers.

- Select label: **Unclear**
- Note: based on the metadata on the Australian MSA registry website, this is a joint statement. However, the text doesn't clearly list multiple reporting entities. Looking at:

    *J & V Holdings Pty Ltd ACN 81 122 411 778 is the controlling entity of the trading entities listed below, collectively called "the Group".*
    *McGrath Canberra Pty Ltd ABN 78 093 024 107*
    *John McGrath Holden Pty Ltd ABN 47 628 781 582*
    *Gusherd Pty Ltd ABN 16 093 033 197*

We see several entities listed but the last three are defined as controlled entities and no information is provided to indicate whether they are also reporting entities. As a result, this statement should be labeled as "Unclear".

**Example 2.3.g** (Solo Group, [SID#7997](SID#7997)):

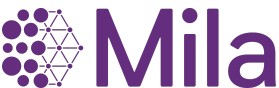
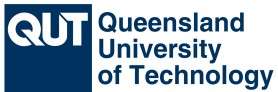

MODERN SLAVERY STATEMENT

**Solo**

Solo Resource Recovery | Industrial Services | Richmond Waste | Solo Water | RRS | PHA

## 1.0 INTRODUCTION

This Modern Slavery Statement is intended to comply with the requirements of the Modern Slavery Act 2018 - *(Cth) Modern Slavery Act*.

This statement is submitted as a joint statement for the Solo Group of companies reporting entities as defined in section 5 of the same Act.

## 2.0 BUSINESS ACTIVITIES

The Solo Group of companies' is an Australian owned and operated family business, which has operated since 1932. Established locally in 1990, Solo Resource Recovery offers a range of waste and resource recovery services to industry, commerce and homes throughout New South Wales, South Australia, Victoria, Western Australia and Queensland. These services include waste collection services, domestic, recycling and organics, liquid waste, waste water and sludge removal and treatment, industrial bins, medical waste transfer and industrial services, including underground asset locations and public utility planning as well as other customised waste management solutions.

The Group currently employs over 700 people nationally.

The size, nature, diversity and geographical locations of operations conducted by the Solo Group opens to risk as identified in the *Australian Modern Slavery Act 2018*.

The below reporting entities are under the control of the same integrated management system operating under the same policy directives from the same board of directors.

Reporting entities are listed in the following table.

| |
|---|
| **Solo Resource Recovery** |
| **Solo Waste** |
| **Solo Water** |
| **Rico Recovery Systems** |
| **Richmond Waste Services** |
| **Pellet Heaters Australia** |

## 3.0 CONSULTATION WITH RELATED ENTITIES

A consultation process has been undertaken with all relevant current operational managers and training will be continually developed to ensure that people accountable are taking the appropriate steps to ensure compliance to the requirements of this statement and the associated policy directive.

The Company's goals, approaches and commitments towards ensuring that we work continually towards identifying and managing these risks are supported within our *Modern Slavery Response Policy Statement*.

Issue No: **1.0**    Effective Date: 02/2021
Uncontrolled when printed

Page | 3

- ■ Select label: **Yes**
- ■ Add the supporting text: *This statement is submitted as a joint statement for the Solo Group of companies reporting entities as defined in section 5 of the same Act.* // *Reporting entities are listed in the following table.*
- ■ Use the "Table" category in the "Visual Element Category" button, and add the following text in the corresponding text box: *Solo Resource Recovery*
  *Solo Waste*
  *Solo Water*
  *Rico Recovery Systems*
  *Richmond Waste Services*
  *Pellet Heaters Australia*
- ■ Select the Supplementary Button: "PDF page number" and add the page number: 3

## 2.4. Mandatory Criterion 2: Structure, operations and supply chains

**Question:** does the statement clearly describe the structure, operations, and supply chains of the reporting entity?

**Background:** the second mandatory criterion requires reporting entities to describe their structure, operations, and supply chains. **We will separately look at each of these in Sections 2.4.1 to 2.4.3.**

**Note:** this criterion is highly intertwined with other mandatory criteria. For instance, the information related to an entity's structure can sometimes be found around the same passages that describe their operations and supply chains. Information about the structure, operations, or supply chains of an entity may also overlap in individual sentences.

## 2.4.1. Structure

**Question:** does the reporting entity describe its structure?

**Background:** the structure is the **legal and organizational form** of the reporting entity. A reporting entity can describe its structure in multiple ways. For example, it can:

- **Explain its general structure.** This covers the entity's legal classification (company, partnership, trust, etc.). If the entity is part of a larger group, the statement should provide details on the general structure of the overall group (upstream and downstream from the reporting entity). The statement may also describe the approximate number of workers employed by the entity and the registered office location of the entity (i.e., its statutory address).
- **Explain what the entity owns or controls.** If the entity does own or control other entities, explain what these entities do and where they are located. For example, the entity may own a foreign subsidiary in another country.
- **Identify any trading names or brand names** associated with the reporting entity and entities it owns or controls.

**Note:** the presence of any relevant information should lead to a "Yes" label for this requirement.

**Label options and expected supporting elements for this question:**

- **Yes**, the statement describes the reporting entity's structure:
    - Copy-paste the text passages from the statement that justify that the reporting entity described its structure.
    - If any relevant information comes in other formats than text, fill in the required information via the Supplementary Button: "Visual Element Category", note the page where it is found, and extract any relevant text it contains (if possible).
- **No**, the statement does not describe the reporting entity's structure:

- Copy-paste the exact text passages from the statement that justify that the entity does not meet this criterion,

  **OR**

- If no information is found about this label, select the Supplementary Button: "No relevant information found in the statement".

- **Unclear**, in any other case:
  - Select this label if the information found is unclear or there are other concerns. If you decide to select this label, you have to choose the Compulsory Supplementary Button "Explanation" and justify why.

**Example 2.4.1.a** (Macquarie, SID#7944)**:**

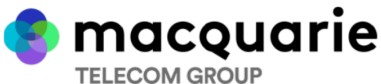

## Macquarie Telecom Group: Modern Slavery Statement

Macquarie Telecom Group is committed to high standards of conduct and ethical behaviour in its business activities and rejects all forms of modern slavery.

### About Macquarie Telecom

Macquarie Telecom Group Limited is Australia's datacentre, cloud, cyber security and telecom company for mid to large business and government customers. We have offices in Sydney, Melbourne, Canberra, Brisbane and Perth. We own and operate five data centres in Sydney and Canberra. We are listed on the ASX (MAQ), and further details of our operations, structure and business segments are available in our 2021 Annual Report which can be found on the investor section of the Macquarie Telecom Group website: https://macquarietelecomgroup.com/investors/

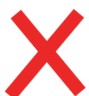

This Modern Slavery Statement covers Macquarie Telecom Group Limited (ABN 72 056 712 228) and all its subsidiaries. Details of our subsidiaries can be found on page 77 of our Annual Report.

### Macquarie Telecom's Policies relating to Modern Slavery

Modern slavery describes situations where victims are being deprived of their freedom through threats, coercion or deception.

Modern slavery includes the following types of exploitation: people trafficking; slavery; servitude; forced marriage; forced labour; debt bondage; child labour; and deceptive recruiting for labour or services.

- Select label: **No**

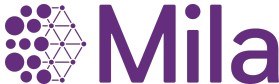

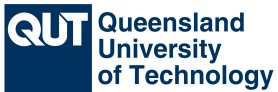

- Note that references to websites or external documents cannot be used as supporting text for 2.4.1. This means that in this case, the company did not adequately describe its structure at all, and therefore the label is **No**.

**Example 2.4.1.b** (Cerner, SID#5508)**:**

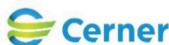

### STATEMENT AGAINST MODERN SLAVERY

This statement sets out the steps Cerner Corporation and its controlled subsidiaries – including Cerner Limited and Cerner Corporation Pty Limited (ABN 23 051 063 435) (collectively, "Cerner", "we", "our" or "us") – have taken to ensure that slavery, human trafficking, child labor and forced labor are not taking place in its business or its supply chains. As part of our overall commitment to supporting human rights around the world, we oppose and prohibit all forms of modern slavery, and we expect the same from our suppliers and business associates.

We are issuing this statement pursuant to the UK Modern Slavery Act and the Australia Modern Slavery Act, which require Cerner and certain of our subsidiaries to disclose efforts to ensure that modern slavery is not taking place in our supply chains and business operations. This statement covers our 2020 financial year (January 1, 2020 – December 31, 2020).

**Business and Organization**

Cerner is a leading supplier of health care information technology ("HCIT") solutions and tech-enabled services. Our mission is to relentlessly seek breakthrough innovation that will shape health care of tomorrow. We offer a wide range of intelligent solutions and tech-enabled services that support the clinical, financial and operational needs of organizations of all sizes. Cerner® solutions and services help clinicians make care decisions and assist organizations in managing the health of their populations. Cerner also offers integrated clinical and financial systems to help manage day-to-day revenue functions, as well as a wide range of services to support clinical, financial and operational needs.

Cerner is headquartered in the United States and operates globally, primarily in Australia, Canada, Europe, India and the Middle East. Cerner Corporation started doing business in 1980, is a public company which trades on the Nasdaq stock exchange under the symbol "CERN", and is headquartered in Kansas City, Missouri. Cerner Corporation Pty Limited (ABN 23 051 063 435) and Cerner Limited are wholly owned, indirect subsidiaries of Cerner Corporation. Cerner Corporation Pty Limited has its registered office and principal place of business at Level 13 & 14, 1 Pacific Highway, North Sydney, NSW 2060, Australia, and Cerner Limited has its registered office and principal place of business at 6th Floor, The Point, 37 North Wharf Road, London W2 1AF.

Cerner employed approximately 26,400 associates worldwide as of December 31, 2020. Of that total population, approximately 74% of our associates were employed in the United States and the remaining associates were employed outside the United States. As reported in Cerner's annual report on Form 10-K for the year ended December 31, 2020, approximately 35% of our associates

- Select label: **Yes**
- Add the supporting text: *This statement sets out the steps Cerner Corporation and its controlled subsidiaries – including Cerner Limited and Cerner Corporation Pty Limited (ABN 23 051 063 435) (collectively, "Cerner", "we", "our" or "us") – have taken to ensure that slavery, human trafficking, child labor and forced labor are not taking place in its business or its supply chains.* // *Cerner is headquartered in the United States and operates globally, primarily in Australia, Canada, Europe, India and the Middle East. Cerner Corporation started doing business in 1980, is a public company which trades on the Nasdaq stock exchange under the symbol "CERN", and is headquartered in Kansas City, Missouri. Cerner Corporation Pty Limited (ABN 23 051 063 435) and Cerner Limited are wholly owned, indirect subsidiaries of Cerner Corporation. Cerner Corporation Pty Limited has its registered office and principal place of business at Level 13 & 14, 1 Pacific Highway, North Sydney, NSW 2060, Australia, and Cerner Limited has its registered office and principal place of business at 6th Floor, The Point, 37 North Wharf Road, London W2 1AF. Cerner employed approximately 26,400 associates worldwide as of December 31, 2020.*
- **Note:** the sentence starting with "*Cerner is headquartered...*" is added here because it describes the overall group (and the parent headquarters) in the USA.

**Example 2.4.1.c** (Coliban Water, SID#7101)**:**

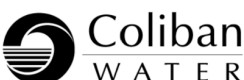

# Modern Slavery Statement FY21

MODERN SLAVERY ACT 2018
COLIBAN WATER 1 JULY 2020 TO 30 JUNE 2021

**Coliban Water's Introduction**
The Modern Slavery Statement is provided as a statement under section 14(1) of the Modern Slavery Act 2018 (Act) for the period from 1 July 2020 to 30 June 2021 (statement) for Coliban Water ACN 149 633 116.

Coliban Region Water Authority was established on 1 July 1992 under the Water Act 1989 (Vic) (the Act) as a Regional Urban Water Authority. The Authority became the Coliban Region Water Corporation on 1 July 2007 and operates as Coliban Water. Our shareholder is the Victoria Government. Our responsible Minister for the period of 1 July 2020 until 30 June 2021 was The Hon. Lisa Neville MP, Minister for Water.

**Coliban Water Modern Slavery Statement**
Our purpose is to deliver, water services for community needs now and into the future. We provide urban water and wastewater services in accordance with the provisions of Part 8- Water supply and Part 9- Sewerage of the Water Act 1989 (Vic). Rural customers are supplied under Part 4 Division 2 Section 51 of the Act.

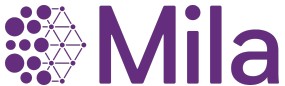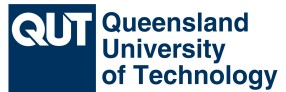

- Select label: **Yes**
- Add the supporting text: *Coliban Region Water Authority was established on 1 July 1992 under the Water Act 1989 (Vic) (the Act) as a Regional Urban Water Authority. The Authority became the Coliban Region Water Corporation on 1 July 2007 and operates as Coliban Water.*

**Example 2.4.1.d** (Queensland Rail, SID#2314)**:**

---

# Table of contents

# A message from our Chair and CEO

Queensland Rail and its subsidiary, Queensland Rail Limited (QRL), collectively known as Queensland Rail, is committed to undertaking business with integrity, probity and accountability and expects its suppliers to commit to the same standards.

Queensland Rail is focused on ensuring modern slavery does not occur in our operations and supply chains and are committed to complying with our obligations under the *Modern Slavery Act 2018 (Modern Slavery Act)*. This includes publishing our Modern Slavery Statement.

Modern slavery describes situations where offenders use coercion, threats or deception to exploit victims and undermine their freedom. It encompasses all forms of contemporary slavery including trafficking in persons, slavery, servitude, forced marriage, forced labour, debt bondage, deceptive recruiting for labour or services and child labour.

Our first Modern Slavery Statement is a joint statement for Queensland Rail (ABN 68 598 268 528) and QRL (ABN 71 132 181 090). It aligns to the seven mandatory criteria in the Modern Slavery Act and outlines the steps we have taken to identify, manage and mitigate the specific risks of modern slavery in our operations and supply chain in financial year from 1 July 2019 to 30 June 2020 (FY2019-20).

[several pages later]

# Company Overview

Queensland Rail's purpose is to provide a customer-focused, safe, reliable, on-time, value for money rail service that benefits the community, supports industry and is integrated with the public transport system.

Founded in 1865, Queensland Rail has a proud history of connecting Queensland communities and supporting local jobs, industries and economies. The organisation's vision is connecting communities through a modern, world-class rail service.

**Structure**

Queensland Rail is a statutory authority established by the Queensland Government under the Queensland Rail Transit Authority Act. Queensland Rail discharges its statutory functions through its wholly owned subsidiary QRL. QRL does not employ any personnel but owns all non-employee related assets and contracts. It performs the role of rail transport operator under the Rail Safety National Law (Queensland).

Queensland Rail's statutory functions include: management of railways; provision of rail transport services, including passenger services; and construction and maintenance of railway infrastructure.

Queensland Rail is structured around two lines of business focused on our differing customer needs in South East and Regional Queensland. This modern slavery statement also covers subsidiaries of QRL.

**Operations**

Queensland Rail employs over 7,500 people and delivers services across its South East Queensland and Regional networks. Queensland Rail's network extends more than 6,600 kilometres across the state and the business operates the following three core services across multiple customer markets: Citytrain, Travel and Tourism, and Regional Network and Freight.

The South East Queensland team provides 8,322 services each week, supporting our customers to reach their destination safely and on-time. In 2019-20, 44 million passenger trips were taken on the South East Queensland network (a decrease of approximately 20 per cent compared to 2018-2019, associated with the COVID-19 pandemic).

Our regional commuter and rail tourism markets are serviced by the Travel and Tourism services. These comprise of eight services connecting regional communities across Queensland with other regional centres and South East Queensland centres. Queensland Rail also supports the Queensland tourism industry through the provision of unique rail tourism experiences including special events and charters through the operation of a small fleet of heritage services.

[next page]

and standards which underpins our ethics and values and guides the way we treat and engage with our employees, customers, suppliers, business partners and the communities in which we operate.

**Modern Slavery disclosure**

In FY2019-20, Queensland Rail did not identify any instances of modern slavery in its operations or supply chain.

# Understanding our risks

**Our operations**

As at 30 June 2020, Queensland Rail had 7,537 employees of which 93 per cent were employed on a permanent basis, five per cent on a temporary basis and two per cent as casual. Of those employees employed on a permanent or fixed term basis, 95 per cent are covered by formal documented agreements ratified by the Fair Work Commission of which approximately 300 employees (Enhanced Remuneration Package and Award Contract) are paid above the applicable Queensland Rail enterprise agreement, while still having their employment linked to the enterprise agreement. Queensland Rail also employs 351 contract employees (those employed on an on-going common law individual employment contract).

- Select label: **Yes**
- Add the supporting text: *Queensland Rail and its subsidiary, Queensland Rail Limited (QRL), collectively known as Queensland Rail, is committed to undertaking business with integrity, probity and accountability and expects its suppliers to commit to the same standards.* // *Queensland Rail is a statutory authority established by the Queensland Government under the Queensland Rail Transit Authority Act. Queensland Rail discharges its statutory functions through its wholly owned subsidiary QRL. QRL does not employ any personnel but owns all non-employee related assets and contracts.* // *Queensland Rail employs over 7,500 people and delivers services across its South East Queensland and Regional networks.* // *As at 30 June 2020, Queensland Rail had 7,537 employees of which 93 per cent were employed on a permanent basis, five per cent on a temporary basis and two per cent as casual.*
- Note: the first sentence ("*Queensland Rail and its subsidiary…*") is extracted only because it provides information on the subsidiary and on their group name, and not because of the commitments.

## 2.4.2. Operations

**Question:** does the reporting entity describe its operations?

**Background:** the operations cover the activities undertaken by the reporting entity or any of its owned or controlled entities to pursue its business objectives and strategy in Australia or overseas. An entity can describe its operations by:

- Explaining the nature and types of activities undertaken by the entity (for example, mining, retail, manufacturing) and any entities it owns or controls;
- Identifying the countries or regions where the entity's operations are located or take place; or
- Providing facts and figures about the entity's operations, such as factories, offices, franchises, and/or stores.

Here are examples of the types of activities that a reporting entity might describe:

1. **The direct purpose of employment of their workers.** For example, a manufacturing company may describe that workers operate the company's manufacturing facilities.
2. **Processing, production, research and development, or construction activities.** For example, a dairy food company may describe the processing and production of milk products. A building company may describe the construction of commercial and/or residential buildings. A mining company may describe its ore processing activities as well as activities related to the construction of new facilities to ship ore products. A medical research company may describe the development and trial of new medicines.
3. **Charitable or religious activities.** For example, a large bank may describe the activities of its charitable foundation, such as donations to child welfare projects overseas. An animal protection charity may describe its work to safeguard threatened habitats overseas. A religious entity may describe its provision of religious services, such as marriages.
4. **The purchase, marketing, sale, delivery, or distribution of products or services.** For example, a marketing agency may describe its activities to market the goods of another company. An accounting firm may describe the provision of financial services to clients. A charitable aged care facility may describe the delivery of aged care services. A bank may describe its financial lending services for its clients. A food and beverage company may describe the distribution of its products through a dedicated transport network it owns and operates. A property developer may describe the purchasing of land overseas. A hotel chain might describe how franchises deliver its services to clients across different locations.
5. **Ancillary activities required for the main operations of the entity.** For example, a mining company may provide transportation to remote locations for its employees. A software development company may offer child care services to its employees.
6. **Financial investments**. For example, fund managers may describe internally managed investment portfolios and assets. Investments may also include financial investments in joint

ventures. For example, an oil company may describe investments in a non-operated joint venture for an offshore oil rig.

**Note**: Externally managed portfolios, such as those managed by another fund manager, may be more appropriately considered part of the reporting entity's supply chains (see Section 2.4.3).

7. **Managed/operated joint ventures.** For example, a mining company may describe domestic and overseas joint ventures that it directly manages/operates.

8. **Leasing of property, products, or services.** For example, a ship leasing business may describe its leasing of vessels. A property owner (or a leasing company) may describe the leasing of retail shops, commercial and/or industrial facilities.

**Label options and expected supporting elements for this question:**

- **Yes**, the statement describes the reporting entity's operations:
  - Copy-paste the text passages from the statement that justify that the reporting entity described its operations.
  - If any relevant information comes in other formats than text, fill in the required information via the Supplementary Button: "Visual Element Category", note the page where it is found, and extract any relevant text it contains (if possible).
- **No**, the statement does not describe the reporting entity's operations:
  - Copy-paste the exact text passages from the statement that justify that the entity does not meet this criterion,

    **OR**
  - If no information is found about this label, select the Supplementary Button: "No relevant information found in the statement".
- **Unclear**, in any other case:
  - Select this label if the information found is unclear or there are other concerns. If you decide to select this label, you have to choose the Compulsory Supplementary Button "Explanation" and justify why.

**Example 2.4.2.a** (McGrath, SID#8609)**:**

## Introduction

The McGrath Auto Group, opposes modern slavery in all forms. We are committed to the highest standards of ethical behavior in our business activities and to promoting a culture of honesty, corporate compliance, and good corporate governance.
This statement is made in accordance with the Australian Modern Slavery ACT 2018.

## McGrath Auto Group Business Structure

### John McGrath Auto Group - Schedule

J & V Holdings Pty Ltd      ACN 81 122 411 778   is the controlling entity of the trading entities listed below, collectively called "the Group".

McGrath Canberra Pty Ltd      ABN 78 093 024 107
John McGrath Holden Pty Ltd  ABN 47 628 781 582
Gusherd Pty Ltd              ABN 16 093 033 197

Trading As
McGrath Auto Group, John McGrath Auto Group, Belconnen MG, Phillip MG,
John McGrath GMSV, John McGrath Holden, John McGrath HSV,
Prestige Porformance Centre, Hyundai Commercial Vehicles Queanbeyan,
John McGrath LDV, John McGrath Maserati, John McGrath MG,
John McGrath Prestige, John McGrath Fleet and Demonstrator Clearance Centre,
Belconnen Budget Cars, John McGrath Ssangyong, John McGrath Kia,
Mercedes-Benz Canberra, McGrath Budget Cars, Phillip Budget Cars,
Phillip Used Cars, John McGrath Isuzu Ute, John McGrath Ford,
John McGrath Auto Group - Phillip.

### John McGrath Auto Group - Operations and supply chains

The Group is composed of 5 Automotive Dealerships with over 200 staff in various locations in Canberra - ACT and Queanbeyan - NSW.
The dealerships retail vehicles, supply spare parts and provide servicing requirements.

Our major suppliers are international vehicle manufacturers or importers, including, Ford, LDV, Peugeot, Mercedes-Benz, Citroen, Isuzu Ute, MG, Maserati, Kia.

We also identify the following non-manufacturer services within our supply chain:
* Cleaning Services
* Office and Print Supplies
* Advertising Services
* Provision of financial services related to automotive supply.

## Identify the risks of modern slavery practices in the operations and supply chains.

These non-manufacturer suppliers comprise a very small proportion of our supply chain, but we consider them to have potential breaches of the modern slavery obligations.

The Vehicle manufacturers are the largest suppliers our organisation trades with. We acknowledge that risks of modern slavery within this group could exist. We acknowledge that we lack visibility in overseas markets and this carries additional risks especially in secondary levels of suppliers to the manufacturers.

- ■ Select label: **Yes**
- ■ Add the supporting text: *The Group is composed of 5 Automotive Dealerships with over 200 staff in various locations in Canberra - ACT and Queanbeyan - NSW. The dealerships retail vehicles, supply spare parts and provide servicing requirements.*
- ■ **Note:** The sentence starting with "*The Group is composed...*" would also be relevant for structure (2.4.1) because it provides an approximate number of workers, but it is also relevant here for operations because it mentions the number of dealerships and locations.

**Example 2.4.2.b** (Reece Group, SID#2503):

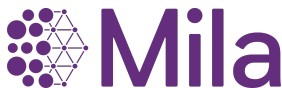
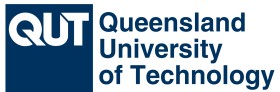

# WE ARE REECE

Reece Group is a leading distributor of plumbing, waterworks, and HVAC-R to more than 200,000 residential, commercial and infrastructure customers across Australia, New Zealand, and the United States.

In 2018, Reece Group acquired MORSCO to create a platform for future growth, and bring the same customised service to the growing Sun Belt region of the US.

In 2019, we built on these foundations with the acquisition of Todd Pipe in Southern California.

We support residential, commercial and infrastructure building trades, and our business is diverse. We are a wholesaler, retailer, and supply chain logistics company, and we invest in services and technology to partner with our customers, so they can provide essential services every day.

We source, import, market and distribute products, both locally and abroad, and we then sell these to customers. Our people facilitate all aspects of this operation, as well as support functions.

Our operations include 640 branches in Australia and New Zealand, and 189 branches across the Sun Belt region in the US. We have two regional head offices to support the branches, located in Melbourne, Australia and Dallas, US. The Group does not have any offshore manufacturing facilities.

## We are global

reece group

**United States**

**Australia & New Zealand**

### ◆ MORSCO
- Plumbing
- Waterworks
- HVAC-R
- Bathrooms & Kitchens

### reece
- Plumbing
- Waterworks
- HVAC-R
- Bathrooms & Kitchens
- Irrigation & Pools

**Suppliers**
# 8K

**Products**
# 300K

**Branches**
# 829

**People**
# 8,000+

**Customers**
# 200K

6    Reece Group

- Select label: **Yes**
- Add the supporting text: *Reece Group is a leading distributor of plumbing, waterworks, and HVAC-R to more than 200,000 residential, commercial and infrastructure customers across Australia, New Zealand, and the United States.* ==//== *We support residential, commercial and infrastructure building trades, and our business is diverse. We are a wholesaler, retailer, and supply chain logistics company, and we invest in services and technology to partner with our customers, so they can provide essential services every day. We source, import, market and distribute products, both locally and abroad, and we then sell these to customers.* ==//== *Our operations include 640 branches in Australia and New Zealand, and*
  *189 branches across the Sun Belt region in the US. We have two regional head offices to support the branches, located in Melbourne, Australia and Dallas, US. The Group does not have any offshore manufacturing facilities.*
- in **Visual Element Category,** select **Infographic** and add the supporting text: *United States* ==//== *Plumbing Waterworks HVAC-R Bathrooms & Kitchens* ==//== *Australia & New Zealand* ==//== *Plumbing Waterworks HVAC-R Bathrooms & Kitchens Irrigation & Pools*
  - Select the Supplementary Button **PDF page number** and enter: 6
- in **Visual Element Category,** select **Table** and add the supporting text: *Products* ==//== *300K* ==//== *Branches* ==//== *829*
  - Select the Supplementary Button **PDF page number** and enter: 6

**Example 2.4.2.c** (Swiss Re, [SID#8933](#)):

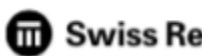 Swiss Re

Swiss Re Australian Modern Slavery Statement

# Swiss Re Modern Slavery Statement

## Introduction

This Modern Slavery Statement (**Statement**) for the financial year ending 31 December 2021 is made pursuant to the *Modern Slavery Act 2018* (Cth) jointly by the following Swiss Re Reporting Entities:

1) Swiss Re Asia Pte. Ltd. (**SRAL**);
2) Swiss Re Australia Ltd (ACN 004 360 909) (**SRAuL**);
3) Swiss Re Life & Health Australia Limited (ACN 000 218 306) (**SRLHA**);
4) Swiss Re International SE (**SRI**); and
5) Swiss Pillar Investments Ltd (**SPIAG**)

(**Swiss Re Reporting Entities, we** or **us**).

Swiss Re Reporting Entities form part of the Swiss Re Group of companies ultimately fully owned by Swiss Re Ltd, a company incorporated in Switzerland and listed on the Swiss Stock Exchange (SIX Swiss Exchange symbol: SREN) (the **Swiss Re Group** or **Swiss Re**).

The Swiss Re Group is a leading wholesale provider of reinsurance, insurance and other insurance-based forms of risk transfer in both the Property and Casualty and Life and Health sectors.

Swiss Re Reporting Entities recognise that 'modern slavery' is used to describe a situation where coercion, threats or deception are used to exploit victims and undermine or deprive them of their freedom. It is used to describe serious exploitation such as human trafficking, servitude, forced marriage, forced labour, debt bondage, deceptive recruiting for labour or services and child labour.

This Statement considers the risk of modern slavery practices in the operations and supply chain of the Swiss Re Reporting Entities and the relevant governance framework.

## Structure, Operations and Supply Chains

### Structure and Operation of Swiss Re Reporting Entities

**SRAL** is a company established in Singapore. SRAL is directly wholly owned by Swiss Re Asia Holding Pte. Ltd. (**SRAHL**), which in turn is ultimately wholly owned by Swiss Re Ltd. SRAL is the entity within Swiss Re that writes Property and Casualty and Life & Health Reinsurance in Asia. In addition to operating in its country of establishment in Singapore, it operates via branches in Australia, Korea, Malaysia, Japan and Hong Kong respectively. As at 31 December 2021, SRAL employed 933 employees in the jurisdictions of its operation described in the foregoing, comprising professional executives, managers and administrative officers.

**SRAuL** is a company incorporated in Australia, wholly owned by SRAHL. SRAuL is the holding company of SRLHA and serves as the Agent in Australia for SRAL and SRI pursuant to s118 of the *Insurance Act 1973* (Cth). SRAuL does not have any employees or operate any business.

**SRLHA** is a company incorporated in Australia, wholly owned by SRAuL. SRLHA also operates in New Zealand via a branch office. SRLHA is a registered life insurance company under

1

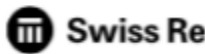

Swiss Re Australian Modern Slavery Statement

**SPIAG** is a company incorporated in Switzerland, wholly owned by Swiss Re Reinsurance Holding Company Ltd which in turn is ultimately owned by Swiss Re Ltd. SPIAG was registered with the Australian Securities and Investments Commission (**ASIC**) on 22 November 2017 as a foreign company carrying on business in Australia. SPIAG is a holding company for Swiss Re Group's real estate business and owns 100% of the shares of its two subsidiaries Swiss Pillar Investments Europe SARL (**Swiss Pillar Europe**), domiciled in Luxembourg, and Swiss Pillar Investments UK Ltd (**Swiss Pillar UK**), domiciled in Cayman. These two subsidiaries hold several Polish, German and UK investment properties. SPIAG held real estate assets in Australia which were disposed of during 2021, bringing its consolidated revenue for the financial year to over A$100 million. SPIAG, Swiss Pillar Europe and Swiss Pillar UK have not had any employees. Other than investing in real estate properties, the acquisition and asset management of which is outsourced to external asset managers, SPIAG and its two subsidiaries do not conduct any opeartions.

## Supply Chain of Swiss Re Reporting Entities

The Swiss Re Reporting Entities procure various goods and services from external suppliers to facilitate their operations. The main categories, by spend, of goods and services procured externally are office and equipment leasing/rent, consulting services, legal services, contractors, recruitment services. Given the global operation of the Swiss Re Reporting Entities, the external suppliers utilised by the Swiss Re Reporting Entities are spread across the globe.

In addition to external suppliers, the Swiss Re Reporting Entities obtain support on certain aspects of its operation of financial services including underwriting, claims, risk, compliance, finance, actuarial, technical accounting, IT, real estate, client management, investment management, business support and internal audit, from other companies within the Swiss Re Group.

2

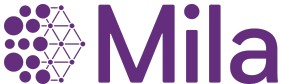

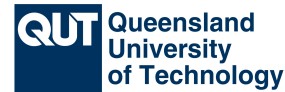

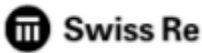

# Assessment of our Operations and Supply Chain under the Modern Slavery Act

## Operational Environment

Swiss Re Reporting Entities operate in a regulated financial services environment, predominantly in developed countries where the rule of law is guaranteed by a form of constitution and independent courts. The nature of our core business – reinsurance and insurance – involves business with other insurance companies and large commercial customers and requires operation by highly skilled or educated employees or external workers. We adhere to Group internal policies and processes relevant in addressing the risk of modern slavery.

### Swiss Re Employees

Recruitment of all employees by Swiss Re is conducted in accordance with our Global Standard on Recruitment. This requires all applicants to undergo formal pre-screening for review of their eligibility, work permits and completeness of documents. All Swiss Re employees are employed by mutual agreement through a written contract governed by local law and terminable by either party.

We also recognise that there are inherent modern slavery risks associated with having operations in countries that have been ranked by the Global Slavery Index as having a higher prevalence of modern slavery. However, these risks are mitigated by virtue of the low risk profile of our highly skilled workforce combined with our compliance with Group internal policies and processes relevant in addressing the risk of modern slavery as described in the previous paragraph.

### External Workforce

The assessment of modern slavery risk involves the review of the process by which and from whom we procure goods and services.

We recognise that the following supply chain categories may have a higher risk of modern slavery associated with the country of origin/manufacture, industry and workforce: IT and telecommunications hardware, IT system support, logistics, postal and delivery services, utilities, promotional merchandise, stationery and uniforms, cleaning and maintenance services, and food, beverage and catering services.

We have also taken note of our non-controlling investments in our risk assessment.

## Risk Analysis Process

We identified the departments that oversee our operations, asset management, human resources and procurement of goods and services for the Swiss Re Reporting Entities as the key focus for the assessment.

A risk assessment of our Australian operations and supply chain for the financial year of 2021, with the additional focus on a number of material operational service providers, was conducted by way of due diligence questionnaires to representatives in these teams. Specifically, we conducted a supplier mapping and risk assessment exercise in relation to suppliers paid by Swiss Re operations.

3

- ■ Select label: **Yes**
- ■ Add the supporting text: *The Swiss Re Group is a leading wholesale provider of reinsurance, insurance and other insurance-based forms of risk transfer in both the Property and Casualty and Life and Health sectors.* // *SRAL is the entity within Swiss Re that writes Property and Casualty and Life & Health Reinsurance in Asia. In addition to operating in its country of establishment in Singapore, it operates via branches in Australia. As at 31 December 2021, SRAL employed 933 employees in the jurisdictions of its operation described in the foregoing, comprising professional executives, managers and administrative officers.* // *SRAuL does not have any employees or operate any business.* // *SRLHA also operates in New Zealand via a branch office. SRLHA is a registered life insurance company under* // *section 21 of the Life Insurance Act 1995 (Cth) and is authorised and supervised by the Australian Prudential Regulation Authority (APRA). SRLHA writes Life and Health Reinsurance in Australia and New Zealand and issues direct life and health insurance to retail customers through third party distributors. SRLHA and the Australia branch of SRAL form Swiss Re's reinsurance business operation in Australia and New Zealand.* // *SRI operates across many jurisdictions via its headquarter in Luxembourg, 13 branch offices in Australia, Denmark, France, Germany, Hong Kong, Italy, Japan, Netherlands, Singapore, Slovakia, Spain, Switzerland, UK, and a wholly owned subsidiary in the People's Republic of China (PRC), Swiss Re Corporate Solutions Insurance China Ltd (SRCSIC), which in turn has two branches within PRC in Beijing and Jiangsu. As at 31 December 2021, SRI and SRCSIC employed 312 and 136 employees respectively in the jurisdictions of its operation described in the foregoing, comprising professional executives, managers and administrative officers.* // // *These two subsidiaries hold several Polish, German and UK investment properties. SPIAG held real estate assets in Australia which were disposed of during 2021, bringing its consolidated revenue for the financial year to over A\$100 million.* // *Other than investing in real estate properties, the acquisition and asset management of which is outsourced to external asset managers, SPIAG and its two subsidiaries do not conduct any operations.* // *The nature of our core business – reinsurance and insurance – involves business with other insurance companies and large commercial customers and requires operation by highly skilled or educated employees or external workers.*
- ■ **Note:** many of the sentences (such as the one starting with "*As at 31 December 2021, SRAL employed…*") extracted here would also be considered relevant for structure (2.4.1).

## 2.4.3. Supply chains

**Question:** does the reporting entity describe its supply chains?

**Background:** the supply chains refer to the sequences of processes involved in the procurement of products and services (including labour) that contribute to the reporting entity's own products and services. Supply chains are not restricted to "tier one" (or direct) suppliers in the context of the statements; these should cover suppliers of products and services in Australia or overseas that extend beyond the reporting entity's direct suppliers.

An entity can describe their supply chains in multiple ways, such as by describing:

1. **Products provided by suppliers**. For example, a supermarket's supply chains include the provision of frozen vegetables via a wholesaler. A hospital's supply chains include the procurement of rubber gloves. A bank's supply chains include the procurement of uniforms and office supplies.
2. **Services provided by suppliers**. For example, a law firm's supply chains include the cleaning services provided to the law firm by a contractor. The supply chains of a superannuation fund or fund manager include its external investment managers, who provide a service by managing its portfolios. A building company's supply chains include its specialist subcontractors.
3. **The location, category, or other attributes of the suppliers.** For example, a company may state that they only interact with suppliers from a specific country or location (e.g. "Bahamas", or "overseas"). They may also describe constraints on the origin of workers in their supply chain (e.g. by mentioning that they prefer indigenous suppliers, or go with client preferences). A company may also describe a supplier based on the category of supplies it provides (e.g. "commodity chemicals suppliers").

**Notes:**

- The presence of any relevant information should lead to a "Yes" label for this requirement.
- Descriptions of products and services used by indirect suppliers in the reporting entity's supply chain must also be extracted as supporting text. For example, the description of the supply chains of an electronics manufacturer should ideally include the mining and refining of minerals used in electronic components purchased by the electronics manufacturer.
- References to any disclosures or documents by the reporting entity about the identity of their suppliers (such as a public supplier list) should be extracted as supporting text. The contents of those documents should however **never** be extracted.

**Label options and expected supporting elements for this question:**

■ **Yes**, the statement describes the supply chains of the reporting entity:

- Copy-paste the text passages from the statement that justify that the reporting entity described its supply chains.
- If any relevant information comes in other formats than text, fill in the required information via the Supplementary Button: "Visual Element Category", note the page where it is found, and extract any relevant text it contains (if possible).

- **No**, the statement does not describe the reporting entity's supply chains:
  - Copy-paste the exact text passages from the statement that justifies that the entity does not meet this criterion,

    **OR**
  - If no information is found about this label, select the Supplementary Button: "No relevant information found in the statement".

- **Unclear**, in any other case:
  - Select this label if the information found is unclear or there are other concerns. If you decide to select this label, you have to choose the Compulsory Supplementary Button "Explanation" and justify why.

**Example 2.4.3.a** (Duratec, SID#7832):

## SUPPLY CHAIN

Our supply chain includes providers of remediation products and other project focused materials purchased and distributed through Duratec warehouses or delivered directly to project sites through third parties. Products are purchased domestically and imported through third party logistics providers. Our suppliers are located principally in Australia and at least 12 foreign countries.

Our supply chain also includes the suppliers of products and services that would typically be required by offices and small warehouse locations.

Due to the various sectors Duratec engages in, the supply chain providers are varied and based heavily on the relevant industry sector.

## Core Service Offerings

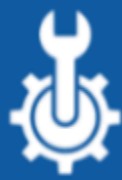

**ENGINEERING**

Our industry professionals provide rapid and accurate diagnosis of asset condition and degradation, including cost-effective durability solutions.

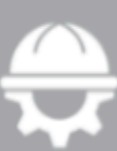

**REMEDIATION**

We offer the complete range of remediation services including responsive, scheduled and performance-based maintenance programs.

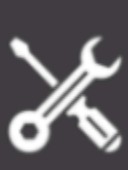

**CONSTRUCTION**

As a licensed builder in all states, we offer design and construction services to replace structures that have reached the end of their serviceable life.

## Client industries

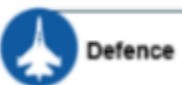 Defence

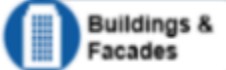 Buildings & Facades

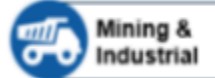 Mining & Industrial

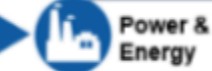 Power & Energy

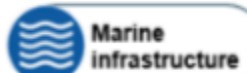 Marine infrastructure

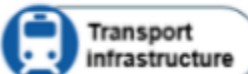 Transport infrastructure

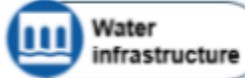 Water infrastructure

## 3. RISKS OF MODERN SLAVERY PRACTICES IN OUR OPERATIONS AND SUPPLY CHAINS

### RISKS IN OUR OPERATIONS

All of our employees and operations are located in the Commonwealth of Australia and we comply with all Australian labour laws and other applicable employment related legislation. Our internal reviews have found no evidence of any modern slavery practices or incidents in our operations and we do not believe there is material modern slavery risk in this area.

### RISKS IN OUR SUPPLY CHAIN

No evidence came to light indicating that any of our suppliers had (or were aware of) any instances of modern slavery in their operations or supply chains.

Our due diligence through the end of the reporting period indicates that the higher risks of modern slavery in our supply chain likely reside principally in (a) the two first tier suppliers that have been specifically identified as High risk in our due diligence done to date, (b) the few direct foreign suppliers located in higher risk countries, and (c) indirect foreign suppliers in the second tier of our supply chain and beyond and for whom we still have quite low visibility (collectively, the "Higher Risk Supplier Group").

The due diligence completed through the end of the reporting period has been limited to Australian based, first tier suppliers and only two of have been identified as High risk on the basis of the due diligence done. One of these has operations in a higher risk country and the other's primary focus is in the maritime area. Certain recurring themes within the Higher Risk Supplier Group include:

- Little or no visibility into the labour practices in their supply chains and no substantive assessment, verification or mitigation of modern slavery risk has been made in their operations or supply chain and no procedures are in place to facilitate such an assessment, verification or any mitigation.
- No contract warranties and covenants are in place that specifically and robustly address modern slavery issues in their procurement agreements.
- No formal company policies are in place re: modern slavery avoidance or Whistleblower encouragement and protection.
- No obligations are imposed by our suppliers on their suppliers to implement robust contract warranties and covenants relating to modern slavery issues on their suppliers.
- Some are unsure of the extent of use of low-skilled (including foreign migrant) workers in their operations or supply chains and/or whether or to what extent high risk factors relating to the use of such workers exist.

A significant number of the suppliers we targeted for due diligence have not yet responded to our due diligence inquiries.

- ■ Select label: **Yes**
- ■ Add the supporting text: *Our supply chain includes providers of remediation products and other project focused materials purchased and distributed through Duratec warehouses or*

*delivered directly to project sites through third parties. Products are purchased domestically and imported through third party logistics providers. Our suppliers are located principally in Australia and at least 12 foreign countries. Our supply chain also includes the suppliers of products and services that would typically be required by offices and small warehouse locations. Due to the various sectors Duratec engages in, the supply chain providers are varied and based heavily on the relevant industry sector. //  The due diligence completed through the end of the reporting period has been limited to Australian based, first tier suppliers and only two of have been identified as High risk on the basis of the due diligence done.  One of these has operations in a higher risk country and the other's primary focus is in the maritime area.*

**Example 2.4.3.b** (Uniqlo, SID#1535):

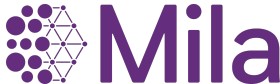

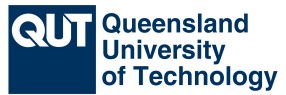

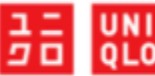

## MODERN SLAVERY STATEMENT OF UNIQLO AUSTRALIA

This statement is in accordance with the requirement that Uniqlo Australia Pty Ltd (**Uniqlo Australia**) report for the fiscal year from 1 September 2019 to 31 August 2020 (**Fiscal Year 2020**) under the *Modern Slavery Act 2018* (Cth) (**Modern Slavery Act**). In this statement, Uniqlo Australia adopts the meaning of modern slavery as defined in the Modern Slavery Act.

Uniqlo Australia is committed to taking action to eliminate modern slavery from our supply chain and business. As a member of the Fast Retailing Group (**Fast Retailing**), our mission is to create truly great clothing with new and unique value, and to enable people all over the world to experience the joy, happiness and satisfaction of wearing such great clothes. The concept of truly great clothing includes conducting ourselves in good faith, respecting the environment and upholding human rights across all business activities.

## REPORTING ENTITY AND CORPORATE STRUCTURE OF REPORTING ENTITY

Uniqlo Australia is a wholly owned subsidiary of Fast Retailing Singapore, itself a wholly owned subsidiary of Fast Retailing Co., Ltd. (**Fast Retailing**), a company which is listed on the Tokyo Stock Exchange. Uniqlo Australia does not itself own or control other entities.

## OPERATIONS AND SUPPLY CHAINS OF THE REPORTING ENTITY

### Operations: retail sale of UNIQLO casual wear

Uniqlo Australia is a retailer of UNIQLO branded men's, women's and children's fashion apparel, footwear, small gift items and related accessories (**UNIQLO casual wear**). We employ over 1,800 people across our Melbourne headquarters and 25 Australian retail stores.

### Supply chains: UNIQLO casual wear

Uniqlo Australia does not manufacture UNIQLO casual wear.

Fast Retailing is an apparel retail group with global operations, mainly conducted through its UNIQLO casual wear brand. Fast Retailing engages with garment factories (**Partner Factories**) and fabric mills to produce fabric and manufacture garments. Fast Retailing does not own Partner Factories or fabric mills.

Uniqlo Australia procures UNIQLO casual wear through our supplier agent, Uniqlo Co., Ltd., also a subsidiary of Fast Retailing, however the Partner Factories are the ultimate suppliers of the product in the supply chain.

In this way, the Partner Factories form part of Uniqlo Australia's supply chain.

### Supply chains: other goods and services

Uniqlo Australia procures a range of other goods and services from Australian and international suppliers. For example, information and communication technology services and products, marketing and media collateral and services, various professional services

UNIQLO Australia Pty Ltd   Level 15, 385 Bourke Street, Melbourne, VIC 3000
ABN 36161922103

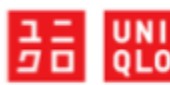

(including facilities and property management, financial, insurance and legal), and transport, storage, construction, security and maintenance services.

## RISKS OF MODERN SLAVERY PRACTICES IN THE OPERATIONS AND SUPPLY CHAINS OF THE REPORTING ENTITY

### Operations

Uniqlo Australia has assessed its retail operations as being at low risk of modern slavery.

Our operations primarily involve our personnel delivering a retail experience in stores or carrying out various other business and operational functions in our Melbourne headquarters. Uniqlo Australia has human resources policies and procedures in place and is compliant with Australia's framework of legislation, including the *Fair Work Act 2009* (Cth), that govern its workplaces. This framework regulates and protects our workplace conditions and employee health and safety and operates to prevent modern slavery in Australian workplaces.

### Supply chains: UNIQLO casual wear

Production of fabric for and manufacture of UNIQLO casual wear is conducted in Partner Factories and fabric mills in countries including China, Vietnam, Indonesia, Bangladesh, Turkey, and India. Workers in fabric and garment manufacturing in some of these regions of the world are known to be at risk of modern slavery.

In particular, Fast Retailing recognises there are salient human rights risks in its supply chain. These can include child labour, illegal employment of young workers, forced labour, coercion and harassment, discrimination, serious health and safety violations, serious violation of freedom of association, insufficient wage payments, excessive working hours, transparency issues such as false records and unauthorised subcontracting. Uniqlo Australia therefore recognises that these human rights risks are also indicators of a risk of modern slavery in our supply chain.

### Supply chains: other goods and services

We procure goods and services from international and Australian providers and acknowledge that suppliers present different forms and degrees of modern slavery risks.

Uniqlo Australia undertook a review of all suppliers of other goods and services in the Fiscal Year 2020. Our review indicated there is not a high risk of modern slavery in this aspect of our supply chain. Most of the suppliers to Uniqlo Australia are service providers, whose personnel are providing services within Australia and therefore will be governed by a framework of Australian legislation.

However, we acknowledge that risks of modern slavery can be "hidden" further along a supply chain. For this reason, Uniqlo Australia will complete a further review of suppliers in future. Areas Uniqlo Australia may focus on, due to those areas having higher modern slavery risk factors, are likely to include security services, ICT and electronics procurement, and cleaning services.

UNIQLO Australia Pty Ltd    Level 15, 385 Bourke Street, Melbourne, VIC 3000
ABN 36161922103

- ■ Select label: **Yes**
- ■ Add the supporting text: *Fast Retailing engages with garment factories (Partner Factories) and fabric mills to produce fabric and manufacture garments.* // *Uniqlo Australia procures UNIQLO casual wear through our supplier agent, Uniqlo Co., Ltd., also a subsidiary of Fast Retailing, however the Partner Factories are the ultimate suppliers of the product in the supply chain. In this way, the Partner Factories form part of Uniqlo Australia's supply chain.* // *Uniqlo Australia procures a range of other goods and services from Australian and international suppliers. For example, information and communication technology services and products, marketing and media collateral and services, various professional services* // *(including facilities and property management, financial, insurance and legal), and transport, storage, construction, security and maintenance services.* // *Production of fabric for and manufacture of UNIQLO casual wear is conducted in Partner Factories and fabric mills in countries including China, Vietnam, Indonesia, Bangladesh, Turkey, and India.* // *We procure goods and services from international and Australian providers and acknowledge that suppliers present different forms and degrees of modern slavery risks.* // *Most of the suppliers to Uniqlo Australia are service providers, whose personnel are providing services within Australia and therefore will be governed by a framework of Australian legislation.* // *Areas Uniqlo Australia may focus on, due to those areas having higher modern slavery risk factors, are likely to include security services, ICT and electronics procurement, and cleaning services.*

**Example 2.4.3.c** (Export Finance Australia, SID#10720)**:**

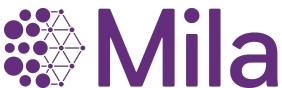
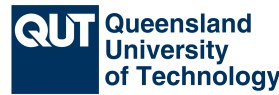

## Our operations, supply chain and risks of modern slavery practices

We take a risk-based approach to modern slavery risk in our business operations, supply chain and customer relationships.

As we are an Australian-based entity and our primary operations of providing finance are office-based, we consider the risk of modern slavery occurring directly within our business operations to be low. However, we recognise that there is the risk of modern slavery occurring and being supported by us indirectly through our supply chain or via our customers and the transactions and projects we support.

In accordance with the United Nations Guiding Principles on Business and Human Rights, we have initially prioritised and continue to prioritise our focus and actions on areas where we consider the potential impact of modern slavery to be greatest and where we might have a greater opportunity to influence outcomes.

As such, at this stage our identification and assessment of modern slavery risk centres primarily on the financial transactions we support, rather than on Key Supplier Contracts. For context, the aggregate annual dollar value of our key supplier contracts for FY2022 was equivalent to around 1 per cent of the dollar value of our FY2022 transactional exposure. In making the statements above, we have considered and contrasted the overall modern slavery risk profile of our customers and transactions with that of our supply chain.

We consider our Key Supplier Contracts to be our key supplier engagements. These are representative of our core supply chain categories, set out below in Figure 3: supply chain categories - by dollar value.

As a general observation, the nature of our key supplier contracts tends to be more medium to long term rather than a large number of high turnover engagements. Around 15 per cent of our key supplier contracts were entered into in FY2022, with the remainder being ongoing contracts entered into in prior financial years.

During FY2022, we continued to provide transparency for our Key Supplier Contracts by disclosing key details about them publicly on our website. We update this listing every six months.

In FY2022 there were 84 Key Supplier Contracts still being performed or entered into. Based on the value of these contracts, Australian-based suppliers comprised approximately 93 per cent, with the balance predominantly comprised of suppliers based in the USA, Ireland and the UK.

The percentage by number of the Key Supplier Contracts rather than by value—was similar, with Australian-based suppliers constituting approximately 88 per cent. A breakdown of these figures is set out in Figure 4: FY2022 Key Supplier Contract jurisdictions by dollar value and Figure 5: FY2022 Key Supplier Contract jurisdictions by number of contracts.

In addition to our Key Supplier Contracts, we have in place property lease arrangements and other arrangements under which we receive services.

In FY2022, these arrangements were relatively small in number and value compared to our Key Supplier Contracts. They were typically entered into with Australian-based entities, including Australian Commonwealth entities. For these reasons, we consider these arrangements and engagements generally as lower risk and therefore have not included these items within our Key Supplier Contract Analysis.

The types of industries, jurisdictions and nature of our FY2022 CA and NIA transactional exposures reflect our role as a provider of financing solutions to support exporters and overseas infrastructure development. It also reflects the assistance we provide to other government entities.

Our FY2022 CA and NIA exposures are set out in Figure 6: FY2022 transaction exposure breakdown.

As noted above, we have identified our customer relationships and the transactions and projects we support as a continued priority for our focus and actions in this space.

### Figure 3: Supply Chain Categories - By Dollar Value

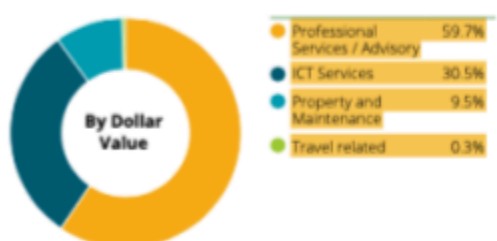

| | |
|---|---|
| Professional Services / Advisory | 59.7% |
| ICT Services | 30.5% |
| Property and Maintenance | 9.5% |
| Travel related | 0.3% |

### Figure 4: FY2022 Key Supplier Contract Jurisdictions by Dollar value

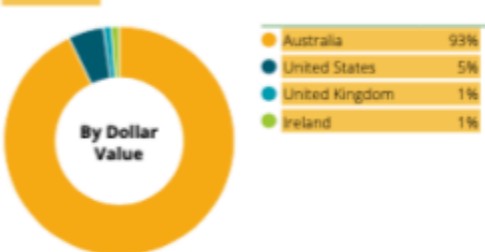

| | |
|---|---|
| Australia | 93% |
| United States | 5% |
| United Kingdom | 1% |
| Ireland | 1% |

### Figure 5: FY2022 Key Supplier Contract Jurisdictions by number of Contracts

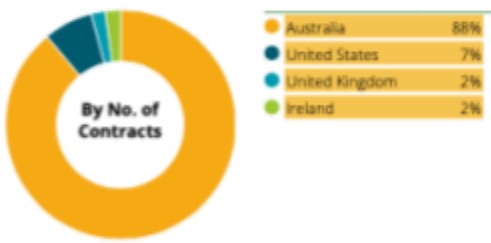

| | |
|---|---|
| Australia | 88% |
| United States | 7% |
| United Kingdom | 2% |
| Ireland | 2% |

- Select label: **Yes**
- Add the supporting text: *As a general observation, the nature of our key supplier contracts tends to be more medium to long term rather than a large number of high turnover engagements.* // *During FY2022, we continued to provide transparency for our Key Supplier Contracts by disclosing key details about them publicly on our website.* // *In FY2022 there were 84 Key Supplier Contracts still being performed or entered into. Based on the value of these contracts, Australian-based suppliers comprised approximately 93 per cent, with the balance predominantly composed of suppliers based in the USA, Ireland and the UK. The percentage by number of the Key Supplier Contracts rather than by value—was similar, with Australian-based suppliers constituting approximately 88 per cent. A breakdown of these figures is set out in Figure 4: FY2022 Key Supplier Contract jurisdictions by dollar value and Figure 5: FY2022 Key Supplier Contract jurisdictions by number of contracts. In addition to our Key Supplier Contracts, we have in place property lease arrangements and other arrangements under which we receive services. In FY2022, these arrangements were relatively small in number and value compared to our Key Supplier Contracts. They were typically entered into with Australian-based entities, including Australian Commonwealth entities.*
- in **Visual Element Category,** select **Infographic** and add the supporting text: *Figure 3: Supply Chain Categories - By Dollar Value* // *Professional Services / Advisory 59.7% ICT Services 30.5% Property and Maintenance 9.5% Travel related 0.3%*
    - Select the Supplementary Button **PDF page number** and enter: 5
- in **Visual Element Category,** select **Infographic** and add the supporting text: *Figure 4: FY2022 Key Supplier Contract Jurisdictions by Dollar value* // *Australia 93% United States 5% United Kingdom 1% Ireland 1%*
    - Select the Supplementary Button **PDF page number** and enter: 5
- in **Visual Element Category,** select **Infographic** and add the supporting text: *Figure 5: FY2022 Key Supplier Contract Jurisdictions by number of Contracts* // *Australia 88% United States 7% United Kingdom 2%* // *Ireland 2%*
    - Select the Supplementary Button **PDF page number** and enter: 5

**Example 2.4.3.d** (Volvo, SID#8717)**:**

# Our operations are truly global

Volvo Car Australia forms part of the Volvo Car Group, which is a global organisation. While our roots are decidedly Swedish, we have manufacturing, research and design operations in Europe, Asia and the Americas and a wide range of suppliers located throughout the world. The Volvo Car Group designs, develops, manufactures and sells a range of premium cars and services. Recognised for our design, safety and technological innovations, our cars are designed and developed by award-winning teams at design centres in Sweden, China and the US.

We have a global presence, with sales in over 100 countries, including in Australia.

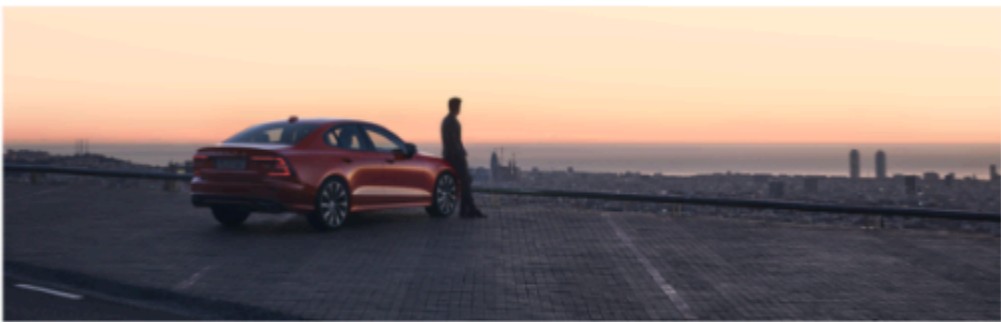

## Our Australian operations and supply chain

Volvo Car Australia markets and distributes Volvo Car products in Australia. Volvo Australia has no active entities which it owns or controls.

We sell, repair and maintain Volvo Cars' products through approved third party partners including third party dealers, mechanics and engineers, financial partners and insurance companies. We provide all business partners with the Volvo Car Group's Code of Conduct for Business Partners which sets out our expectations and principles for third parties (described further below). Knowledge of the Code of Conduct for Business Partners is assessed as part of our standard dealer audits.

We import Volvo Car products into Australia from our parent company, Volvo Car Corporation (Volvo Cars). No products are manufactured by Volvo Car Australia or in Australia. The Volvo Car Group manages the procurement of products and services from third parties connected to the manufacturing and distribution of Volvo Cars' products, which end up for sale to consumers in Australia. Volvo Car Australia relies on the Volvo Cars to ensure that third party suppliers meet the Volvo Car Group's high ethical standards and undertakes assessments to ensure business partners are compliant.

Locally, Volvo Car Australia has implemented processes requiring local business partners to comply with the law and meet Volvo Car's standards of ethical business conduct. Many of our suppliers and third party dealers are large companies with established social governance policies, including in relation to modern slavery, which affords Volvo Car Australia some assurance in engaging in those commercial relationships.

Details on the steps that Volvo Car Australia and Volvo Cars takes to address modern slavery risk in its supply chain are set out below.

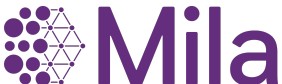
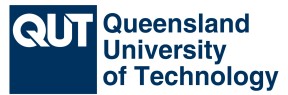

## Risks of modern slavery in our supply chains

As a subsidiary of a global company with complex supply chains, Volvo Car Australia acknowledges that challenges connected to our operations exist. Modern slavery risk covers a broad range of practices and includes slavery, debt bondage, forced labour, child labour, forced marriage and other exploitative practices including wage theft, abuse and suppression of freedom of association.

In our work to secure responsible sourcing we take a risk-based approach. Based on this approach, we have enhanced our due diligence efforts for suppliers located in high-risk countries (discussed further below). Moreover, with our shift to electrification, we acknowledge the ethical, social and environmental risks (including modern slavery) connected to the extraction, processing, trade and transportation of the raw materials needed for the production of EV batteries (e.g. cobalt, nickel, lithium). Tin, tantalum, tungsten and gold – also known as '3TG' or 'conflict minerals[2]' – are other raw materials used in components in our cars, which in some cases are related to negative impacts on people and planet, including modern slavery. To prevent and mitigate these risks, we have implemented due diligence efforts for these supply chains (discussed further below).

The Volvo Car Group acknowledges that due diligence is an ongoing, proactive, and reactive process and that information and risk monitoring systems will progressively be improved over time as a result from constructive engagement with suppliers and other stakeholders.

There is also the risk that modern slavery can occur more locally, closer to home. Volvo Car Australia is aware and remains vigilant about the increased risk connected to other vulnerable groups such as base-skilled workers engaged by our business partners in Australia.

## Our approach towards subcontracting

Subcontracting by our suppliers is a usual practice in the manufacturing and automotive industry. Volvo Cars manages subcontracting by imposing rules for subcontracting in its contracts with suppliers. The terms include a requirement that subcontractors accept Our Code of Conduct for Business Partners, follow the same terms as Volvo Cars has agreed with suppliers and includes the right to audit any subcontractor. Volvo Cars continuously seeks improvements on its ability to monitor its entire supply chain.

At Volvo Car Australia we manage this by permitting subcontracting, subject to our consent and subject to any subcontractor also agreeing to Our Code of Conduct for Business Partners and ensuring we have the right to audit any subcontractor.

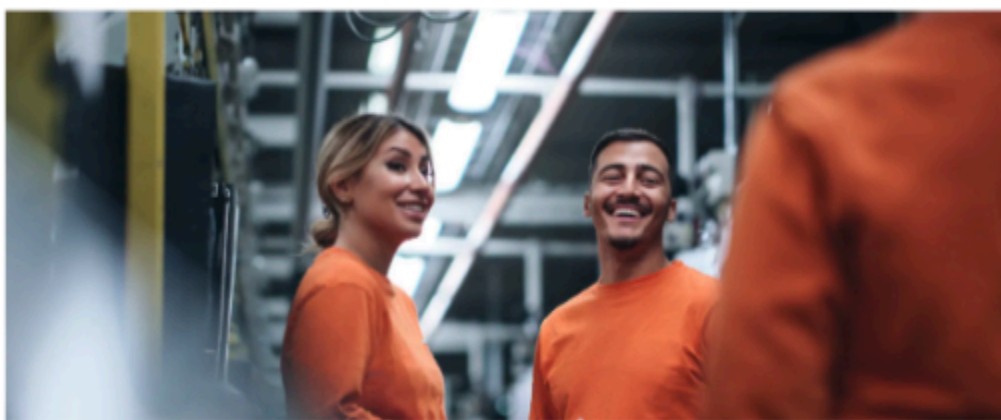

2. "Conflict minerals" i.e. tin, tantalum, tungsten and gold (3TG) from the Democratic Republic of the Congo or its nine adjoining countries as well as other minerals from conflict-affected and high-risk areas.

**VOLVO CAR AUSTRALIA**
**MODERN SLAVERY STATEMENT 2021**

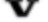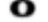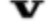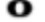

7

Mila – Quebec Institute of Artificial Intelligence
https://mila.quebec/en/ai4humanity/applied-projects/ai-against-modern-slavery-aims
92
6666, Saint-Urbain Street, Suite 200 Montreal, Quebec H2S3H1

- Select label: **Yes**
- Add the supporting text: *While our roots are decidedly Swedish, we have manufacturing, research and design operations in Europe, Asia and the Americas and a wide range of suppliers located throughout the world.* // *We import Volvo Car products into Australia from our parent company, Volvo Car Corporation (Volvo Cars).* // *The Volvo Car Group manages the procurement of products* // *and services from third parties connected to the manufacturing and distribution of Volvo Cars' products, which end up for sale to consumers in Australia.* // *Moreover, with our shift to electrification, we acknowledge the ethical, social and environmental risks (including modern slavery) connected to the extraction, processing, trade and transportation of the*
  *raw materials needed for the production of*
  *EV batteries (e.g. cobalt, nickel, lithium). Tin, tantalum, tungsten and gold – also known as '3TG' or 'conflict minerals2' – are other raw materials used in components in our cars, which in some cases are related to negative impacts*
  *on people and planet, including modern slavery.* // *2. "Conflict minerals" i.e. tin, tantalum, tungsten and gold (3TG) from the Democratic Republic of the Congo or its nine adjoining countries as well as other minerals from conflict-affected and high-risk areas.* // *Subcontracting by our suppliers is a usual practice in the manufacturing and automotive industry.*
- **Note:** this is an example of "Descriptions of products and services used by indirect suppliers in the reporting entity's supply chain" because the reporting entity mentions the use of various minerals in the manufacturing of components for the cars it procures from its parent company.

- 

---

**Annotation Workflow:**
**The labels and their supplementary buttons should be:**

**2.4.1. Structure**
- Yes:
  - Copy-Paste supporting text passage
  - Supplementary Button: "Visual Element Category"
- No:
  - Copy-Paste supporting text passage
  - Supplementary Button: "No relevant information found in the statement"
- Unclear:
  - Compulsory Supplementary Button "Explanation"

---

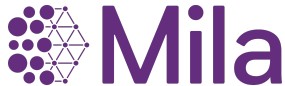
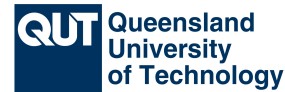

### 2.4.2. Operations
- Yes:
    - Copy-Paste supporting text passage
    - Supplementary Button: "Visual Element Category"
- No:
    - Copy-Paste supporting text passage
    - Supplementary Button: "No relevant information found in the statement"
- Unclear:
    - Compulsory Supplementary Button "Explanation"

### 2.4.3. Supply chain
- Yes:
    - Copy-Paste supporting text passage
    - Supplementary Button: "Visual Element Category"
- No:
    - Copy-Paste supporting text passage
    - Supplementary Button: "No relevant information found in the statement"
- Unclear:
    - Compulsory Supplementary Button "Explanation"

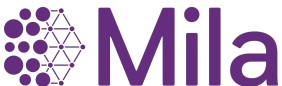

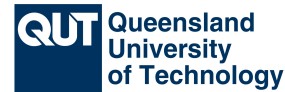

## 2.5. Mandatory Criteria 3 to 5: Risks and actions

**Background for the "risks and actions" questions:** the Australian Modern Slavery Act (MSA) requires reporting entities to describe the risks of modern slavery in their supply chains and operations and their actions to address these risks. Specifically,

- **Criterion 3** mandates reporting entities to provide a description of the risks related to modern slavery practices within their operations and supply chains. This description should also cover the operations and supply chains of entities they own or control.
- **Criterion 4** mandates reporting entities to outline their actions to assess and address these risks, such as implementing due diligence and remediation processes.
- **Criterion 5** mandates reporting entities to explain how they assess the effectiveness of their actions to assess and address these risks.

The following sections (**2.5.1 to 2.5.4**) break down these criteria into four questions:

**2.5.1.** **Does the reporting entity describe its modern slavery risks?** We want to know what modern slavery risks the reporting entity found in its activities and supply chains.
**2.5.2.** **Does the reporting entity describe the actions applied to identify, assess, and mitigate the modern slavery risks it identified?** We want to know how the reporting entity took action to identify and assess the risks as well as to avoid modern slavery from occurring in its operations and supply chains.
**2.5.3.** **Does the reporting entity describe remediation actions for modern slavery cases?** When cases of modern slavery have been found (or when they might be found in the future), we want to know what actions the reporting entity took (or will take) to remediate these cases.
**2.5.4.** **Does the reporting entity describe how it assesses the effectiveness of its actions?** Given the actions taken to identify and mitigate risks as well as to remediate modern slavery issues described above, we want to know how the reporting entity assesses whether these actions are effective.

**Notes:**

🚩 **Descriptions should cover owned and controlled entities.** Criteria 3, 4, and 5 require reporting entities to also provide information that covers their owned or controlled entities. For simplicity, in the following subsections, we do not always mention that statements covering owned or controlled entities must be extracted, but this is always the case.

🚩 **Regarding future plans and intentions of entities.** The reporting entities may state their plans and intentions for future reporting periods, and this can be considered supporting text. However, **vague plans** (e.g. "*we will review all risks in this category*") and **vague intentions** (e.g. "*we will address this problem later*") **are NOT evidence** that they are fulfilling their requirements, and should therefore not be

considered supporting text. The proposed actions should be described in order to be considered valid evidence (e.g. "*we will have suppliers fill a questionnaire to review all risks in this category*").

🚩 **Regarding sentence relevance.** To answer the questions of Criteria 3 to 5, it is important to distinguish between concrete and vague information. We aim to extract only supporting text that describes requirements by the Act along with relevant context. When the reporting entity states for example that it "*understands its responsibility*" or that it "*wants to fight modern slavery*", this is not relevant information for any criteria. Avoid including sentences that only contain such empty declarations.

🚩 **Accounting for linguistic and cultural differences in the identification of "modern slavery".** When a reporting entity **explicitly** states that their "*Human Rights Policy*" or "*Corporate Social Responsibility Initiative*" (or any other similar practice) covers the issue of "*modern slavery*", then all references to these policies, initiatives, or practices **should be treated as potentially relevant** for the questions of Sections 2.5.1 to 2.5.4. They might for instance specify that their "*human rights policy covers modern slavery*", in which case all references back to their human rights policy (e.g. "*We train our staff on our human rights policy*") should be considered potentially relevant text. See **Example 2.5.a** below.

**Example 2.5.a** (Obayashi, SID#10080) a case where the entity's "*CSR Procurement Guidelines*" (also referred to as its "*CSR Procurement Policy*") explicitly covers forms of modern slavery, and as such, any reference to these guidelines or policy should be considered potentially relevant for the following questions.

### 2-4. Obayashi Group CSR Procurement Policy

The Group established the Obayashi Group CSR Procurement Policy to specify the actions aimed at realizing a sustainable society described in the Obayashi Basic Principles. It includes compliance with laws and regulations, respect for human rights, assurance of safety and health, ensuring quality, and consideration for the environment. The Basic Policy on CSR Procurement Initiatives specifies items that Obayashi Group officers and employees need to comply with in promoting CSR procurement.

The CSR Procurement Guidelines prescribe items that the Obayashi Group expects its suppliers to implement and to make sure that everyone in the suppliers' own supply chains understand, in line with the Basic Policy on CSR Procurement Initiatives. Specific guidelines state that suppliers will "not discriminate . . . nor engage in harassment, or other activities that infringe upon human rights," "not engage in or permit misconduct such as the use of forced labor or child labor," "ensure equal employment opportunity, practice appropriate labor management, pay appropriate wages, and maintain and improve working environments so they are healthy and easy to work in," "guarantee freedom of association and collective bargaining rights for employees," "strive to understand and cooperate with the human rights due diligence performed by Obayashi Group companies," etc.

See the following website link for further information on the Obayashi Group CSR Procurement Policy and CSR Procurement Guidelines:

Obayashi Group CSR Procurement:
https://www.obayashi.co.jp/en/sustainability/suppliers/csr_procurement.html

Obayashi Group CSR Procurement Guidelines:
https://www.obayashi.co.jp/en/sustainability/suppliers/csr_procurement.html#section2

## 2.5.1. Risk descriptions

**Question:** does the reporting entity describe the modern slavery risks it identified?

**Background:** the reporting entity should describe in its statement **the modern slavery risks** in its operations and supply chains. By "risk of modern slavery", we mean the potential for the entity to cause or to contribute to modern slavery through its domestic and international operations and through its supply chains.

The supporting text for this criteria should only present the risks, and not how the company identified, assessed or addressed these risks.

The reporting entity might describe its modern slavery risks in relation to specific:
- **geographic regions** (e.g. Indonesia, south-east asia, or simply "overseas");
- **industries** or **industrial sectors** (e.g. maritime shipping, or agriculture);
- **resources** or **commodities** that are commonly tied to modern slavery (e.g. petroleum, palm oil, electronics, cocoa, conflict minerals);
- **product/service categories** (e.g. fast fashion, gig work);
- **workforces** (e.g. migrant workers, temporary workers, third-party labour);
- **world events** (e.g. sporting events, natural disasters, the COVID-19 pandemic);
- **existing or past cases of modern slavery**;
- **individual risks** (i.e. risks tied to specific entities or suppliers linked with the reporting entity).

While entities may not have directly identified modern slavery **cases** (or, synonymously, incidents, issues, or situations) in their operations or supply chains, every entity possesses modern slavery **risks** that need to be declared.

**Notes:**

- Stating that the reporting entity in its entirety has a "*low risk*" of modern slavery or that its operations or supply chains are "*slavery-free*" **is not sufficient** to fulfill this requirement. The reporting entity needs to describe at least one specific area of its supply chains or operations that has a modern slavery risk (whether low or not), or an actual case of modern slavery.
- If the reporting entity only describes an actual case of modern slavery and describes no other risk, the label for this question should still be **Yes**.
- Modern slavery risks and cases can be properly declared without naming specific operations, subsidiaries, suppliers, facilities, or products. For example, a reporting entity may adequately declare its risks by reporting that it "*identified risks relating to its sourcing of timber products from countries A, B and C, which may be produced using forced labor*", or that it "*has suppliers providing cleaning services at higher risk for breaches of modern slavery obligations*".

**Label options & their associated supporting text passages extracted from the statements:**

- **Yes,** the statement describes one or more modern slavery risks or cases:
  - Copy-paste the text passages from the statement that describe risks or cases of modern slavery.
  - If any relevant information comes in other formats than text, fill in the required information via the Supplementary Button: "Visual Element Category", note the page where it is found, and extract any relevant text it contains (if possible).
- **No,** the statement does not describe any modern slavery risk or case:
  - Copy-paste the text passages from the statement that justify that the reporting entity does not meet this requirement,
    **OR**
  - If no information is found about this label, select the Supplementary Button: "No relevant information found in the statement".
- **Unclear,** in any other case:
  - Select this label if the information found is unclear or there are other concerns. If you decide to select this label, you have to choose the Compulsory Supplementary Button "Explanation" and justify why.

**Example 2.5.1.a** (BRP, SID#10056):

## SUPPLY CHAIN

BRP's main purchases from its suppliers include raw materials, tooling, parts and systems, information technology services, marketing and transportation services.

The Company has implemented a certification process to evaluate the suitability of potential suppliers, which includes a review of suppliers' financial condition and their capacity to produce components in conformity with BRP's requirements and specifications as well as with applicable labor and environmental standards. All suppliers must comply with applicable trade sanctions and the BRP Supplier Code of Conduct, which outlines a clear set of standards on ethical matters such as health and safety, environment as well as prevention of child labor and modern slavery. BRP's responsible supply chain is also supported by our ongoing procurement activities, which include site visits and audits.

More than 75% of BRP Australia's purchases are inter-company transactions. The remaining 25% is allocated to suppliers that support the business by providing services in relation to financial and legal activities, facilities maintenance, technology, marketing, sales support, transportation, customs and logistics, or offer consumables such as stationery and office supplies. Most of these suppliers are in Australia.

Telwater has over 220 suppliers, most of which provide the raw materials and components used in the manufacturing of aluminum boats and trailers, in addition to service suppliers like those detailed above. Most of these components are manufactured in China, Bahrain, Italy, New Zealand and the United States of America.

## RISK ASSESSMENT

BRP entities in Australia face different risks of modern slavery, depending on their activity.

**BRP Australia** being responsible for the sales, marketing and support to its distribution network has a very limited supply chain and considers the likelihood of modern slavery occurring in its operations to be low.

Some of the countries from which **Telwater** purchases material, parts and components, necessary for the manufacturing of boats, present an increased risk of modern slavery where the highest likelihood of slavery occurring would be in China.

BRP | MODERN SLAVERY STATEMENT

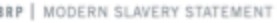

## ADDRESSING THE RISKS OF MODERN SLAVERY PRACTICES

At BRP, we are committed to conducting business in a responsible and ethical manner, commitment which is promoted through various undertakings.

### CODE OF ETHICS FOR EMPLOYEES

Our **Code of Ethics**, which applies to all members of BRP's Board of Directors, employees and business partners, promotes a culture of integrity and establishes principles governing relationships with suppliers, clients, employees and the public. Amongst other things, the values and rules set out in the Code encourage a safe, healthy and productive workplace.

### CODE OF CONDUCT FOR SUPPLIERS

Our **Supplier Code of Conduct** sets out BRP's expectations of its suppliers' conduct and addresses important topics such as child labour, compensation and working hours as well as harsh or inhumane treatment, and clearly states our expectations on forced or compulsory labour:

> *BRP will not engage in nor support the use of forced or compulsory labor. Supplier shall not use forced or compulsory labor in any of its facilities. All work must be voluntary and workers be free to leave work with reasonable notice. Supplier must not require that worked hand over government-issued identification, passports or work permits as a condition of employment.*

### SUSTAINABILITY

At BRP, we believe that the promotion of economic and social well-being is fundamental to the future success of our Company. Therefore, all BRP business units around the world integrate sustainability into their decisions, details of which can be found in our **Corporate Social Responsibility** report.

### CONFLICT MINERALS

BRP has no direct relationships with mines, smelters or refiners of gold, tin, tantalum or tungsten ("3TG"). As we are committed to having a global supply chain that follows the conflict mineral rules, we rely on our direct suppliers to provide information on the origin of the 3TGs contained in components and materials supplied to us and expect them to take similar measures with their suppliers to ensure a consistent policy alignment throughout the supply chain.

BRP | MODERN SLAVERY STATEMENT 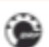

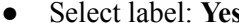

- Select label: **Yes**
- Add the supporting text: *BRP Australia being responsible for the sales, marketing and support to its distribution network has a very limited supply chain and considers the likelihood of modern slavery occurring in its operations to be low. Some of the countries from which Telwater purchases material, parts and components, necessary for the manufacturing of boats, present an increased risk of modern slavery where the highest likelihood of slavery occurring would be in China. // BRP has no direct relationships with mines, smelters or refiners of gold, tin, tantalum or tungsten ("3TG").*

**Example 2.5.1.b** (Tesserent, SID#10077):

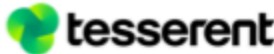

In this first reporting period, the working group commenced the design and formulation of the following new policies directed towards modern slavery risks in our supply chain:

**Modern Slavery Policy**

aims to confirm our commitment to human rights and mitigating risks of slavery and human trafficking in our operations and supply chain.

**Supplier Code of Conduct**

describes our minimum expectations of our suppliers in the areas of integrity, ethics and conduct, conflict of interest, health and safety and labour and human rights including the reduction of modern slavery practices.

We are in the process of drafting and refining both these policies for Board approval in FY23.

**DUE DILIGENCE, SUPPLIER ASSESSMENTS AND SUPPLIER MANAGEMENT**

As a primarily professional services focused organisation with 98% of our staff employed in either Australia or New Zealand, Tesserent's operations are generally considered to be low risk of slavery or human trafficking practices.

Our 400+ employees are subject to the relevant Australian or New Zealand labour laws including the Australian Fair Work Act 2009 (Cth) or the New Zealand Employment Rights Act 1996.

Tesserent has one operation outside of the Australia and New Zealand region limited to professional consulting services. These are considered low risk of modern slavery practices due to the high-skill, highly educated level of our employees.

Accordingly, as the major operational expenses involve the employment or engagement of staff in our offices in Australia or New Zealand to deliver high-skilled professional services, Tesserent has assessed the risk of modern slavery in its main operations as low.

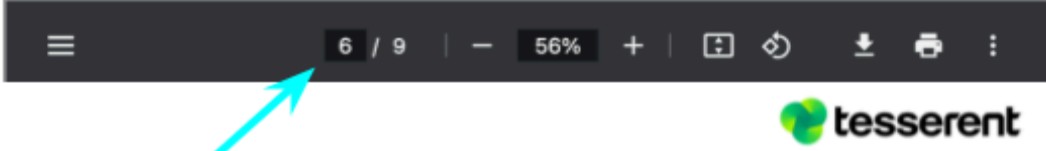

## SUPPLIER PROFILES

Tesserent recognises that some of its expenditure is directed towards suppliers that source goods and services from outside Australia, including some geographies that have a higher modern slavery risk profile. In this reporting period, Tesserent has undertaken a mapping exercise of our suppliers and identified the following supply chain categories, together with an initial assessment of their relative exposure to modern slavery risks:

| Supply Chain Category | Modern Slavery Risk Exposure |
|---|---|
| **IT Technology** (Software and Hardware) | Electrical goods and software development are recognised as high-risk industries globally. We procure from leading, globally recognised suppliers through their operations in Australia who have mature measures to reduce modern slavery risks in their supply chains, including independent auditing and public reporting of key measures and KPIs. Our initial assessment of these providers indicates they are taking adequate steps to identify and manage modern slavery risks. |
| **Telecommunications** (Carriage Services and Facilities) | These services and facilities are sourced from Tier-1 licenced telecommunications providers based in Australia or New Zealand that have their own modern slavery policies and procedures in place. Considered low risk. |
| **Business Premises and Facilities** (Rental, cleaning, and environmental management) | Cleaning and maintenance services are recognised as high-risk services globally. Tesserent typically uses the cleaning services provided by the building management at each of our Australian and New Zealand offices. Further due diligence of this category of our supply chain is planned during the next reporting period. |
| **External Professional Services** (Legal, Accounting, Taxation and Insurance.) | Considered low risk. Sourced from Australian and New Zealand based high-skilled professionals. |
| **Travel and Administration** (Staff Travel and Staff Amenities) | Considered low risk. Nearly all travel occurs either domestically within Australia or New Zealand on recognised major carries. |

## SUPPLIER ASSESSMENTS

Following the supplier mapping process undertaken in the current reporting period, Tesserent has commenced planning for the implementation of a supplier assessment and modern slavery risk screening process for all existing and new suppliers.

The aim of this assessment will be to better understand the extent of the potential risks relating to modern slavery practices throughout the Tesserent supply chain and how these can be better mitigated or eliminated. Once the risk level has been determined for an individual supplier, Tesserent is committed to undertake remedial actions as appropriate to manage our overall group modern slavery risk.

- Select label: **Yes**
- Add the supporting text: *As a primarily professional services focused organisation with 98% of our staff employed in either Australia or New Zealand, Tesserent's operations are generally considered to be low risk of slavery or human trafficking practices.* == // == *Tesserent has one operation outside of the Australia and New Zealand region limited to professional consulting services. These are considered low risk of modern slavery practices due to the high-skill, highly educated level of our employees. Accordingly, as the major operational expenses involve the employment or engagement of staff in our offices in Australia or New Zealand to deliver high-skilled professional services, Tesserent has assessed the risk of modern slavery in its main operations as low.*
- **Use the "Table" category in the "Visual Element Category" button, and add the following text in the corresponding text box**: *IT Technology (Software and Hardware)*
  *Electrical goods and software development are recognised as high-risk industries globally. We procure from leading, globally recognised suppliers through their operations in Australia who have mature measures to reduce modern slavery risks in their supply chains, including independent auditing and public reporting of key measures and KPIs. Our initial assessment of these providers indicates they are taking adequate steps to identify and manage modern slavery risks. Telecommunications*
  *These services and facilities are sourced from Tier-1 licenced*
  *(Carriage Services and Facilities) telecommunications providers based in Australia or New Zealand that have their own modern slavery policies and*
  *procedures in place. Considered low risk. Business Premises and Facilities*
  *(Rental, cleaning, and environmental management)*
  *Cleaning and maintenance services are recognised as high-risk services globally. Tesserent typically uses the cleaning services provided by the building management at each of our Australian and New Zealand offices.* == // == *External Professional Services Considered low risk. Sourced from Australian and New (Legal, Accounting, Taxation and Zealand based high-skilled professionals.*
  *Insurance.) Travel and Administration Considered low risk. Nearly all travel occurs either domestically (Staff Travel and Staff Amenities) within Australia or New Zealand on recognised major carries.*
- **Select the Supplementary Button:** "PDF page number" and add the page number: 6

**Example 2.5.1.c** (JFC, SID#10073):

## Risk assessment

In order to minimise modern slavery risks, we request that our major suppliers complete a questionnaire which contains questions relevant to modern slavery and human trafficking. Responses to this questionnaire will provide us with a greater understanding of the risk of modern slavery and human trafficking within a supplier's supply chains or operations. It will also enable us to work with a supplier to reduce any identified risks. We will actively improve our understanding and oversight of all tiers of our supply chains.

Whilst the majority of our products are supplied by headquarters in Japan, we recognise that certain sectors and industries, such as agriculture, are globally recognised as high risk industries. We will further strengthen our risk assessment program through analysis of our supply chain by product category, segment and production tier.

- Select label: **Yes**
- Add the supporting text: *Whilst the majority of our products are supplied by headquarters in Japan, we recognise that certain sectors and industries, such as agriculture, are globally recognised as high risk industries.*

**Example 2.5.1.d** (RetireAustralia, SID#10171):

## Modern Slavery Statement

### We enable a nimble and empowered workforce.
Providing quality service to our residents takes a richly skilled team, who ably demonstrate a deep sense of conviction in our Purpose and Vision. Our team members work dynamically and with simplicity to meet performance expectations while consistently upholding our Culture of Care and our values of Leadership, Integrity, Respect, One Team and Excellence, which are the foundation of all our actions, behaviours and decisions.

### We are growing an optimal portfolio of retirement communities.
We are developing new retirement communities in metropolitan areas while consistently optimising our established retirement communities to ensure they provide high-quality retirement lifestyles, with choices in care and support available to residents when needed. Our new villages are contemporary in design, integrated with the surrounding community and have age-friendly features making them perfect for now and ideal for later.

Within all of our operations, RetireAustralia is actively committed to acting ethically and with integrity in the engagement and management of all employees, contractors, volunteers and consultants and minimising the risk of Modern Slavery practices within our supply chains.

### Our Supply Chains
RetireAustralia uses the goods and services obtained through our supply chains to support our operations, including to provide relevant goods, carry out projects, professional advice, works, and services at our villages, development sites and offices.

In our past Financial Year, RetireAustralia's supply chains included approximately 1,340 vendors of goods and services, including:

- Corporate vendors delivering - professional services, IT, stationery and travel;
- Operational vendors delivering - grounds maintenance, cleaning, waste management and catering; and
- Property services vendors delivering - air conditioning, fire maintenance, security and valuation services.

Approximately 49% of the instances of the supply of goods and/or services from vendors occurred under formal contracts for the provision of those services.

The vast majority of RetireAustralia's vendors are located in Australia in Queensland, New South Wales and South Australia.

Only seven of RetireAustralia's vendors are located (or have a head office) outside of Australia. These vendors were based in New Zealand and United States of America.

==Of these foreign vendors four provide consulting services and three are software providers. Together they only accounted for a total supply of $187,750 in goods and services. None of the goods and/or services provided by these vendors are considered high risk for Modern Slavery.==

### 3. Risks:

*Section 16(1)(c) of the Modern Slavery Act 2018 requires modern slavery statements to describe the risks of modern slavery practices in the operations and supply chains of the reporting entity and any entities it owns and controls.*

Commonwealth Modern Slavery Act 2018 - The Guidance for Reporting Entities explains that Modern Slavery refers to situations where victims are subjected to coercion, threats, deception or exploitation to undermine their freedom, for the perpetrator's gain. For example, Modern Slavery includes trafficking in persons, slavery, servitude, forced labour, forced marriage, debt bondage, exploitation of child labour, deceptive recruiting for labour or services.

Version: 1
Implemented by: 30 September 2022

Page **6**

**Modern Slavery Statement**

It is intended that Modern Slavery describes forms of serious exploitation, where a worker is deprived of their personal freedom. It does not necessarily include situations where workers are subjected to poor working conditions or underpaid. However, these circumstances may escalate into Modern Slavery or provide an indication that Modern Slavery is also taking place.

### Risks in RetireAustralia's operations and supply chains

RetireAustralia has not identified any instances of Modern Slavery in our supply chains in the last Financial Year.

Despite this, RetireAustralia remains alert to the potential risk of Modern Slavery in our supply chains and monitors and regularly reviews our supply chain for Modern Slavery risks.

RetireAustralia performed a high-level risk assessment of our supply chain from May to June 2022. This assessment took into consideration the volume and category of goods and or services procured, as well as the geographic location of our vendors and of their manufacturers for goods and services we purchase from them.

To identify the areas of highest risk, we considered the following risk indicators (table below) for the potential occurrence of Modern Slavery:

| Risk Indicator | Description |
|---|---|
| Industry sector | Certain industries may have a high prevalence of Modern Slavery because of their characteristics, products and processes. For example – manufacturing, cleaning, construction, waste management, food, hospitality, uniforms, PPE and security. |
| Product and services | Certain products and services may have a high prevalence for Modern Slavery because of the way they are produced or used. For example – use of under paid and over worked labour, child labour and forced labour. |
| Geographic location | Certain countries have a high prevalence of Modern Slavery due to poor government structures, corruption, weak labour laws, conflict and or political instability, migration flow and poverty. |
| Supply chain model | Where supply chain models are larger and more complex, they are more prevalent to carry a higher risk of Modern Slavery. For example – levels of subcontracting, recruiter fees, lack of supply chain processes, poor audit trails and lack of transparency. |

Based on these risk indicators and analysis of our vendors during the risk assessment period, RetireAustralia has identified that 99% of vendors have Australian supply origins.

### 4. Actions:

*Section 16(1)(d) of the Modern Slavery Act 2018 requires reporting entities to describe the actions they have taken to assess and address the risks outlined in the previous section, including due diligence and remediation processes.*

RetireAustralia is committed to ensuring that in providing the services it offers it minimises the risk of Modern Slavery being present in our supply chains. We strive to achieve this through our Procurement Framework and ensuring that goods and services purchased are provided by vendors who respect and adhere to appropriate human rights standards.

- Select label: **Yes**
- Add the supporting text: *Of these foreign vendors four provide consulting services and three are software providers. Together they only accounted for a total supply of $187,750 in goods and services. None of the goods and/or services provided by these vendors are considered high risk for Modern Slavery.* // *RetireAustralia has not identified any instances of Modern Slavery in our supply chains in the last Financial Year.*
- **Notes:**
  - The sentence **"***RetireAustralia has not identified any instances of Modern Slavery in our supply chains in the last Financial Year.***"** is extracted because it is a **negative declaration** related to **existing or past cases of modern slavery.**
  - The text in the table should not be extracted because it describes the "risk indicators" they considered in their risk analysis: it does not correspond to the actual risks they identified.

## 2.5.2. Risk mitigation

**Question:** does the reporting entity describe the actions applied to identify, assess, and mitigate the modern slavery risks it identified?

**Background:** after describing the risks they face, reporting entities are required to describe how these risks are **mitigated**. This is done by describing the actions taken by the reporting entity to identify and assess risks as well as to avoid causing or contributing to modern slavery.

Mitigation actions may include:

- Participating in or launching **industry initiatives** (e.g. UN Global Compact, or Finance Against Slavery and Trafficking Initiative), or by providing financial support to such initiatives;
- Requiring suppliers and third-party entities to comply with internal, regional, or international **policies or labor laws**;
- Requiring employees, suppliers, or other stakeholders to follow **training sessions** on modern slavery;
- Adopting or updating **internal policies, frameworks or standards** related to modern slavery, for instance:
  - Adopting formal processes used to identify and assess modern slavery risks. These processes may include:
    - Conducting desk research by reviewing secondary sources of information (e.g. reports from international organizations, frameworks such as the UN Guiding Principles, or research papers and articles on modern slavery) or by reviewing external expert advisers' inputs;
    - Utilizing risk management tools or software (e.g. SEDEX);
    - Conducting risk-based questionnaires, surveys, or interviews with employees, suppliers, or other stakeholders;
    - Auditing, screening, or directly engaging with employees, suppliers, or other stakeholders.
  - **Committing** to paying all employees, migrant workers, temporary workers, and third-party workers a "**living wage**", to support their **freedom of association (or unionization)**, or to support **collective bargaining**;
  - Adopting a **code of conduct** or **code of ethics** that relates to combatting or preventing modern slavery and forced labour practices;
  - Incorporating provisions for onboarding and engaging with suppliers, including **contract clauses and requirements,** and extending the code of conduct to cover suppliers;

- ○ Enforcing **responsible recruitment** of workers (e.g., by not allowing the payment of recruitment fees by workers, or by not withholding part of their compensation for housing or licensing costs);
- ○ Implementing **responsible procurement practices** when establishing new supply chains (e.g., no excessive pressure on lead times for products or services they source);
- ○ Having a **zero-tolerance policy** that would force the entity to take action regarding threats, regarding intimidation and attacks against human rights defenders, or regarding modern slavery cases tied to their suppliers;
- ○ Establishing a **whistleblowing policy** that encourages and safeguards workers, employees, suppliers, or other stakeholders when reporting concerns;
- ○ Ensuring the application of these policies or frameworks by having an executive or board member participate in their elaboration or in oversight committees.

**Notes:**

- Vague language stating that the reporting entity has "*zero tolerance*" towards modern slavery without mentioning a policy in this regard, or that it "*is committed to fight*" modern slavery is very common. However, such declarations are **not evidence** of risk mitigation by themselves. These declarations should be linked with an actual company policy or linked to industrial initiatives or certifications frameworks. If no other relevant information is found, the correct label in statements containing only such declarations is **NO**.
- Not all policies adopted by an entity may be relevant to modern slavery. If a policy does not explicitly cover modern slavery (which includes forced labour, human trafficking, child labour, etc.), then the entity should itself detail why it is relevant in this context for it to be considered evidence of a risk mitigation action. Only extract policy-related sentences if the policy is truly linked to modern slavery practices somewhere in the statement.

**Label options & their associated supporting text passages extracted from the statements:**

- ○ **Yes,** the statement describes one or more risk mitigation actions:
  - Copy-paste the text passages from the statement that describe the actions taken to mitigate the identified risks.
  - If any relevant information comes in other formats than text, fill in the required information via the Supplementary Button: "Visual Element Category", note the page where it is found, and extract any relevant text it contains (if possible).
- ○ **No,** the statement does not describe any risk mitigation action:
  - Copy-paste the text passages from the statement that justify that the reporting entity does not meet this requirement,
    **OR**
  - If no information is found about this label, select the Supplementary Button: "No relevant information found in the statement".

- ○ **Unclear,** in any other case:
  - ● Select this label if the information found is unclear or there are other concerns. If you decide to select this label, you have to choose the Compulsory Supplementary Button "Explanation" and justify why.

**Example 2.5.2.a** (Webjet, SID#10084), a case where the company's proposed "Code of Conduct" which applies to all employees is NOT relevant to modern slavery, and therefore should NOT be extracted.

**Actions to assess and address modern slavery risks**

(continued)

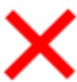

In May 2022, we launched a new global Code of Conduct, which applies to all employees, directors, contractors and everyone working at Webjet or its subsidiaries, regardless of location. The new code embeds our values, prioritises the safety and wellbeing of our people, and highlights our commitment to creating and maintaining a diverse and inclusive workforce and ensuring that laws and ethical standards are complied with and the human rights of our people and those of our customers, suppliers and business partners are respected and upheld.

**Our people**

All Webjet employees are employed in accordance with the applicable labour laws of the country in which they are employed, including in respect of minimum wages, hours of work, leave entitlements and safe working conditions. In addition, all employees have access to our Employee Assistance Program which provides employees with paid access to

have a supplier relationship with us. During the Reporting Period, we followed up on our findings by requesting the remaining 16 medium-risk suppliers to complete a modern slavery self-assessment questionnaire. While the results suggest a low level of modern slavery governance amongst some of the suppliers we contacted, the process of completing the questionnaire had the effect of raising modern slavery awareness amongst many of our suppliers. We have also identified opportunities to improve upon the supplier self-assessment process to produce more meaningful results in future assessments.

We have introduced provisions relating to modern slavery compliance in our template accommodation provider agreement, B2B client agreement, car and motorhome rental provider agreement and standard services agreement, and have sought to incorporate these provisions into other high-risk supplier contracts, where possible. These provisions require our suppliers to:

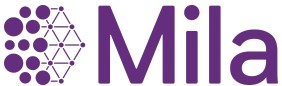
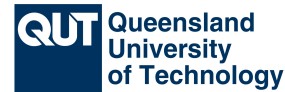

**Example 2.5.2.b** (Landis+Gyr, SID#10095):

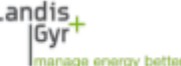

## LANDIS+GYR AUSTRALIA - JOINT MODERN SLAVERY STATEMENT 2022

### 1. Introduction and scope

This Joint Modern Slavery Statement is made in accordance with the Modern Slavery Act 2018 (Cth) (Australia) and applies to the operations of Landis & Gyr Holdings Pty Ltd, Bayard Metering Pty Ltd and Landis & Gyr Pty Ltd (together, **Landis+Gyr Australia, our, we**) during our financial year 1 April 2021 to 31 March 2022.

In this statement, we outline our processes for assessing, preventing, and addressing modern slavery risks in our business operations and supply chains.

In preparing this statement, we have consulted with our key business leaders across the Landis+Gyr Group supply chain teams, procurement teams, operations and human resources teams, compliance and legal teams, the Asia Pacific business teams, and the Board of Directors of each of the three reporting entities.

### 2. Landis+Gyr Group

Landis+Gyr Australia is part of the Landis+Gyr Group, which is headquartered in Switzerland. Our ultimate parent company, Landis+Gyr Group AG (SIX: LAND), together with its subsidiary and affiliate businesses, (**Landis+Gyr Group**) employs approximately 5000 employees worldwide and has a presence in 30 countries, across five continents with the mission of helping the world manage energy better.

The Landis+Gyr Group is a leading global provider of integrated energy management solutions for the utility sector. Offering one of the broadest portfolios, the Landis+Gyr Group delivers innovative and flexible solutions to help utilities solve their complex challenges in Smart Metering, Grid Edge Intelligence and Smart Infrastructure.

The Landis+Gyr Group designs, manufactures, and supplies a broad portfolio of products for the energy industry, including smart meters for electricity, heat, gas, and water, together with energy management solutions such as demand side management solutions, advanced sensing and automation tools, EV hardware and software charging solutions, cybersecurity solutions and data analytics.

Our products and services help drive positive and sustainable environmental outcomes, by enabling better energy infrastructure management, which in turn improves energy efficiency, energy conservation and decarbonisation.

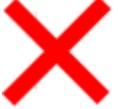 The Landis+Gyr Group operates to high standards of ethical conduct, and environmental, social and corporate governance (**ESG**). Our leaders, managers, and employees are also expected to conduct themselves with the highest level of ethics and integrity.

1

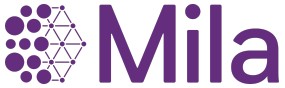
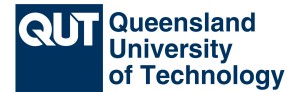

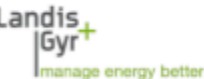

## 3. Organisational structure and operations

Landis+Gyr Australia is led by Mr. David Maclean, the Vice-President for Australia, New Zealand and South East Asia. Mr. Maclean reports to the Executive Vice President, Steve Jeston, for Landis+Gyr Asia Pacific, who in turn reports to the CEO, Werner Lieberherr, of the Landis+Gyr Group.

The Landis+Gyr Australia operations include manufacturing, supply chain management, research and development (**R&D**), product and solution delivery management, services delivery management together with sales, professional support services and after-sales support.

The core products and services supplied by Landis+Gyr Australia to its customers include gas, electricity and water metering products, demand side management tools, technology solutions, software, energy management solutions and related support services.

We operate from four locations across Australia including Brisbane, Sydney, Perth, and Melbourne. Our Australian and Asia Pacific head office is located in Sydney, together with our R&D centre, where our team of engineers development components of the products and solutions sold to our Australian customers. Our main manufacturing facilities are located in Melbourne, where our staff carry out assembly, testing, calibration, configuration and certification work for a suite of Landis+Gyr water, gas and electricity meters.

Our operations are certified to the following ISO standards: ISO9001, ISO45001, ISO14001, ISO17025 and recently ISO22301.

## 4. Supply Chains

Our supply chains are diverse, consisting of a mixture of components, sub-assemblies, and finished goods, relating to our gas, electricity and water meter products and solutions.

We source most of our direct materials (such as component parts and sub-assemblies) from suppliers in China, Vietnam, Europe, and the USA. We also source various products and services from within the Landis+Gyr Group, for example IT software and solutions from our sister companies in the USA and India, and various meter products and sub-assemblies manufactured by our sister company in China.

## 5. Risks of modern slavery

We recognise the significant global challenge associated with modern slavery and the inherent risks of modern slavery in supply chains.

Based on internal processes and procedures, ongoing supplier due diligence and risk assessments, as well as other factors, we have identified the risk of Modern Slavery across our Asia Pacific business, including Australia, in our internal and supply chain operations as high impact and medium probability.

## 6. Policies and procedures to identify and manage modern slavery risks

The Landis+Gyr Group has implemented detailed policies and procedures to help identify and manage the risks of modern slavery both within our own business operations and within our Tier 1 and Tier 2 supply chains. These policies and procedures are reviewed on an annual basis.

2

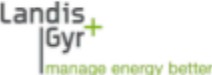

### Landis+Gyr Group Employee Code of Business Ethics and Conduct

The Landis+Gyr Group Employee Code of Business Ethics and Conduct is a cornerstone document within our business. It guides our internal employment processes and helps to ensure fair work standards for all employees. The code applies to all employees and workers across the 30 countries in which the Landis+Gyr Group operates. The code requires us to ensure that:

- employees are free to choose their employment and are not forced into any involuntary labour;
- employees have freedom of association and freedom to join trade unions;
- employees receive wages and benefits in compliance with local labour law requirements;
- there is no unlawful harassment or workplace discrimination;
- there is no harsh or inhumane treatment within the workplace; and
- we never use child labour in any operations or at any stage of the manufacturing process.

### Corporate Social Responsibility Directive

This Directive defines the commitment by Landis+Gyr Group's senior management to manage ESG in accordance with industry best practice. The Directive addresses corporate governance, business conduct, employment practices, health and safety, environmental practices, and supplier management.

### UN Global Compact, UN Declaration of Human Rights and ILO Declaration of Fundamental Rights at Work

The Landis+Gyr Group joined the UN Global Compact in 2020. Our fair labour standards are aligned with the principles set by the UN Global Compact, as well as the global standards and guidelines set by the United Nations Universal Declaration of Human Rights and the Declaration on the Fundamental Principles and Rights at Work of the International Labor Organization (ILO).

### Landis+Gyr Group SpeakUp and Whistle Blowers Policies

The Landis+Gyr Group encourages all its employees, customers, suppliers, and other business partners to report any concerns they may have in relation to the Landis+Gyr business activities, or its supply chains, including any risks associated with slavery or human trafficking via our 'Speak Up' system. Employees, customers, suppliers, or others who have concerns can use our confidential Speak Up helpline or submit their compliant by contacting a third-party Ombudsperson. In addition, our Australian Whistleblower Policy ensures that employees, suppliers, and business partners can raise concerns, anonymously if they wish, about any misconduct or improper state of affairs within our organisation.

### Employment Equity Discrimination Harassment and Bullying Practices

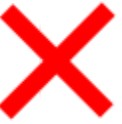 As detailed in our Code of Business Ethics and Conduct we are committed to providing a safe and healthy working environment for our workers, ensuring they feel valued and respected. This policy is designed to prevent discrimination, harassment, bullying and victimisation and enable workers to carry out their work in a safe environment.

### Occupational health and safety policies

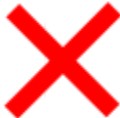 Our workplace health and safety policies help ensure that our business complies with its legal

3

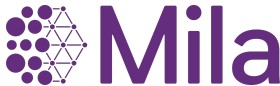

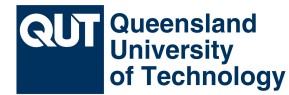

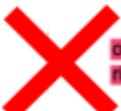

obligations under relevant workplace health and safety legislation and proactively seeks to reduce the risk of workplace accidents.

**New Supplier Induction Process**

Landis+Gyr Group follows a 'New Supplier Induction' (**NSI**) process for all new suppliers, which includes a rigorous process for vendor selection and management.

As part of the NSI process, the Landis+Gyr Group supply chain and quality teams take a risk-based approach to determine the appropriate scope of supplier due diligence for new vendors. The supplier due diligence is then completed before entering into a supply arrangement with a new vendor. For Tier 1 supplier of direct materials, the due diligence often involves supplier factory audits and supplier interviews by our quality control and procurement team. We also require new suppliers of direct materials to complete a detailed supplier questionnaire. Suppliers are required to demonstrate that they provide safe working conditions, treat workers with dignity and respect, and act ethically and within the law in their use of labour. Finally, our suppliers (of both direct materials and indirect materials or services) as subject to an online screening via a third-party tool prior to onboarding. The screening covers sanctions, watch lists, and enforcements, as well as adverse media—these screenings are thus relevant for modern slavery risks as well.

**Supplier Code of Conduct**

Landis+Gyr uses the Supplier Code of Conduct agreements to define and agree the level of corporate social responsibility controls with its Tier 1 suppliers (being businesses that supply products or services directly to any Landis+Gyr Group entity). We require all Tier 1 suppliers of direct materials to sign up to the Landis+Gyr Code of Conduct for Suppliers. This code sets out onerous requirements relating to modern slavery compliance, environmental compliance, fair employment standards, and a workplace free of harassment and unlawful discrimination. We require our suppliers to ensure that their suppliers and subcontractors also comply with the code. Failure to comply with the Code of Conduct for Suppliers may result in the Supplier's removal from the Landis+Gyr approved supplier base.

**Supplier audit programme**

The Landis+Gyr Group supplier quality and procurement teams conduct detailed onsite audits of many of our Tier 1 suppliers (suppliers with direct contractual commitments to Landis+Gyr) on an annual basis, following a risk-based approach. Our supplier audit programme is rigorous and aligned with the Responsible Business Alliance (**RBA**) code of conduct requirements, which addresses and seeks to minimise the risks of modern slavery and human trafficking. Our internal auditors have received training on the RBA code of conduct requirements.

The Landis+Gyr Group takes a risk-based approach to management of modern slavery with Tier 2 suppliers (being the businesses that supply products or services to our Tier 1 suppliers). Certain Tier 2 suppliers (based on risk profile) are required to sign the Landis+Gyr Supplier Code of Conduct and are subject to a Landis+Gyr supplier audit.

Unfortunately, COVID-19 has significantly impacted our audit team's ability to travel and perform onsite audits with suppliers in 2021 and 2022. However, Landis+Gyr was able to leverage our Landis+Gyr team based in China to support local based supplier audits. In conjunction, Landis+Gyr has invested in the purchase of HoloLens™ technology. This has enabled us to perform virtual audits remotely at other key suppliers, that we were not able to access due to COVID-19 travel restrictions. Landis+Gyr was able to perform key supplier audits during this reporting year, using a combination of

4

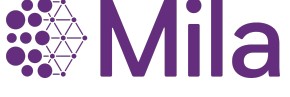

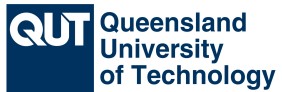

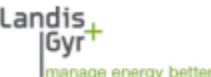

onsite and virtual auditing techniques. These were a combination of Tier 1 and Tier 2 audits. There were zero non-conformances identified related to modern slavery as an outcome of these audits.

### Green Procurement Requirements

Landis+Gyr actively strives to prevent the use of Conflict Minerals within our products and requires that its suppliers and partners share this objective. Landis+Gyr has adopted the OECD Due Diligence Guidance for Responsible Supply Chains of Minerals from Conflict-Affected and High-Risk Areas. Suppliers are requested to design and implement a system for the thorough management of those materials, components and processes which may have negative environmental impact. A special focus lies on chemical substances in products, including the following actions:

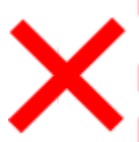

- Establishment of a design approach to pursue resource sustainability (green design), for example: reduce material usage, design assemblies to be easily disassembled at end of life to maximise recyclability.
- Establishment of a system for identification and management of hazardous chemical substances in products.
- Procurement of parts, components, and materials with minimal environmental impacts, including a reduction in the use of hazardous chemical substances.

Landis+Gyr Group collects supplier declarations regarding compliance with our Conflict Minerals requirements, in accordance with a pre-defined plan as set out in the Green Procurement Annex to the Supplier Code of Conduct.

### Employee training plan

Landis+Gyr continues to conduct mandatory modern slavery training program for management, human resources, procurement, and supply chain personnel located in Australia and across Asia Pacific. The training scope included: (i) modern slavery risk identification, assessment, mitigation, and management; (ii) tools and processes used by Landis+Gyr to identify and mitigate modern slavery risks; and (iii) reporting and legal requirements under the Modern Slavery Act. The training served to raise awareness of Modern Slavery risks and the various internal tools and processes utilized by Landis+Gyr to identify, mitigate and manage modern slavery risks in its supply chains and operations.

In addition, Landis+Gyr Group office-based employees are required to undertake annual mandatory Code of Conduct training, which for fiscal year 2021-2022 was completed in November 2021.

We continue to run annual refresher training on modern slavery for our management, Human Resources and Procurement and Supply Chain personnel located in Australia and across the Asia Pacific region.

## Approval

This statement was approved by principal governing body (being the board of directors) of each of the 3 reporting entities, namely Landis & Gyr Holdings Pty Ltd, Bayard Metering Pty Ltd and Landis & Gyr Pty Ltd. The board of directors of these 3 reporting entities approved this statement on 20 September 2022.

5

- Select label: **Yes**
- Add the supporting text: *The Landis+Gyr Group has implemented detailed policies and procedures to help identify and manage the risks of modern slavery both within our own business operations and within our Tier 1 and Tier 2 supply chains. These policies and procedures are reviewed on an annual basis.* // *The Landis+Gyr Group Employee Code of Business Ethics and Conduct is a cornerstone document within our business. It guides our internal employment processes and helps to ensure fair work standards for all employees. The code applies to all employees and workers across the 30 countries in which the Landis+Gyr Group operates. The code requires us to ensure that:*

*• employees are free to choose their employment and are not forced into any involuntary labour;*

*• employees have freedom of association and freedom to join trade unions;*

*• employees receive wages and benefits in compliance with local labour law requirements;* // *• we never use child labour in any operations or at any stage of the manufacturing process.* // *The Landis+Gyr Group joined the UN Global Compact in 2020. Our fair labour standards are aligned with the principles set by the UN Global Compact, as well as the global standards and guidelines set by the United Nations Universal Declaration of Human Rights and the Declaration on the Fundamental Principles and Rights at Work of the International Labor Organization (ILO).* // *The Landis+Gyr Group encourages all its employees, customers, suppliers, and other business partners to report any concerns they may have in relation to the Landis+Gyr business activities, or its supply chains, including any risks associated with slavery or human trafficking via our 'Speak Up' system. Employees, customers, suppliers, or others who have concerns can use our confidential Speak Up helpline or submit their complaint by contacting a third-party Ombudsperson. In addition, our Australian Whistleblower Policy ensures that employees, suppliers, and business partners can raise concerns, anonymously if they wish, about any misconduct or improper state of affairs within our organisation.* // *Landis+Gyr Group follows a 'New Supplier Induction' (NSI) process for all new suppliers, which*

*includes a rigorous process for vendor selection and management.*

*As part of the NSI process, the Landis+Gyr Group supply chain and quality teams take a risk-based approach to determine the appropriate scope of supplier due diligence for new vendors. The supplier due diligence is then completed before entering into a supply arrangement with a new vendor. For Tier 1 suppliers of direct materials, the due diligence often involves supplier factory audits and supplier interviews by our quality control and procurement team. We also require new suppliers of direct materials to complete a detailed supplier questionnaire. Suppliers are required to demonstrate that they provide safe working conditions, treat workers with dignity and respect, and act ethically and within the law in their use of labour. Finally, our suppliers (of both direct materials and indirect materials or services) as subject to an online*

*screening via a third-party tool prior to onboarding. The screening covers sanctions, watch lists, and enforcements, as well as adverse media—these screenings are thus relevant for modern slavery risks as well. // Landis+Gyr uses the Supplier Code of Conduct agreements to define and agree the level of corporate social responsibility controls with its Tier 1 suppliers (being businesses that supply products or services directly to any Landis+Gyr Group entity). We require all Tier 1 suppliers of direct materials to sign up to the Landis+Gyr Code of Conduct for Suppliers. This code sets out onerous requirements relating to modern slavery compliance, environmental compliance, fair employment standards, and a workplace free of harassment and unlawful discrimination. We require our suppliers to ensure that their suppliers and subcontractors also comply with the code. // The Landis+Gyr Group supplier quality and procurement teams conduct detailed onsite audits of many of our Tier 1 suppliers (suppliers with direct contractual commitments to Landis+Gyr) on an annual basis, following a risk-based approach. Our supplier audit programme is rigorous and aligned with the Responsible Business Alliance (RBA) code of conduct requirements, which addresses and seeks to minimise the risks of modern slavery and human trafficking. Our internal auditors have received training on the RBA code of conduct requirements.*

*The Landis+Gyr Group takes a risk-based approach to management of modern slavery with Tier 2 suppliers (being the businesses that supply products or services to our Tier 1 suppliers). Certain Tier 2 suppliers (based on risk profile) are required to sign the Landis+Gyr Supplier Code of Conduct and are subject to a Landis+Gyr supplier audit.*

*Unfortunately, COVID-19 has significantly impacted our audit team's ability to travel and perform onsite audits with suppliers in 2021 and 2022. However, Landis+Gyr was able to leverage our Landis+Gyr team based in China to support local based supplier audits. In conjunction, Landis+Gyr has invested in the purchase of HoloLensTM technology. This has enabled us to perform virtual audits remotely at other key suppliers, that we were not able to access due to COVID-19 travel restrictions. Landis+Gyr was able to perform key supplier audits during this reporting year, using a combination of // onsite and virtual auditing techniques. These were a combination of Tier 1 and Tier 2 audits. // Landis+Gyr actively strives to prevent the use of Conflict Minerals within our products and requires that its suppliers and partners share this objective. Landis+Gyr has adopted the OECD Due Diligence Guidance for Responsible Supply Chains of Minerals from Conflict-Affected and High-Risk Areas. // Landis+Gyr Group collects supplier declarations regarding compliance with our Conflict Minerals requirements, in accordance with a pre-defined plan as set out in the Green Procurement Annex to the Supplier Code of Conduct. // Landis+Gyr continues to conduct mandatory modern slavery training program for management, human resources, procurement, and supply chain personnel located in Australia and across Asia Pacific. The training scope included: (i) modern slavery risk identification, assessment, mitigation, and management; (ii) tools and processes used by Landis+Gyr to identify and mitigate modern slavery risks; and (iii) reporting and legal*

*requirements under the Modern Slavery Act. The training served to raise awareness of Modern Slavery risks and the various internal tools and processes utilized by Landis+Gyr to identify, mitigate and manage modern slavery risks in its supply chains and operations.*

*In addition, Landis+Gyr Group office-based employees are required to undertake annual mandatory Code of Conduct training, which for fiscal year 2021-2022 was completed in November 2021.*

*We continue to run annual refresher training on modern slavery for our management, Human Resources and Procurement and Supply Chain personnel located in Australia and across the Asia Pacific region.*

- **Note that the text highlighted in red should not be extracted because it is related to corporate policies that are not directly related to modern slavery.**

**Examples 2.5.2.c** (BRP, SID#10056):

## SUPPLY CHAIN

BRP's main purchases from its suppliers include raw materials, tooling, parts and systems, information technology services, marketing and transportation services.

The Company has implemented a certification process to evaluate the suitability of potential suppliers, which includes a review of suppliers' financial condition and their capacity to produce components in conformity with BRP's requirements and specifications as well as with applicable labor and environmental standards. All suppliers must comply with applicable trade sanctions and the BRP Supplier Code of Conduct, which outlines a clear set of standards on ethical matters such as health and safety, environment as well as prevention of child labor and modern slavery. BRP's responsible supply chain is also supported by our ongoing procurement activities, which include site visits and audits.

More than 75% of BRP Australia's purchases are inter-company transactions. The remaining 25% is allocated to suppliers that support the business by providing services in relation to financial and legal activities, facilities maintenance, technology, marketing, sales support, transportation, customs and logistics, or offer consumables such as stationery and office supplies. Most of these suppliers are in Australia.

Telwater has over 220 suppliers, most of which provide the raw materials and components used in the manufacturing of aluminum boats and trailers, in addition to service suppliers like those detailed above. Most of these components are manufactured in China, Bahrain, Italy, New Zealand and the United States of America.

## RISK ASSESSMENT

BRP entities in Australia face different risks of modern slavery, depending on their activity.

**BRP Australia** being responsible for the sales, marketing and support to its distribution network has a very limited supply chain and considers the likelihood of modern slavery occurring in its operations to be low.

Some of the countries from which **Telwater** purchases material, parts and components, necessary for the manufacturing of boats, present an increased risk of modern slavery where the highest likelihood of slavery occurring would be in China.

BRP | MODERN SLAVERY STATEMENT

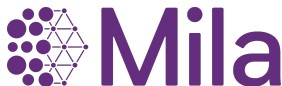
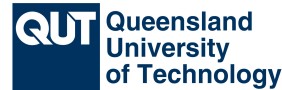

## ADDRESSING THE RISKS OF MODERN SLAVERY PRACTICES

At BRP, we are committed to conducting business in a responsible and ethical manner, commitment which is promoted through various undertakings.

### CODE OF ETHICS FOR EMPLOYEES

❌ Our **Code of Ethics**, which applies to all members of BRP's Board of Directors, employees and business partners, promotes a culture of integrity and establishes principles governing relationships with suppliers, clients, employees and the public. Amongst other things, the values and rules set out in the Code encourage a safe, healthy and productive workplace.

### CODE OF CONDUCT FOR SUPPLIERS

Our **Supplier Code of Conduct** sets out BRP's expectations of its suppliers' conduct and addresses important topics such as child labour, compensation and working hours as well as harsh or inhumane treatment, and clearly states our expectations on forced or compulsory labour:

> BRP will not engage in nor support the use of forced or compulsory labor. Supplier shall not use forced or compulsory labor in any of its facilities. All work must be voluntary and workers be free to leave work with reasonable notice. Supplier must not require that worked hand over government-issued identification, passports or work permits as a condition of employment.

### SUSTAINABILITY

At BRP, we believe that the promotion of economic and social well-being is fundamental to the future success of our Company. Therefore, all BRP business units around the world integrate sustainability into their decisions, details of which can be found in our **Corporate Social Responsibility** report.

### CONFLICT MINERALS

BRP has no direct relationships with mines, smelters or refiners of gold, tin, tantalum or tungsten ("3TG"). As we are committed to having a global supply chain that follows the conflict mineral rules, we rely on our direct suppliers to provide information on the origin of the 3TGs contained in components and materials supplied to us and expect them to take similar measures with their suppliers to ensure a consistent policy alignment throughout the supply chain.

BRP | MODERN SLAVERY STATEMENT   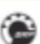

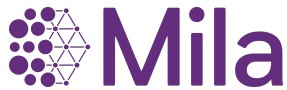
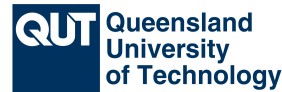

## OUTCOMES AND MEASURING EFFECTIVENESS

### COMES OF NON-COMPLIANCE

If a practice that is not compliant with BRP's Supplier Code of Conduct is identified, we expect and support suppliers to take action to correct it. If a supplier does not take effective action in a timely manner, appropriate sanctions will be imposed, up to potential termination of the business relationship with BRP.

### MEASURING EFFECTIVENESS THROUGH OUR WHISTLEBLOWING HOTLINE

Employees but also suppliers are encouraged to report any concerns they may have, including situations of possible modern slavery, either to management or through our anonymous whistleblower hotline, EthicsPoint (https://secure.ethicspoint.eu/domain/media/en/gui/100215/index.html). Any allegation brought forward is reviewed and investigated when required. In the past year, there have been no issues raised internally or externally concerning modern slavery.

## FURTHER STEPS TO PREVENT MODERN SLAVERY

BRP will continue to work collaboratively with its suppliers to ensure the risks of Modern Slavery are managed effectively. To that end, our plans for its next financial year are to:

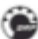

- Launch our new employee Code of Ethics which includes a dedicated section on human rights;
- Update our supplier Code of Conduct and expand its coverage of modern slavery;
- Conduct a CSR survey to qualify and monitor our suppliers on various topics including human rights and labor.

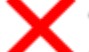 MODERN SLAVERY STATEMENT

- Select label: **Yes**
- Add the supporting text: *The Company has implemented a certification process to evaluate the suitability of potential suppliers,*
  *which includes a review of suppliers' financial condition and their capacity to produce components*
  *in conformity with BRP's requirements and specifications as well as with applicable labor and environmental standards. All suppliers must comply with applicable trade sanctions and the BRP Supplier Code of Conduct, which outlines*
  *a clear set of standards on ethical matters such as health and safety, environment as well as*

*prevention of child labor and modern slavery. BRP's responsible supply chain is also supported by our ongoing procurement activities,*

*which include site visits and audits.* // *Our Supplier Code of Conduct sets out BRP's expectations of its suppliers' conduct and addresses*

*important topics such as child labour, compensation and working hours as well as harsh or inhumane treatment, and clearly states our expectations on forced or compulsory labour:*

*BRP will not engage in nor support the use of forced or compulsory labor. Supplier shall not use forced or compulsory labor in any of its facilities. All work must be voluntary and workers*

*be free to leave work with reasonable notice. Supplier must not require that worked hand over government-issued identification, passports or work permits as a condition of employment.* // *As we are committed to having a global supply chain that follows the conflict mineral rules, we rely on our direct suppliers to provide information on the origin of the 3TGs contained*

*in components and materials supplied to us and expect them to take similar measures with*

*their suppliers to ensure a consistent policy alignment throughout the supply chain.* // *Employees but also suppliers are encouraged to report any concerns they may have, including situations of possible modern slavery, either to management*

*or through our anonymous whistleblower hotline, EthicsPoint (https://secure.ethicspoint.eu/domain/media/en/gui/100215/index.html).* // *• Update our supplier Code of Conduct and expand its coverage of modern slavery;*

*• Conduct a CSR survey to qualify and monitor our suppliers on various topics including human rights and labor.*

- **Note that the text related to the "Code of Ethics" is not extracted because it is not directly related to modern slavery.**

## 2.5.3. Modern slavery remediation

**Question:** does the reporting entity describe remediation actions for modern slavery cases?

**Background:** if one or more **cases** (or, synonymously, incidents, issues, or situations) have been declared by the reporting entity where it caused or contributed to modern slavery, the statement should describe the actions taken to **remediate** (or, synonymously, manage, address, or remedy) these cases. If no modern slavery case has been identified by the reporting entity, it may still describe an action plan used to remediate hypothetical cases if one should occur in the future. This is relevant information for this requirement.

The supporting text for this criteria should clearly refer to actions proposed by the reporting entity to remediate modern slavery cases, and not to actions proposed to mitigate the risks of modern slavery. The remediation actions could ensure that the harm that was caused cannot recur, they could include formal apologies and compensation, and they could lead to a stop in certain activities of the reporting entity.

Specifically, examples of corrective actions and sanctions to remediate modern slavery cases include:
- **Inquiries and investigations** into instances of modern slavery involving stakeholders;
- **Worker assistance** (e.g. giving back passports or helping file legal claims);
- **Compensation** (e.g. paying owed wages, penalties, or indemnities);
- **Issuing formal apologies**;
- **Notifying** management, authorities, or other organizations of incidents;
- **Stopping business activities** with partners or suppliers who are in violation (i.e., by canceling or terminating contracts).

**Label options & their associated supporting text passages extracted from the statements:**

- **Yes,** the statement describes one or more remediation actions:
  - Copy-paste the text passages from the statement that describe the actions taken (or that would be taken) to remediate cases of modern slavery.
  - If any relevant information comes in other formats than text, fill in the required information via the Supplementary Button: "Visual Element Category", note the page where it is found, and extract any relevant text it contains (if possible).
- **No,** the statement does not describe any modern slavery remediation action:
  - Copy-paste the text passages from the statement that justify that the reporting entity does not meet this requirement,
    **OR**
  - If no information is found about this label, select the Supplementary Button: "No relevant information found in the statement".
- **Unclear,** in any other case:

- Select this label if the information found is unclear or there are other concerns. If you decide to select this label, you have to choose the Compulsory Supplementary Button "Explanation" and justify why.

**Example 2.5.3.a** (iOR Group, SID#10070):

procurement contracts and we review procurement contracts and mechanisms for any indication of unethical procurement with the aim of strengthening legal measures to ensure our partners warrant their conformity to Modern Slavery Legislation. We reserve the right to terminate any contractual agreement or partnership if there is a breach of our Modern Slavery Statement.

We have access to the employment contracts of our Philippines based staff with full visibility of their wages and employment conditions. We conduct regular videoconferencing with our staff in the Philippines and Poland

5. **How the reporting entity assesses the effectiveness of such actions;**

The effectiveness of these actions is assessed by means of:

- Awareness of Modern Slavery obligations and sensitivity to any reports of issues in this regard with any of our suppliers;
- Conducting risk-based assessments to determine which areas of the business and supply chain are most at risk from modern slavery and ensure appropriate measures are in place to eliminate the risk; and
- Enquiries to major suppliers on a regular basis.

6. **Process of consultation with any entities the reporting entity owns or controls or is issuing a joint modern slavery statement with**

The IOR Group Board of Directors have ultimate responsibility for governance and oversight of modern slavery risks and all reporting entities are controlled and operated through a single management team.

This joint statement was approved for and on behalf of SMRM Holdings Pty Ltd (SMRM), and each of the IOR Group entities listed in Annexure 1, on 21 December 2021 by the Board of Directors of SMRM.

**Annexure 1: IOR Group Entities**

| | Company | ABN |
|---|---|---|
| 1 | SMRM Holdings Pty Ltd | 85 168 760 861 |
| 2 | IOR Petroleum Pty Ltd | 36 009 653 070 |
| 3 | IOR Aviation Pty Ltd | 60 056 487 453 |
| 4 | AF Townsville Pty Ltd | 54 638 510 059 |
| 5 | IOR Energy Pty Ltd | 60 010 504 444 |
| 6 | IOR Terminals Pty Ltd | 14 057 379 067 |
| 7 | IOR Transport Pty Ltd | 66 162 200 099 |
| 8 | IOR Transport Services Pty Ltd | 15 605 338 396 |
| 9 | Fuelcharge Pty Ltd | 32 616 990 339 |
| 10 | IOR Services Pty Ltd | 46 151 774 159 |
| 11 | IOR Sales & Rentals Pty Ltd | 49 137 156 097 |
| 12 | Henty Aviation Pty Ltd | 34 238 922 481 |
| 13 | IOR Rural Holdings Pty Ltd | 33 641 448 277 |
| 14 | IOR Rural Assets Pty Ltd | 83 641 450 473 |
| 15 | IOR Rural Pty Ltd | 76 641 451 630 |
| 16 | IOR Rural Services Pty Ltd | 11 641 449 809 |
| 17 | Outback Truckstops Pty Ltd | 25 065 395 244 |

Stewart Morland
Managing Director

Ross Mackenzie
Executive Director

- Select label: **Yes**
- Add the supporting text: *We reserve the right to terminate any contractual agreement or partnership if there is a breach of our Modern Slavery Statement.*

**Example 2.5.3.b** (JFC, SID#10073):

marketing; print and promotional goods and services; fleet management; office supplies; and corporate clothing.

We establish a relationship of trust and integrity with our suppliers, which is built upon mutually beneficial factors. Our employees are required to select suppliers in accordance with our internal purchasing policy including factoring into due diligence of the supplier's reputation, respect for the law, compliance with health and safety and environmental standards.

To the present we have not been made aware of any allegations of human trafficking/slavery activities against any of our suppliers, however in the case we were, we would immediately take action against the supplier and report as such to the relevant authorities.

### Risk assessment

In order to minimise modern slavery risks, we request that our major suppliers complete a questionnaire which contains questions relevant to modern slavery and human trafficking. Responses to this questionnaire will provide us with a greater understanding of the risk of modern slavery and human trafficking within a supplier's supply chains or operations. It will also enable us to work with a supplier to reduce any identified risks. We will actively improve our understanding and oversight of all tiers of our supply chains.

Whilst the majority of our products are supplied by headquarters in Japan, we recognise that certain sectors and industries, such as agriculture, are globally recognised as high risk industries. We will further strengthen our risk assessment program through analysis of our supply chain by product category, segment and production tier.

### Actions taken

*Policies*

JFC Group has implemented a Modern Slavery Policy which covers the following topics to identify and prevent modern slavery in our operations:

1. What is the purpose of this policy?
2. Who does this policy apply to?
3. What is modern slavery?
4. What are some indicators of modern slavery practices?
5. Why we want to eliminate risks of modern slavery practices?
6. What are our responsibilities?
7. What are your responsibilities?
8. How do I report slavery concerns?
9. Who is responsible for this policy?

*Questionnaire to major suppliers*

In March 2022, the JFC Group sent letters to suppliers to seek engagement regarding assessment of modern slavery. The letter consists of a comprehensive explanatory statement about modern slavery

- Select label: **Yes**
- Add the supporting text: *To the present we have not been made aware of any allegations of*

*human trafficking/slavery activities against any of our suppliers, however in the case we were, we would immediately take action against the supplier and report as such to the relevant authorities.*

**Example 2.5.3.c** (SRWA, SID#10079):

are linked to high risk industries, are high risk products themselves or are in high risk countries. Supplier location is one of the considerations made when sourcing our fertilizer and also one of the higher risk areas regarding modern slavery. We have suppliers located in Asia, the Middle East, North America, and Europe, as well as domestically in Australia.

**Our operations**

SRWA considers the risk of modern slavery within its operations to be low because its workforces are in a low-risk country and SRWA maintains robust policies, such a whistleblowing and grievance mechanisms, which create greater transparency of modern slavery issues. There have been no significant impacts upon the existing supply chain or suppliers, other than the continued increased costs associated with sea freight for imported goods. This is due to basic supply and demand dynamics. The price of fertilizer has increased significantly, in part due factors such as the ongoing situation with Russian and Ukraine and other supply side issues.

All employees can raise concerns about modern slavery in a confidential manner. Should an occurrence or potential occurrence of modern slavery be identified, either internally related to employment practice or working conditions, or externally within the supply chain, then appropriate corrective action shall be taken in line with the Company's response plan which forms part of the Company's Modern Slavery Procedure.

**Our policies and approach to Governance**

SRWA is committed to working towards preventing and mitigating risks of modern slavery in its operations and supply chains. During this financial year the actions taken included:

- Completed initial review of all 1400 existing suppliers at the time the modern slavery policy was established:
  - risk assessment (based on geographic, product/services and industry specific risk) carried out by appropriate department executive manager
  - further desk top due diligence carried out for those with any risk indicators
  - supplier contacted and initial request for Modern Slavery policy or statement
  - supplier questionnaires sent to those suppliers deemed high risk
  - prepared clauses relating to modern slavery for new contracts with suppliers if required

2

- Review of existing overseas suppliers has 2 outstanding questionnaires. Communication regarding this is ongoing.

SRWA continues to have the following procedures in place to measure its effectiveness in ensuring that slavery is not taking place in any of its business or supply chains:

- An annual review of modern slavery statement and policies
- Communication and personal contact, where feasible, with domestic suppliers to ensure their understanding of, and compliance with our expectations
- Communication with overseas suppliers to ensure their understanding of, and compliance with, our expectations

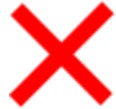

**Plan for remediation**

There were no remediation requirements for this reporting period. SRWA understands remediation is multifaceted and would include supporting suppliers through training, open discussions or involvement in industry-based initiatives.

**Our ongoing commitment**

Over the next year SRWA will focus on:

- Completion of obtaining completed questionnaires from overseas suppliers
- Review and update the Modern Slavery Policy and Procedure documents as required
- Ensure all relevant staff are fully trained on areas relating to Modern Slavery as required

This statement was approved by the Board of Summit Rural (WA) Pty Ltd on 12 September 2022

Hajime Uchiike, Chairperson, 13 September 2022

4

- Select label: **Yes**
- Add the supporting text: *Should an occurrence or potential occurrence of modern slavery be identified, either internally related to employment practice or working conditions, or externally*

*within the supply chain, then appropriate corrective action shall be taken in line with the Company's response plan which forms part of the Company's Modern Slavery Procedure.* // *SRWA understands remediation is multifaceted and would include supporting suppliers through training, open discussions or involvement in industry-based initiatives.*

- **Note:** the sentence "*There were no remediation requirements for this reporting period.*" is not extracted because it does not actually state what remediation strategies would be and so it is not relevant here.

## 2.5.4. Assessing the effectiveness of actions

**Question:** does the reporting entity describe how it assesses the effectiveness of its actions?

**Background:** the reporting entity should describe in its statement how it **assesses the effectiveness of actions** taken to **mitigate risks** as well as to **remediate modern slavery cases**. This is important to ensure that the entity knows whether these actions are actually helping combat modern slavery.

The supporting text for this criteria should relate to the examination of whether the measures and processes implemented by the reporting entity are successful in achieving their intended goals related to mitigating risks and remediating cases of modern slavery. In practical terms, this means evaluating how well the company's strategies and initiatives are performing in practice. This assessment often involves tracking key performance indicators, monitoring the outcomes of risk mitigation efforts, seeking feedback from stakeholders, and conducting regular reviews to ensure that the actions in place align with the company's commitment to address modern slavery.

Specifically, assessing the effectiveness of actions can be accomplished by:
- **Examining** or **revising** action plans or policies, such as by forming a review committee or conducting annual assessments, to ensure that:
  - they remain up to date, particularly when operating in high-risk countries, regions, or industries;
  - they adhere to the internal organization guidelines regarding application deadlines or in terms of uniformity in their application; or
  - they have the intended results (e.g. in terms of dissuasive power).
- Setting up a **feedback process** between key departments (or across owned or controlled entities) to facilitate regular engagement and communication among procurement, human resource management, and legal teams;
- Engaging with **industry groups**, **external auditors**, **trusted NGOs**, or **stakeholders** (such as the workers themselves) to provide **feedback** and/or conduct an **independent review** of actions;
- Implementing **internal checks** to evaluate the efficiency of the actions of risk identification, mitigation or remediation processes;
- Reviewing **trends in reported modern slavery cases** through grievance or whistleblowing mechanisms and evaluating the handling of these cases;
- Reporting and monitoring **Key Performance Indicators (KPIs)** that measure the effectiveness of actions, such as:
  - The number of contracts in which modern slavery cases were identified;
  - The number of trainings delivered to employees, suppliers, or other stakeholders;
  - The awareness of employees, suppliers, or other stakeholders towards modern slavery as measured through polls;
  - The number of initiatives taken to enhance suppliers' capacity to respond to modern slavery risks;

- ○ The number of filed grievances;
- ○ The resolution rates of reports filed through grievance or whistleblowing mechanisms;
- ○ The number of completed audits;
- ○ The number of visits in facilities operated by the reporting entity, by one of its owned or controlled entities, or by a third-party;
- ○ The number of assigned and completed corrective action plans;
- ○ The number of contracts canceled due to identified modern slavery cases;
- ○ The number of identified modern slavery incidents.

**Note:** entities will often state that they "*improved their modern slavery risk mitigation*" over a reporting period, but this is not useful information for this question. Supporting text for this question should instead detail how the reporting entity determined that its actions were effective or not effective. For example, if the reporting entity stated that it "*improved its modern slavery risk mitigation actions following a comprehensive review process*", then it can be considered relevant for this question.

**Label options & their associated supporting text passages extracted from the statements:**

- ○ **Yes**, the statement describes how the reporting entity assessed the effectiveness of its actions:
  - ● Copy-paste the text passages from the statement that describe how the reporting entity assesses the effectiveness of its actions.
  - ● If any relevant information comes in other formats than text, fill in the required information via the Supplementary Button: "Visual Element Category", note the page where it is found, and extract any relevant text it contains (if possible).
- ○ **No,** the statement does not describe how the reporting entity assessed the effectiveness of its actions:
  - ● Copy-paste the text passages from the statement that justify that the reporting entity does not meet this requirement,
    **OR**
  - ● If no information is found about this label, select the Supplementary Button: "No relevant information found in the statement".
- ○ **Unclear,** in any other case:
  - ● Select this label if the information found is unclear or there are other concerns. If you decide to select this label, you have to choose the Compulsory Supplementary Button "Explanation" and justify why.

**Example 2.5.4.a** (JFC, SID#10073):

and 30 questions to the suppliers for assessment of risk of modern slavery. The letter and questionnaire were sent to suppliers whose trading amount with JFC Group in the financial year 1ˢᵗ April 2021 to 31ˢᵗ March 2022 exceeded AU $7,000,000 (gross).

*Training*

JFC Group implemented the Modern Slavery Policy which is incorporated within the company's Employment Policy. The Modern Slavery Policy covers:

- Various forms of modern slavery in which people can be held and exploited
- The scale of the problem and the risk to our organisation
- How employees can identify the signs of slavery and human trafficking, including unrealistically low prices
- How employees should respond if they suspect slavery or human trafficking
- What external help is available for the victims of slavery

All JFC Group employees are provided with the Modern Slavery Policy. Each employee must read the Policy and prove their understanding of our role and their responsibilities under the Policy by declaring so in writing or via JFC's employee portal system.

JFC Group has defined a set of key performance indicators and controls to combat modern slavery and human trafficking in our organisation and supply chain. These include:

- How many employees have completed mandatory training by completing active reading of Modern Slavery Policy?
- What percentage of suppliers have filled out our ethics questionnaire?
- How many reports have been made by our employees that indicate their awareness of and sensitivity to ethical issues?

JFC Group will collect records of these numbers each year to assess our performance annually. The first assessment date is 1ˢᵗ April 2023 for the fiscal year of 1ˢᵗ April 2022 to 31ˢᵗ March 2023.

## Consultation with subsidiary

JFC Australia has regular meetings with JFC New Zealand and ensures that JFC New Zealand is aware of our policy on modern slavery. JFC Australia invites JFC New Zealand to participate in the preparation of this statement to raise awareness within the JFC Group and gain insight from them.

This statement covers the period between 1ˢᵗ April 2021 to 31ˢᵗ March 2022 and has been approved by the board of JFC Australia Co Pty Ltd at the board meeting held on 18ᵗʰ March 2022.

[Signature]                          [Date]        15/09/2022

*MANAGING DIRECTOR*

Shigeru Ogawa
Managing Director

- Select label: **Yes**
- Add the supporting text: *JFC Group has defined a set of key performance indicators and controls*

*to combat modern slavery and human trafficking in our organisation and supply chain. These include:*

*· How many employees have completed mandatory training by completing active reading of Modern Slavery Policy?*

*· What percentage of suppliers have filled out our ethics questionnaire?*

*· How many reports have been made by our employees that indicate their awareness of and sensitivity to ethical issues?*

*JFC Group will collect records of these numbers each year to assess our performance annually. The first assessment date is 1st April 2023 for the fiscal year of 1st April 2022 to 31st March 2023.*

**Example 2.5.4.b** (SRWA, SID#10079):

> o communication with suppliers, electronically or face to face if possible, to discuss modern slavery as required
> - All new suppliers were reviewed for modern slavery risks, as per the onboarding system, and the same process followed as mentioned above for existing suppliers
> - Reported on modern slavery project status and statistics to executive management team
> - Reported on statistics captured per ERP system on number of new suppliers and number of completed modern slavery reviews
> - Team meeting to discuss/continue review of modern slavery processes and effectiveness
>
> The initial review started in 2020 and is now completed and all suppliers have been contacted and desk top due diligence work carried out for each supplier. For those still deemed a higher risk after the desk top due diligence, an additional questionnaire has been sent to the supplier. There are 2 outstanding questionnaires from overseas suppliers. These suppliers cover both product supply and freight services supply. For all other suppliers where the initial review has been completed, there have been no reports of breaches with the Modern Slavery Act.
>
> In addition, the principal objectives of SRWA's Modern Slavery policy are delivered through associated policies and procedures, such as:
> - Compliance Policy and Compliance Manual
> - Whistleblowing Policy and Whistleblowing Procedure
>
> The policies apply to all people working for or behalf of SRWA in any capacity, including employees, directors, officers, agency workers, contractors, consultants, and any other third-party representatives. SRWA is committed to ensuring all staff are kept up to date with ongoing training and updates in all areas stated above.
>
> **Assessing effectiveness**
> The effectiveness of steps taken to address modern slavery risks in our business and supply chains have been assessed for the current reporting period.
>
> 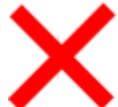 - Processes for new suppliers when onboarding are working well and have assisted in engagement and awareness of modern slavery issues for both staff and suppliers.
>
> 3

- Review of existing overseas suppliers has 2 outstanding questionnaires. Communication regarding this is ongoing.

SRWA continues to have the following procedures in place to measure its effectiveness in ensuring that slavery is not taking place in any of its business or supply chains:

- An annual review of modern slavery statement and policies
- Communication and personal contact, where feasible, with domestic suppliers to ensure their understanding of, and compliance with our expectations
- Communication with overseas suppliers to ensure their understanding of, and compliance with, our expectations

### Plan for remediation

There were no remediation requirements for this reporting period. SRWA understands remediation is multifaceted and would include supporting suppliers through training, open discussions or involvement in industry-based initiatives.

### Our ongoing commitment

Over the next year SRWA will focus on:

- Completion of obtaining completed questionnaires from overseas suppliers
- Review and update the Modern Slavery Policy and Procedure documents as required
- Ensure all relevant staff are fully trained on areas relating to Modern Slavery as required

This statement was approved by the Board of Summit Rural (WA) Pty Ltd on 12 September 2022

Hajime Uchiike, Chairperson, 13 September 2022

4

- Select label: **Yes**

- Add the supporting text: *Reported on modern slavery project status and statistics to executive management team*
*• Reported on statistics captured per ERP system on number of new suppliers and number of completed modern slavery reviews*
*• Team meeting to discuss/continue review of modern slavery processes and effectiveness //* • *An annual review of modern slavery statement and policies*
- **Note that the following text does not actually describe how effectiveness is assessed ("working well" is vague):** *Processes for new suppliers when onboarding are working well and have assisted in engagement and awareness of modern slavery issues for both staff and suppliers.*

**Example 2.5.4.c** (Mitsui-Itochu, SID#10088):

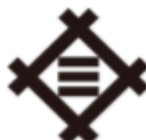

**Mitsui-Itochu Iron Pty Ltd**

**MODERN SLAVERY STATEMENT 2022**

**5. Assessing effectiveness and looking ahead – s 16(1)(e) of the Act**

We are assessing the effectiveness of actions by:

- monitoring the results of our supplier checks and continuously undertaking periodic reviews on the relevancy of these checks; and
- working together with Mitsui and MCA to benefit from their reflections on their own human rights due diligence efforts.

In the financial year ending 31 March 2023, we intend to continue our efforts on our suppliers checks to assess for modern slavery risks, analyse the check results and engage with relevant stakeholders (e.g., Board of Directors, President & Managing Director, General Managers) to discuss what appropriate actions can be taken, if required.

We recognise that the risks of modern slavery are complex and evolving. We will continue to work to address these risks in our business and engage with a range of stakeholders to assess the effectiveness of our actions and enhance our approach. 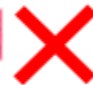

I, Motoi Yamamoto, hereby certify that this Modern Slavery Statement 2022 has been approved by the Board of Directors of Mitsui-Itochu Iron Pty Ltd on 19 September 2022.

*Motoi Yamamoto*

**MOTOI YAMAMOTO**
President & Managing Director
19 September 2022

- Select label: **Yes**

- Add the supporting text: *We are assessing the effectiveness of actions by:*
  *• monitoring the results of our supplier checks and continuously undertaking periodic reviews on the relevancy of these checks; and*
  *• working together with Mitsui and MCA to benefit from their reflections on their own human rights due diligence efforts.*
- **Note:** the sentence *"We will continue to work to address these risks in our business and engage with a range of stakeholders to assess the effectiveness of our actions and enhance our approach."* is not extracted because "engage with a range of stakeholders" is too vague (i.e. the action following the engagement is not described).

**Annotation Workflow:**

**The labels and their supplementary buttons for the questions related to Criteria 3 to 5 should be:**

### 2.5.1. Risk description

- Yes:
    - Copy-Paste supporting text passages
    - Supplementary Button and fields for visual elements
- No:
    - Copy-Paste supporting text passages
    - Supplementary Button: "No relevant information found in the statement"
- Unclear:
    - Compulsory supplementary text field for explanations

### 2.5.2. Risk mitigation

- Yes:
    - Copy-Paste supporting text passages
    - Supplementary Button and fields for visual element(s
- No:
    - Copy-Paste supporting text passages
    - Supplementary Button: "No relevant information found in the statement"
- Unclear:
    - Compulsory supplementary text field for explanations

### 2.5.3. Modern slavery remediation

- Yes:
    - Copy-Paste supporting text passages
    - Supplementary Button and fields for visual elements
- No:
    - Copy-Paste supporting text passages
    - Supplementary Button: "No relevant information found in the statement"
- Unclear:
    - Compulsory supplementary text field for explanations

### 2.5.4. Effectiveness of actions

- Yes:
    - Copy-Paste supporting text passages
    - Supplementary Button and fields for visual elements
- No:
    - Copy-Paste supporting text passages
    - Supplementary Button: "No relevant information found in the statement"
- Unclear:

> ■ Compulsory supplementary text field for explanations

## 2.6. Mandatory Criterion 6: Process of consultation

**Question:** does the reporting entity describe how it consulted on its statement with any entities it owns or controls?

**Background:** this mandatory reporting criterion requires statements given by reporting entities to describe how they **consulted** on the preparation of their statement with any entities they own or control. If the reporting entity **owns or controls** other entities (see Section 2.4.1), the reporting entity **must** describe their process of consultation. Proper consultation ensures that all relevant areas of the reporting entity and any entities it owns or controls are aware of what actions they need to take and that modern slavery risks relating to these areas and entities have been identified, assessed, and addressed. If the reporting entity **does not own or control** other entities, it does not need to describe their process of consultation, but it needs to explicitly indicate so in the statement**.**

**Notes:**

- If the reporting entity states that it has consulted with **any** owned or controlled entities, the label for this question should still be "Yes".
- If the reporting entity has stated that it does not own or control other entities (as already extracted for Criterion 2 in Section 2.4.1), then the label for this question should be "No".

**Label options & their associated supporting text passages extracted from the statements:**
- **Yes**, the reporting entity identified the process of consultation with any entities it owns or controls:
    - Copy-paste the exact text passages from the statement that shows that the reporting entity consulted with any entities it owns or controls.
    - If any relevant information comes in other formats than text, fill in the required information via the Supplementary Button: "Visual Element Category", note the page where it is found, and extract any relevant text it contains (if possible).
- **No**, the reporting entity did not identify the process of consultation with any entities it owns or controls, or it did not need to consult because it does not own or control other entities (or for any other reason described by the entity):
    - Copy-paste the exact text passages from the statement that shows that the reporting entity does not meet or that it is not required to meet this criterion,
         **OR**
    - If no information is found about this label, select the Supplementary Button: "No relevant information found in the statement".
- **Unclear**, in any other case:

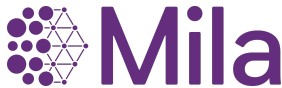

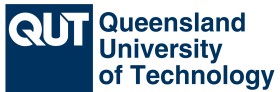

> - Select this label if the information found is unclear or there are other concerns. If you decide to select this label, you have to choose the Compulsory Supplementary Button "Explanation" and justify why.

---

**Annotation Workflow:**
**The labels and their supplementary buttons should be:**

**Consultation on the preparation of the statement:**
- Yes:
    - Copy-Paste supporting text passage
    - Supplementary Button: "Visual Element Category"
- No:
    - Copy-Paste supporting text passage
    - Supplementary Button: "No relevant information found in the statement"
- Unclear:
    - Compulsory Supplementary Button "Explanation"

---

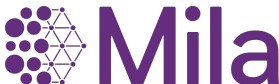
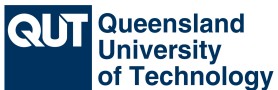

**Example 2.6.a** (BIC Services, SID#1693)**:**

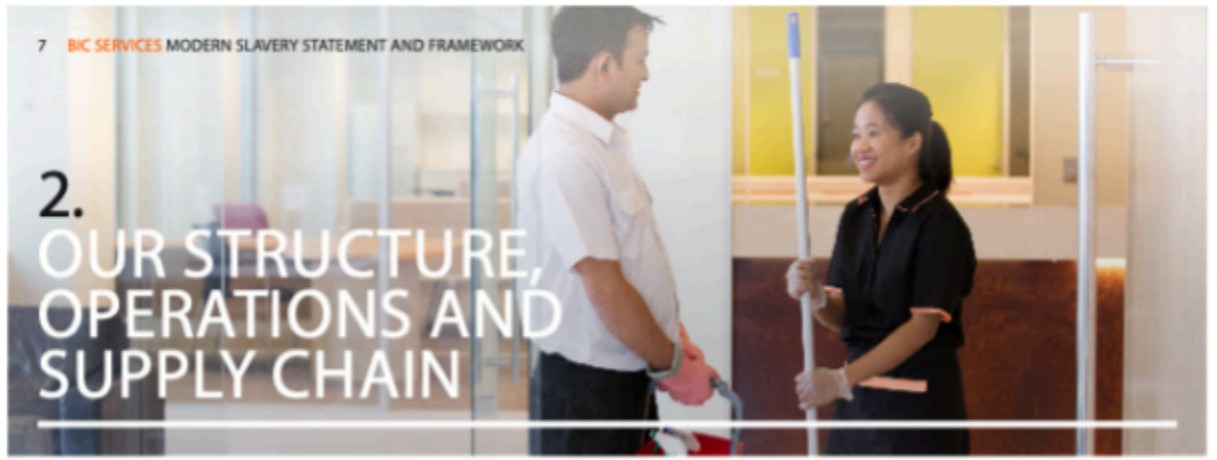

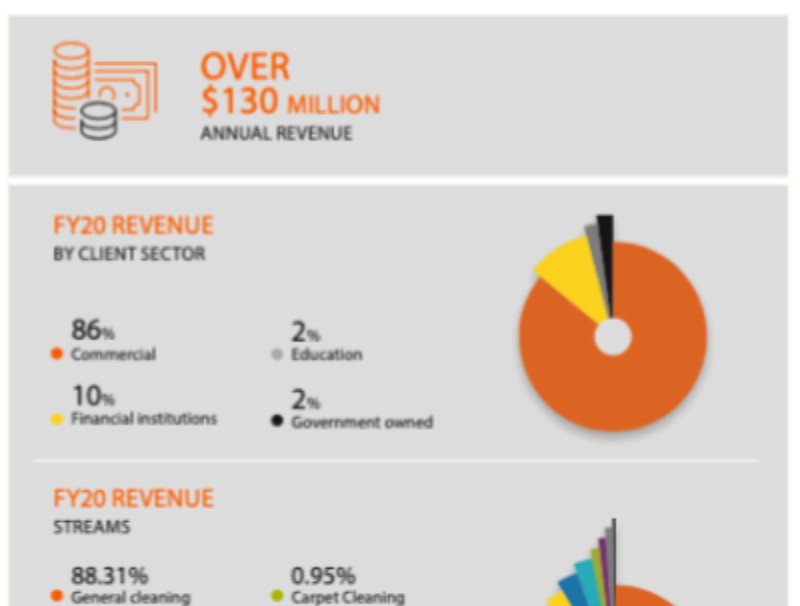

- ■ Select label: **No**
- ■ Select the Supplementary Button "No relevant information found in the statement", as the entity did not explain why it did not consult.

**Example 2.6.b** (PVH+SunshineA, SID#320)**:**

> The following PVH entities are required to submit a statement under the Australian Commonwealth Modern Slavery Act: PVH Brands Australia and Sunshine A Pty Ltd. (the "reporting entities"). This Statement is a joint statement. It has been approved on behalf of the reporting entities by the CR Committee of the PVH Board of Directors, which is the principal governing body of the higher entity parent company of the reporting entities, and the undersigned is a responsible member of the parent company. ==This Statement was prepared in consultation with the reporting entities, including by making those entities aware of its preparation and providing them with an opportunity to participate therein.== ❌

*[signature: Emanuel Chirico]*

Emanuel Chirico
Chairman & Chief Executive Officer

October 30, 2020

- ■ Select label: **Unclear**
- ■ Add to the Compulsory Supplementary Button "Explanation": *The reporting entities are PVH Brands Australia and Sunshine A Pty Ltd. It is unclear if they own or control any other entities.*
- ■ Note: the text indicates that the reporting entities were consulted in the preparation of the document. This however is not supporting text for this criterion, which is about the entities owned or controlled by the reporting entities, not the reporting entities themselves.

**Example 2.6.c** (Mineral Resources Limited, SID#1674):

**APPENDIX 2:** SUBSIDIARY COMPANIES AND JOINT ARRANGEMENTS

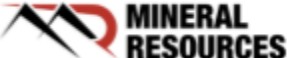

Mineral Resources Limited (ABN 33 118 549 910) is the parent entity and is registered at 1 Sleat Road, Applecross, Western Australia, 6153. For further information, refer to the MRL 2020 Annual Report.

**MATERIAL SUBSIDIARIES**

MRL has material subsidiaries that support our mining services and commodities businesses, by undertaking the following activities:

- Mine Development
- Mining
- Crushing, screening and processing
- Facilities: Mine accommodation villages, utilities and airports
- Logistics: Road, rail and port

MRL incorporates the assets, liabilities and results of the following material subsidiaries in accordance with its accounting policies. Most of these entities have only been established for financial reasons and all entities follow policies set by the reporting entity, MRL. As such, a detailed consultation process is not necessary. All of MRL's operations are in Australia.

| Name | Country of incorporation | Ownership interest 2020 % | 2019 % |
|------|-------------------------|---------------------------|--------|
| Crushing Services International Pty Ltd | Australia | 100.00% | 100.00% |
| Mesa Minerals Limited | Australia | 59.40% | 59.40% |
| PIHA Pty Ltd | Australia | 100.00% | 100.00% |
| Polaris Metals Pty Ltd | Australia | 100.00% | 100.00% |

- Select label: **No**
- Add the supporting text: *MRL incorporates the assets, liabilities and results of the following material subsidiaries in accordance with its accounting policies. Most of these entities have only been established for financial reasons and all entities follow policies set by the reporting entity, MRL. As such, a detailed consultation process is not necessary.*"

**Example 2.6.d** (Cerner, SID#5508):

the supplier poses with respect to modern slavery and other unlawful business practices. The Supply Chain Team also identifies issues, communicates to executive leadership, escalates for decision-making purposes and supports the overall negotiation and communication process throughout the supplier engagement, as appropriate. Additionally, our supplier agreements include language that requires all suppliers to comply with all applicable laws in the country in which they are operating, and new supplier agreements will include reference to compliance with applicable laws regarding human trafficking, forced labor and other forms of modern slavery. To date, we have not identified or been informed of any incidents of slavery and human trafficking within our business or supply chain.

**Impacts of COVID-19 Pandemic**

Our focus on health and safety and human rights has continued through the challenges of the COVID-19 pandemic. Throughout the COVID-19 pandemic, we have acted to support our associates (employees), clients and communities during these challenging times, responded to the associated business impact and developed processes to ensure we were meeting the needs of our clients. These actions have included:

- Transitioning all associates whose work allowed to a virtual environment.
- Mobilizing a taskforce to provide work-from-home and travel guidance based on information from the World Health Organization, the U.S. Centers for Disease Control and other global health agencies and input from provider clients.
- Enhancing health, wellness and family resources for our associates.
- Supporting our communities with community giving, including donations of meals for healthcare professionals and first responders, personal protective equipment and hand sanitizer for students.
- Hosting a series of mass COVID-19 vaccination events at our World Headquarters through a collaboration with North Kansas City Hospital, Liberty Hospital, Clay County Public Health Center and cities within Clay County.
- We have implemented social distancing measures as we plan towards re-introducing colleagues back into our business premises once it is safe to do so in line with applicable guidance.

We are continuously monitoring the situation and reviewing our existing measures in line with applicable government guidance. The Boards of Directors of all Cerner entities have been kept up to date throughout the pandemic regarding risk levels and mitigation plans.

**Consultation**

In order to prepare this statement, Cerner Corporation engaged with Cerner Limited and Cerner Corporation Pty Limited and consulted with them about the contents of this statement.

- ■ Select label: **Yes**
- ■ Add the supporting text: *In order to prepare this statement, Cerner Corporation engaged with Cerner Limited and Cerner Corporation Pty Limited and consulted with them about the contents of this statement.*

# Appendix 1: Project AIMS Annotation Cheat Sheet

**Important note:** this cheat sheet is **not** meant to be a replacement of the official specifications. It is instead provided as a guide to help interpret terminology and actions commonly found in statements as relevant or not.

All of the examples given here are obviously relevant or irrelevant as supporting text for the label of a given criterion. No extra context is needed to assert the relevance of these examples, but this is not always the case in practice. In real statements, always extract neighboring sentences ("context") if those extra sentences are required to understand the meaning/relevance of the supporting text.

Remember also that a single sentence may be relevant to multiple criteria, so it may be extracted and used in multiple supporting text boxes.

## C2 - Structure

**Examples of relevant text:**
- Description of "corporate structure", "governance structure", or "governance framework"
- Description of the company type (e.g. "publicly traded", "partnership", "trust", …)
- Description of a parent group or of a parent company that owns the reporting entity (e.g. "we are wholly owned by…", or "our holding company is…")
- Description of owned or controlled entities (including descriptions of their structure)
- The number of employees (either overall, or at specific locations, and can be an estimate)
- Owned brand names, or trading names (trademarks)
- The registered office location of the entity (its "statutory address", or its headquarters location)

**Examples of irrelevant text:**
- Giving the full name of the company (e.g. XYZ Pty Ltd)
- Giving the ABN (e.g. ABN 01548756523)
- "Established in YYYY."
- "We operate in Australia", or any other location-only information that is not associated with its statutory address/headquarters
- The number of clients or consumers the company has (or any information about those)
- "We make this statement on behalf of our associated entities." or "This statement covers our wholly-owned subsidiaries." (too vague)
- "We work in the XYZ sector with offices located in 123 countries."

## C2 - Operations

**Examples of relevant text:**
- Description of the work done by or duties of employees (e.g. "We hire researchers…", "We employ accountants…", "Our drivers deliver goods…")
- Description of an activity of the company (e.g. "Our company builds X…", "We provide Y…", "123 Pty Ltd manufactures Z…")

- Description of the facilities (or offices) of the company where operations take place, where the description might be related to location ("We operate in asia…") or of the role of the facility/office
- Description of operations for a joint venture of the company

**Examples of irrelevant text:**
- "Company XYZ trades on the New York Stock Exchange" (this does not mean that the company has a trading activity itself, instead it means that its shares are traded)
- Description of operations of a PARENT company, or of a PARTNER company (only operations of the reporting entity and of its owned/controlled entities are relevant)
- Description of operations for an entity in which the reporting entity is invested

# C2 - Supply Chains

**Examples of relevant text:**
- Description of procured goods or services (e.g. from raw materials processed by parent companies down to office supplies used by the reporting entity itself, and from out-of-country labour-for-hire services down to local office cleaning services)
- Analysis of spending categories or sectors
- Description of suppliers (e.g. their location, their business relationship, their structure or operations)
- A description of their supply chain itself (what they consider direct or first-tier suppliers, what they consider indirect or second-tier suppliers, etc.)

**Examples of irrelevant text:**
- Description of the downstream supply chain, i.e. the supply chain towards customers who procure goods or services from the reporting entity (this is more relevant for operations, as it describes the distribution network of the reporting entity)
- "We continue due diligence across all our direct and indirect suppliers." (missing supplier description)
- "We procure goods and services from trusted suppliers across the world." (too vague)

# C3 - Risk Description

**Examples of relevant text:**
- Description of how the reporting entity's operations or supply chain could be involved (or could contribute to) modern slavery
- Description of an actual case of modern slavery found in the reporting entity's operations or supply chains (but NOT a hypothetical/fictitious case)
- Description of suppliers that are considered higher-risk (due to their location, product/service categories, past cases, etc.)
- Description of procured goods that are more commonly tied to modern slavery (e.g. conflict minerals, cocoa, palm oil, electronics)
- Description of events or changes that could increase the risk of modern slavery (e.g. having to rely on unvetted suppliers due to COVID-19 shortages, having missed scheduled audits due to travel restrictions)

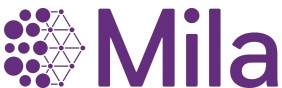

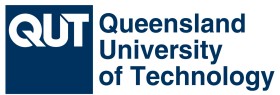

- "There is a possibility that our suppliers do not adhere to our standards" (slightly vague, but relevant if "standards" have been clearly defined with respect to modern slavery obligations)
- "A minor portion of our goods originate from X, which is a region where modern slavery is known to occur."
- "Due to XYZ reasons, we consider that our risk is low." (if they do not provide at least one reason, it should not be extracted, as it is too vague)
- "Because of XYZ, there is no identifiable risk in our operations." (negative evidence is still relevant)

**Examples of irrelevant text:**
- "Modern slavery has the potential to exist in the technology sector." (too vague, not related enough to operations or supply chains)
- "Modern slavery is defined as forced labor, debt bondage, or servitude." (generic description, not tied to their operations or supply chains)
- "We have a low risk of modern slavery." or " Our operations or supply chains are slavery-free." (too vague, they need to define why, or based on what analysis)
- A description of a hypothetical or fictitious case of modern slavery
- A description of other business risks without clearly explaining how they are related to modern slavery (e.g. health risks (due to hazardous environments or materials), environmental risks, etc.).

# C4 - Risk Mitigation Actions

**Examples of relevant text:**

- Actions to identify, analyze, categorize, scope, or assess modern slavery risks in the company's operations or supply chains (as detailed for C3), for example conducting audits of internal operations, suppliers, or procurement activities (e.g. via questionnaires, surveys, research, or interviews), or tasking a workgroup or an external company to conduct such activities
- Actions to adopt, draft, update, or uphold a policy, framework, standard, or program that is meant to mitigate risks of modern slavery in the reporting entity's operations or procurement decisions
- Actions to launch, participate, contribute, or engage with an industry initiative to combat modern slavery (e.g. the UN Global Compact)
- Actions that require employees to follow company guidelines regarding human rights, labour rights, association (or unionization) rights, responsible recruitment, and/or responsible procurement (e.g. adoption of a code of conduct, or a code of ethics that are clearly linked with modern slavery)
- Actions that require suppliers to follow company guidelines, policies, or labor laws (e.g. adoption of a supplier code of conduct or contract terms)
- Actions to ensure that stakeholders are aware of the company's policies and requirements (e.g. conducting training sessions on modern slavery or on policies)
- Actions to ensure that modern slavery cases or concerns are reported and investigated (e.g. adopting a whistleblowing hotline or any other reporting mechanism)
- Actions involving company leaders and/or board members in the elaboration of action plans or in action oversight committees

**Examples of irrelevant text:**

- "Company XYZ recognizes the importance of acting to reduce the risk of modern slavery in its operations and supply chains", or "We are committed to maintaining the highest level of integrity and honesty throughout all aspects of our business." (there is no description of a real action here)
- "Our approach towards modern slavery is zero-tolerance." (unless they have a policy or enforce this approach somehow, this is too vague)
- "We respect human rights and expect the same from our suppliers", or "We expect our suppliers to respect local laws." (the "expectation" is too weak since there is no description of a requirement or enforcement action towards suppliers; would be OK if this was for example enforced through a supplier code-of-conduct or through contract terms)
- "We have adopted a new sustainability policy" or "We have adopted a new worker safety policy." (irrelevant unless they clearly state that their definition of "sustainability" or "worker safety" covers modern slavery elsewhere in the statement — same reasoning applies to "worker health", "diversity", "discrimination", and "harassment" policies)
- "We follow the law." (irrelevant, as this is always expected of the reporting entities)
- Description of audit measures for processes and actions, not risks, meant to prevent modern slavery (this should be extracted in C5, assessing the effectiveness of actions)
- Description of the actual tools (e.g. dashboards, software) used to monitor and track modern slavery risk (only the development and use of those tools are actions and therefore relevant to extract)
- Overall, information that does not refer to the **action** of mitigating risks.

# C4 - Modern Slavery Remediation Actions

**Examples of relevant text:**
- Description of actions taken (or that would be taken in a potential case) to inquire and investigate when a concern is voiced through a whistleblower hotline (or through any other reporting mechanism)
- Description of actions taken (or that would be taken in a potential case) when victims of modern slavery are identified (e.g. helping them file legal claims, offering compensation)
- Description of actions taken (or that would be taken in a potential case) when suppliers are found to be in breach of their obligations (e.g. contract termination, report to authorities)
- Description (or adoption) of a response plan to enact in the event that a modern slavery incident occurs
- Formal apologies made as a result of a modern slavery case found in the company's operations or supply chains

**Examples of irrelevant text:**
- "We understand the importance of workers knowing their rights and addressing violations when necessary." (no action described here)
- "Remediation was not needed in this reporting period." (no action described here)
- Overall, information that does not refer to the **action** of remediation of existing or potential cases of modern slavery.

# C5 - Assessing the Effectiveness of Actions

**Note: an important distinction between C5 (effectiveness) and C4 (mitigation) is that the actions in C5 (effectiveness) are taken on top of C4 (mitigation) actions. In other words, the effectiveness of mitigation actions is evaluated in a second "review" stage. Pay close attention to the keywords below and to the situations where they apply.**

**Examples of relevant text:**
- Actions to review, examine, or revise action plans or policies that were established to mitigate modern slavery risks (e.g. setting up a team to make sure actions and policies are up-to-date, adhere to expectations, and have the intended results)
- Actions to improve engagement between stakeholders across the company, between the company and its suppliers, or between the company and external organizations (e.g. industry groups, trusted NGOs) to exchange ideas and provide feedback on actions
- Actions to monitor and review trends in modern slavery cases or concerns reported via grievance mechanisms
- Description of internal processes that check whether planned mitigation or review actions have been taken when necessary
- Description of Key Performance Indicators (KPIs) that measure the effectiveness of mitigation actions (e.g. number of employee training conducted, grievances reported, supplier questionnaires received, completed audits, etc.)
- "We will review and update our modern slavery policy."
- "We continuously evolve and adapt our supplier audit program to support a dynamic economic environment and business needs."
- "Our Code of Conduct undergoes an annual internal review process by our audit team."(if the Code of Conduct includes clauses relevant to modern slavery).

**Examples of irrelevant text:**
- "We have improved our modern slavery risk mitigation over the last reporting period." (they do not describe how they improved it, or how they observed the improvement)
- "We will continue to work to address risks and engage with a range of stakeholders to assess the effectiveness of our actions." (too vague, did not identify stakeholders)
- "We will monitor the risk of new suppliers when onboarding them." (this is mitigation, not effectiveness)
- "We will continue to be alert in regards to identifying new exploitation practices within our supply chains." (too vague)
- "We have established an audit team." (too vague, need to mention what kind of audits they are doing; it could be mitigation or effectiveness)
- "We will develop key performance indicators to ensure the robustness of our monitoring process." (too vague, should describe the KPIs)
- Overall, information that does not clearly explain **how** the company assesses the effectiveness of actions.

# C6 - Consultation

**Examples of relevant text:**
- Actions to engage with different stakeholders across owned or controlled entities to make them aware of policies, obligations, and requirements detailed in the statement
- Actions to engage with different stakeholders across owned or controlled entities to obtain feedback on the statement
- A mention that the reporting entity does not own or control any other entity, and therefore that the consultation process is not necessary
- "All our owned and controlled entities share the same governance structure, and therefore all of them were consulted in the preparation of this statement."
- "We consulted with the principal governing body of all our subsidiaries in the preparation of this statement."

**Examples of irrelevant text:**
- "This statement was prepared in consultation with all reporting entities." (the statement should be prepared in consultation with owned/controlled entities, not reporting entities, which are themselves making the statement)
- "We do not own or control any entities." (they do not mention the consultation process at all here, so no need to extract this sentence as supporting text for the C6-"No" label)
- Overall, information that does not clearly link the consultation with owned or controlled entities to the development of **the modern slavery statement/document**.